# The conformational space of RNase P RNA in solution

Yun-Tzai Lee[1,4], Maximilia F. S. Degenhardt[1,4], Ilias Skeparnias[2], Hermann F. Degenhardt[1], Yuba R. Bhandari[1], Ping Yu[1], Jason R. Stagno[1], Lixin Fan[3], Jinwei Zhang[2] & Yun-Xing Wang[1✉]

RNA conformational diversity has fundamental biological roles[1–5], but direct visualization of its full conformational space in solution has not been possible using traditional biophysical techniques. Using solution atomic force microscopy, a deep neural network and statistical analyses, we show that the ribonuclease P (RNase P) RNA adopts heterogeneous conformations consisting of a conformationally invariant core and highly flexible peripheral structural elements that sample a broad conformational space, with amplitudes as large as 20–60 Å in a multitude of directions, with very low net energy cost. Increasing Mg²⁺ drives compaction and enhances enzymatic activity, probably by narrowing the conformational space. Moreover, analyses of the correlations and anticorrelations between spatial flexibility and sequence conservation suggest that the functional roles of both the structure and dynamics of key regions are embedded in the primary sequence. These findings reveal the structure–dynamics basis for the embodiment of both enzymatic precision and substrate promiscuity in the RNA component of the RNase P. Mapping the conformational space of the RNase P RNA demonstrates a new general approach to studying RNA structure and dynamics.

RNA conformational dynamics have essential roles in splicing[1], packaging[2], cellular transactivation[3] and responding to environmental stimulation[4,5]. Indeed, ample experimental evidence[1,6–18], including from single molecule-based approaches[5,19–21], indicates that RNA molecules exist in a much broader conformational space than is depicted by a small number of static snapshots[22,23]. Various techniques have been applied to study RNA conformational diversity and dynamics, which probe either ensemble behaviours[10,12,24–27] or characterize single molecules using sparse distances[21,28,29]. None of those techniques results in the determination of explicit three-dimensional (3D) topological structures of individual conformers, which are required for mapping the conformational landscape that defines not only the motion directions and amplitudes of structural elements or domains, but also the invariant structural core essential for folding and function. Such information may reveal the correlation between sequence conservation and conformational dynamics, which could be used to identify possible druggable pockets and to elucidate the roles of RNAs in interaction with other biomacromolecules.

Ribonuclease P (RNase P) is the only *trans*-acting RNA enzyme found in all kingdoms of life, a multiturnover ribozyme that edits the 5′ ends of precursor tRNA (pre-tRNA) and other RNAs[10,30,31]. Human RNase P processes the metastasis-associated lung adenocarcinoma transcript 1 (MALAT-1), which is upregulated in various cancers[32], to remove the tRNA-like module. Bacterial RNase P regulates gene expression via cutting in specific intergenic regions of a bacterial polycistronic operon in mRNA[33,34]. It is promiscuous in terms of substrate specificity and can process pre-tRNA as well as other RNA substrates[35–37]. The ability to recognize various substrates suggests conformational flexibility and adaptivity, which appear to be dialectical with the required structural rigidity for its enzymatic activity. This duality underlines an important general question about the relationship between the conformational flexibility and the structural rigidity of various domains, whose interplay is relevant to the functions and general roles of this RNA, and other RNAs in general. Elucidating this complex relationship requires a quantitative ensemble description of the full conformational space sampled by the inherently dynamic RNA that is far beyond what can be offered by a few structural snapshots. In this study, we report mapping of the conformational space of the full-length RNase P RNA from *Geobacillus stearothermophilus* (*Gst*)[38] under solution conditions using atomic force microscopy (AFM) images of individual molecules and machine learning approaches. Our results provide a structural dynamics basis for the substrate promiscuity and catalytic inefficiency of this RNA through sampling of a wide conformational space, and, at the same time, demonstrates a new approach to interrogate RNA conformational dynamics. Importantly, mapping of the conformational landscape of an RNA through direct observation at the single molecule level has not previously been demonstrated.

## Visualizing conformational heterogeneity

Solution AFM provides a shotgun approach for studying RNA conformational heterogeneity by capturing the topography of many individual particles at once, each of which may exist in a distinct conformation. These individual snapshots, therefore, represent a sampling of the conformational space of the RNA at a time when the molecules are immobilized on the mica surface. AFM topographical images of individual

[1]Protein–Nucleic Acid Interaction Section, Center for Structural Biology, National Cancer Institute, Frederick, MD, USA. [2]Laboratory of Molecular Biology, National Institute of Diabetes and Digestive and Kidney Diseases, Bethesda, MD, USA. [3]Leidos Biomedical Research, Inc., Frederick, MD, USA. [4]These authors contributed equally: Yun-Tzai Lee, Maximilia F. S. Degenhardt. ✉e-mail: wangyunx@mail.nih.gov

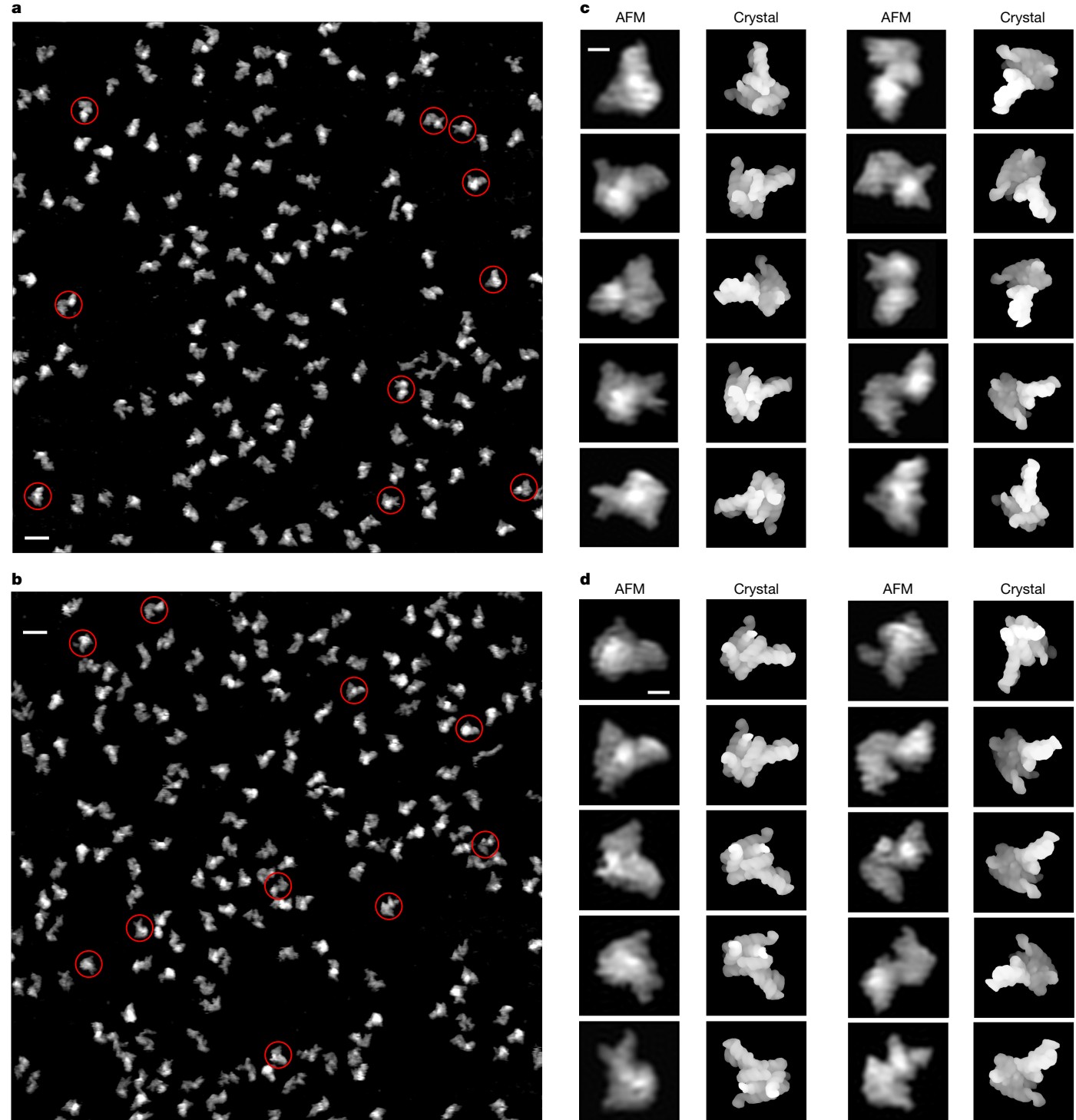

**Fig. 1 | AFM images of individual *Gst* RNase P RNA molecules in solution at 1 mM Mg²⁺. a,b**, Two representative AFM scanning areas. **c,d**, Representative particles of individual molecules highlighted in red circles in **a** and **b**, compared with the surface rendering of the crystal structure (PDB 2A64, missing residues added) in the matching orientations. Scale bars, 20 nm (**a**,**b**), 5 nm (**c**,**d**).

RNase P RNA molecules were obtained under a physiologically relevant Mg²⁺ concentration of 1 mM (refs. 39,40), revealing the broad conformational heterogeneity of this RNA (Fig. 1a,b and Extended Data Fig. 1a). These heterogeneous conformers of the folded molecules are distinctly different from unfolded molecules (Extended Data Fig. 1b). At the image resolution of 10–15 Å (ref. 41), the major structural features and overall folding are clearly discernible. The direct visualization and comparisons of the individual particles with surface renderings

of the crystal structure (PDB 2A64)[42] in matching orientations show conformational differences or similarities. This observation is not surprising, given that the RNase P RNA molecules in solution at 1 mM Mg²⁺ can freely sample a much wider conformational space than those packed in a crystal lattice at 70 mM Mg²⁺ (ref. 42). Notably, all reported structures of RNase P RNA were determined at Mg²⁺ concentrations of 10–100 mM (refs. 10,38,42), which is also true of most reported RNA structures in general, some of which had Mg²⁺ concentrations as high

as several hundreds of mM (ref. 43). In contrast to single snapshots of lowest energy conformations, the direct visualization by AFM of individual molecules under 1 mM $Mg^{2+}$ solution makes it possible to determine 3D topological structures of many distinct conformers of a given RNA.

## Structures of individual conformers

For the RNase P RNA, a total of 158 individual particles were selected from the AFM images on the basis of the quality of the particle images and being free from overlap or aggregation (Fig. 1 and Extended Data Fig. 1) and the 3D topological structures of each particle were then determined using HORNET[44], with an average estimated accuracy in terms of root mean square deviation (r.m.s.d.) of $4.1 \pm 0.4$ Å (Fig. 2a, Extended Data Fig. 2, Supplementary Figs. 1–3 and Supplementary Table 1). Here we describe the complete 3D structure of the full-length RNase P RNA of the *Bacillus* type (B type). These 158 conformers are indeed different from one another (Extended Data Figs. 1 and 3), with the population of each unknown. The smallest r.m.s.d. between two conformers in the ensemble of 158 structures is approximately 6.0 Å, which is greater than the uncertainty of the HORNET method. The coexistence of heterogeneous conformers cannot be attributed to RNA unfolding, which exhibits considerably higher heat exchange than conformational sampling (Fig. 2b). The full matrix of pairwise r.m.s.d. comparison among the conformers is shown in Extended Data Fig. 4a.

We approximate the ensemble behaviour by semiquantitative analysis of small-angle X-ray scattering (SAXS) data[23] (Methods and Extended Data Fig. 4b). Low-resolution SAXS data are useful in approximating ensemble behaviours even when not all explicit coordinates of conformers are available. On the basis of the combination of radius of gyration ($R_g$) values and overall SAXS profiles, the conformers could be divided into roughly three conformer classes: C1, C2, C3 (Fig. 2c–f). C1 consists of the compact group of conformers with $R_g$ of $46.5 \pm 2$ Å at one end; C3, with $R_g$ of $58 \pm 3$ Å, consists of extended conformers in which the S-domain is open more widely relative to the C-domain at another end of the conformation spectrum; and C2, with $R_g$ of $51 \pm 2$ Å, consists of the conformers in between the compact and extended ones. The open conformers have been implicated previously[10]. The compact C1 conformers are more akin to the crystal structure. Thus, we believe that the pool of the 158 conformers covers the nearly full, if not complete, spectrum of conformational space.

The three classes of conformers occupy different volume fractions, which are proportional to their individual populations and have characteristic radii of gyration (Fig. 2c–e). C2 and C3 classes account for more than 80% of the SAXS-derived fractionated volume (Fig. 2d). The back-calculated SAXS curve of the crystal structure does not fit the experimental data at 1 mM $Mg^{2+}$ ($\chi^2 = 21.0$, discussed below), nor does the SAXS curve from any one of the 158 conformers ($\chi^2$ range of 2.2–38.7; Fig. 2f). The more conformers used in the fitting, the better the fit in terms of mean, standard deviation and minimum $\chi^2$ (Fig. 2f). One notable difference among the conformers in C2 and C3 is the position of the stacked P8–P9 helices (S-domain) relative to the C-domain. The A-minor groove interaction involving the loop of P8 is observed in C2, but is absent in C3 (Fig. 2g). Using the ensemble optimization method[45] and the principal component analysis (PCA)[46] as two alternative fitting approaches with different sets of criteria also leads to three classes (Extended Data Fig. 4c,d).

## Conformational space of the RNase P RNA

The conformational trajectories, including the conversion of transient conformers, could be interpolated on the basis of PCA of individual conformer structures. The completeness of the derived conformational trajectories can be estimated on the basis of the population of variance and the total number of discrete conformers (Extended Data

Fig. 4) in the basis set. In our case, the population of variance is 70% and 80% with the top five and seven eigenvalue components, respectively (Extended Data Fig. 5a–c), with 158 discrete conformers in the basis set. For comparison, two previously reported protein studies used only 53 conformers of transducin[47] and 37 conformers of the kinesin motor domain[48] in the basis set.

Throughout the text we refer to motion as spatial motion without a temporal aspect. The root mean square fluctuation (r.m.s.f.) per residue among all conformers indicates that the S-domain, including P7–9, P10.1 and P12, exhibits the greatest spatial motions, with amplitudes as large as 55 Å (Fig. 3a, top panel). The r.m.s.f. profile of motion amplitude bears a resemblance to, and is thus corroborated by, the crystallographic *B* factor[38] (Fig. 3a, bottom panel), with a Pearson correlation coefficient of 0.8 (Fig. 3b). Moreover, the regions in the S-domain with markedly high structural fluctuation are the same regions in the crystal structure that could not be modelled due to missing electron density[38] (Fig. 3a, bottom, grey shaded areas). The r.m.s.f. plot derived from the ensemble of observed conformers provides explicit motion amplitudes for the entire RNA, including important regions that are too dynamic to be observed by high-resolution structure determination methods, or that may have been artificially restrained by crystal contacts. Therefore, this r.m.s.f. plot provides more complete and realistic spatial information about the motion of a free RNA molecule in solution and offers a new way to interrogate the structural dynamics of individual domains and structural elements in the context of the entire RNA.

PCA of the axes of maximal variance among the 158 discrete conformers can decompose motions into the minimum number of components while retaining the variance in terms of both amplitude and direction[46]. In principle, all PCA components cover the full motions of structural elements within the maximal variance. Our analysis indicates that the top five PCs cover approximately 70% of the total structural variance (Extended Data Fig. 5a), resulting in explicit motion directions and amplitudes, which are linearly proportional to the eigenvalues of each motion component (Fig. 3c,d). Consistent with the r.m.s.f. and *B* factor profiles, the largest motion amplitudes are observed among the peripheral structural elements, the majority of which reside in the S-domain (Fig. 3a, blue shaded region). The directional motions for each component are represented by an interpolation of the minimum to maximum spatial amplitude in the PCs (Fig. 3c and Supplementary Videos 1–5). Differing from the relatively less pronounced motions in protein enzymes, in which only a few residues or structural elements undergo a large scale of motions whereas the majority of the structure remains relatively invariant[47,48], all peripheral structural elements in RNase P RNA, including P1, P9, P15, P19, P10.1 and P12, undergo large movements of various amplitudes and directions (Fig. 3c). Notably, the secondary structural elements remain intact (Fig. 3e), consistent with chemical probing results[10,13], as they pivot around bulges, internal loops and junctions, as seen in bulge-containing small RNA fragments using NMR[12]. The P5.1–P15.1 kissing loop interaction that brings together two structural elements located at opposite ends of the RNA sequence also remains intact, moving in tandem to the order of 15 Å (Fig. 3a,c,e). P15, which connects P5.1 and P15.1, undergoes a displacement of about 20 Å (Fig. 3a,c). This motion is exemplified in PC5 (Fig. 3c). Importantly, P15 is directly involved in substrate recognition[49], in which the GGU sequence forms complementary base pairing with the 3′-CCA of the precursor tRNA or other RNA substrates[50]. The large-scale motion of P15 is indicative of flexibility and may explain why this ribozyme is capable of recognizing diverse substrates.

The analysis of the conformational space also reveals correlated motions, which could not be easily obtained using conventional biophysical methods. The atomic movement similarity matrix (AMSM) calculated among the 158 conformers illustrates the degree of motion correlations among specific regions of the molecule (Fig. 3f). The motions of the residues in P1-P2-P3 are highly correlated with those in P5, P15, P15.1, P15.2 and P4, whereas the motions of the residues in

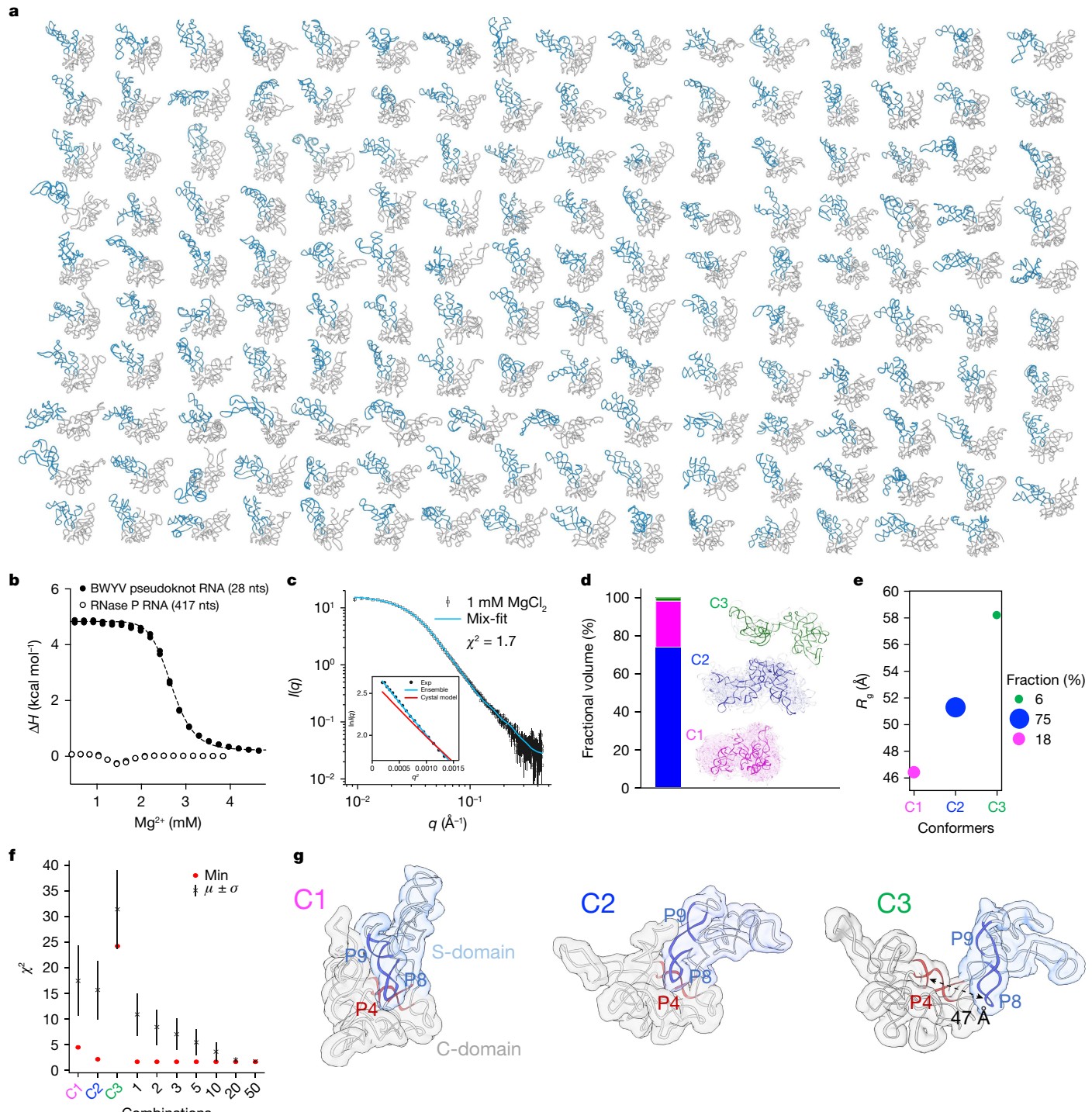

**Fig. 2 | Topological structures of the conformers. a**, Ribbon diagrams of all 158 conformers with the S- and C-domains coloured in blue and grey, respectively. **b**, Isotherm comparison between the $Mg^{2+}$-dependent compaction of RNase P RNA and refolding of the beet western yellow virus (BWYV) pseudoknot RNA. $N = 2$ for the duplicate measurements. nts, nucleotides. **c**, Experimental SAXS profile at 1 mM $Mg^{2+}$ (black) fit with the back-calculated SAXS curve (blue, $\chi^2 = 1.7$) using a linear combination of the three conformers representative of C1, C2 and C3, with volume fractions of 18%, 75% and 6%, respectively. **d**, Structures (transparent ribbons) of the 158 conformers classified into C1 (magenta), C2

(blue) and C3 (green) on the basis of their individual back-calculated scattering profiles relative to those of the 3 representative conformers (solid ribbons) with volume fractions. **e**, $R_g$ of the three representative conformers and class of the corresponding volume fractions. The diameter of each circle is linearly proportional to the population. **f**, $\chi^2$ for the fitting of conformers in the three classes (Methods). The mean values and standard deviation (s.d.) were obtained for 10,000 rounds of randomized selection of combinations (Methods). **g**, Representative structures of the three classes with key structural elements highlighted.

the S-domain (residues 151–251) show no correlation with those in P19 and P1–P2–P3. Structural elements P12 and P10.1 within the S-domain, which form a tetraloop (TL) to tetraloop receptor (TLR) interaction

(Fig. 3e), show highly correlated motions with one another, but little or no correlation with the C-domain (Fig. 3f). Furthermore, the motions of P4 and P8, which form an A-minor motif (Fig. 3e), are highly

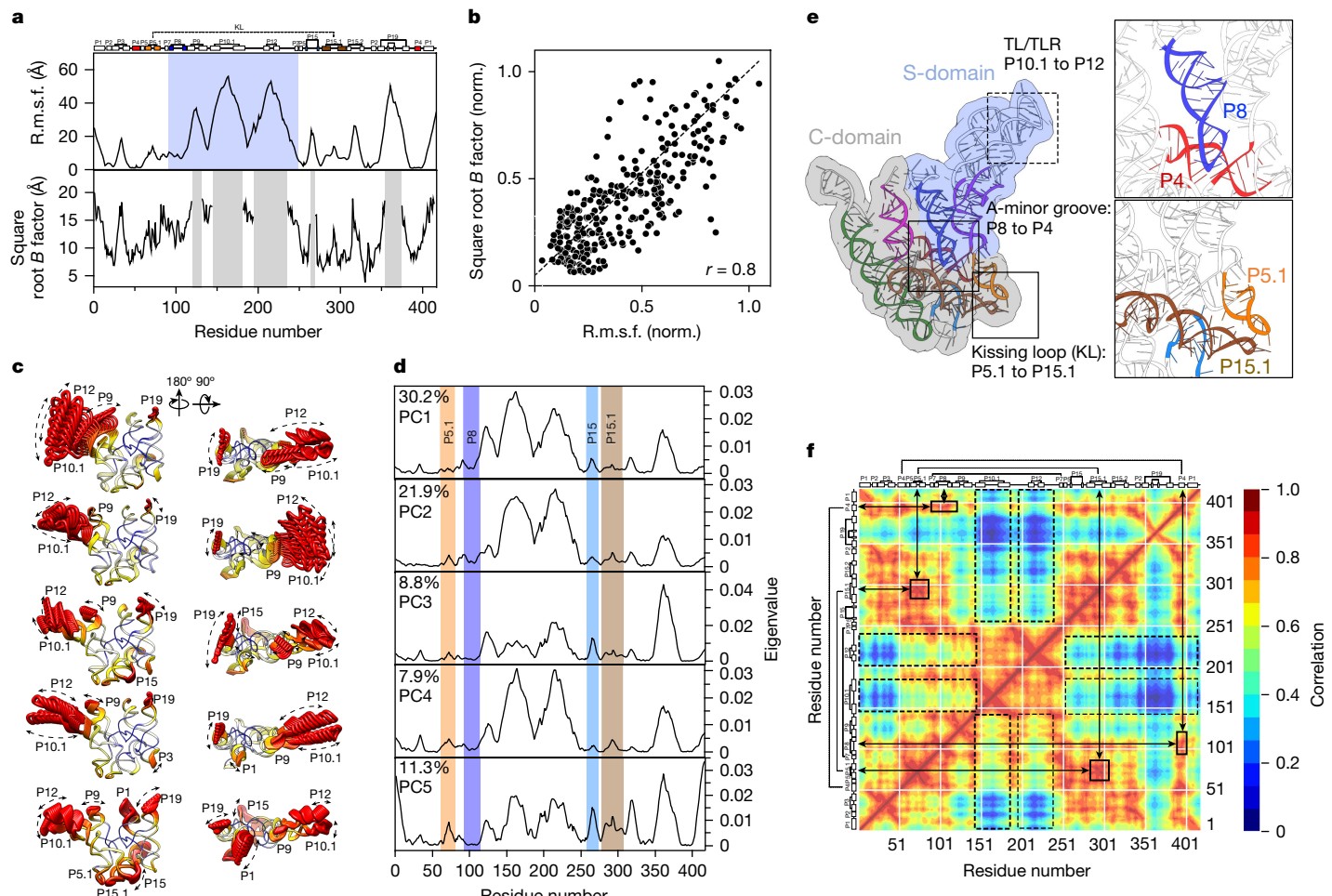

**Fig. 3 | Spatial and directional motions of structural elements. a**, Comparison of r.m.s.f. among all 158 conformers (top) and crystallographic *B* factor (PDB 2A64) (bottom) plotted versus residue number, with secondary structural elements indicated. The S-domain is shaded in blue (top panel). The grey shaded areas (bottom panel) indicate regions with missing electron density[38]. **b**, Correlation plot of the normalized (norm.) r.m.s.f. versus *B* factor. **c**, Directional motions derived from the top five PCs with structural variation coloured in descending order from red, to yellow, to blue, representing the extent of motion amplitude (Supplementary Videos 1–5). **d**, Eigenvalues per residue, which are linearly proportional to motion amplitude, for the top five PCs. The highlighted regions in coloured boxes indicate the structural elements involved in key tertiary interactions. **e**, The structural model showing the three key tertiary interactions (insets), coloured as in **d**. **f**, Pairwise AMSM, colour ramped from low (dark blue) to high (red) correlated spatial motions, with secondary structural elements indicated. Solid black boxes: correlated spatial motions between P5.1 and P15.1 (kissing loop interaction), between P8 and P4 (A-minor interaction). Dashed black boxes: non-correlated spatial motions between P10.1 TL and P12 TLR and all other structural elements.

correlated, and the same is true for P5.1 and P15.1, which form the kissing loop interaction. The AMSM therefore demonstrates that these key tertiary interactions are mostly maintained by the correlated large flexible motions of the peripheral domains. Importantly, conformers in PC2 (Fig. 3c) revealed a conformational transition between flat and concave states, showing that the S-domain can move towards the C-domain and form a state that may be more conducive to the binding of pre-tRNA substrate (Extended Data Fig. 5e and Supplementary Video 6). Such a concave conformation has been observed in structures of A-type RNase P RNA[10], but has never been observed for B-type RNase P RNA, whose reported structures in the absence of substrate show a flat conformation[10,42].

## Mg²⁺ reduces conformational space

$Mg^{2+}$ is critical for RNase P RNA function[31]. The effect of $Mg^{2+}$ on molecular compaction and reduced flexibility can be readily seen in the SAXS profiles (Fig. 4a), the dimensionless Kratky plots (Fig. 4b) and the pair distance distribution (Fig. 4c) at $Mg^{2+}$ concentrations from 0.1–10 mM. The orthogonal analysis[23] of the conformation distribution based on the SAXS data, in terms of the C1–C3 classes, shows that increasing levels of $Mg^{2+}$ lead to an increased population of the compact C1 and a decrease in the moderately extended C2 conformers (Extended Data Fig. 6 and Extended Data Table 1). Interestingly, the sharp conversion from extended to compact conformations occurs just above 1 mM $Mg^{2+}$ (Fig. 4d). By contrast, the minor population of the most extended C3 conformers remains relatively unchanged over the entire range of $Mg^{2+}$ concentrations (Fig. 4d and Extended Data Table 1). The compaction by $Mg^{2+}$ drives the RNA into a more crystal structure-like conformation, as evidenced by a lower $\chi^2$ between the back-calculated SAXS profile of the crystal structure and the experimentally measured SAXS data at higher $Mg^{2+}$ concentrations (Fig. 4e and Extended Data Fig. 7). At 10 mM $Mg^{2+}$, the $\chi^2$ decreases to 2.9, indicative of a much narrower conformational space, which is closer to the snapshot captured by crystallography. Of note, the crystal structure (PDB 2A64) was determined at 70 mM $Mg^{2+}$ (ref. 42). The $Mg^{2+}$-induced structural compaction is correlated with RNase P RNA enzyme activity, which was measured by hydrolysis of the human pre-tRNA$^{Gln}$ substrate (Fig. 4f, Extended Data Fig. 8 and Extended Data Table 1). Consistent with the decrease in $R_g$, the half-life of the cleavage

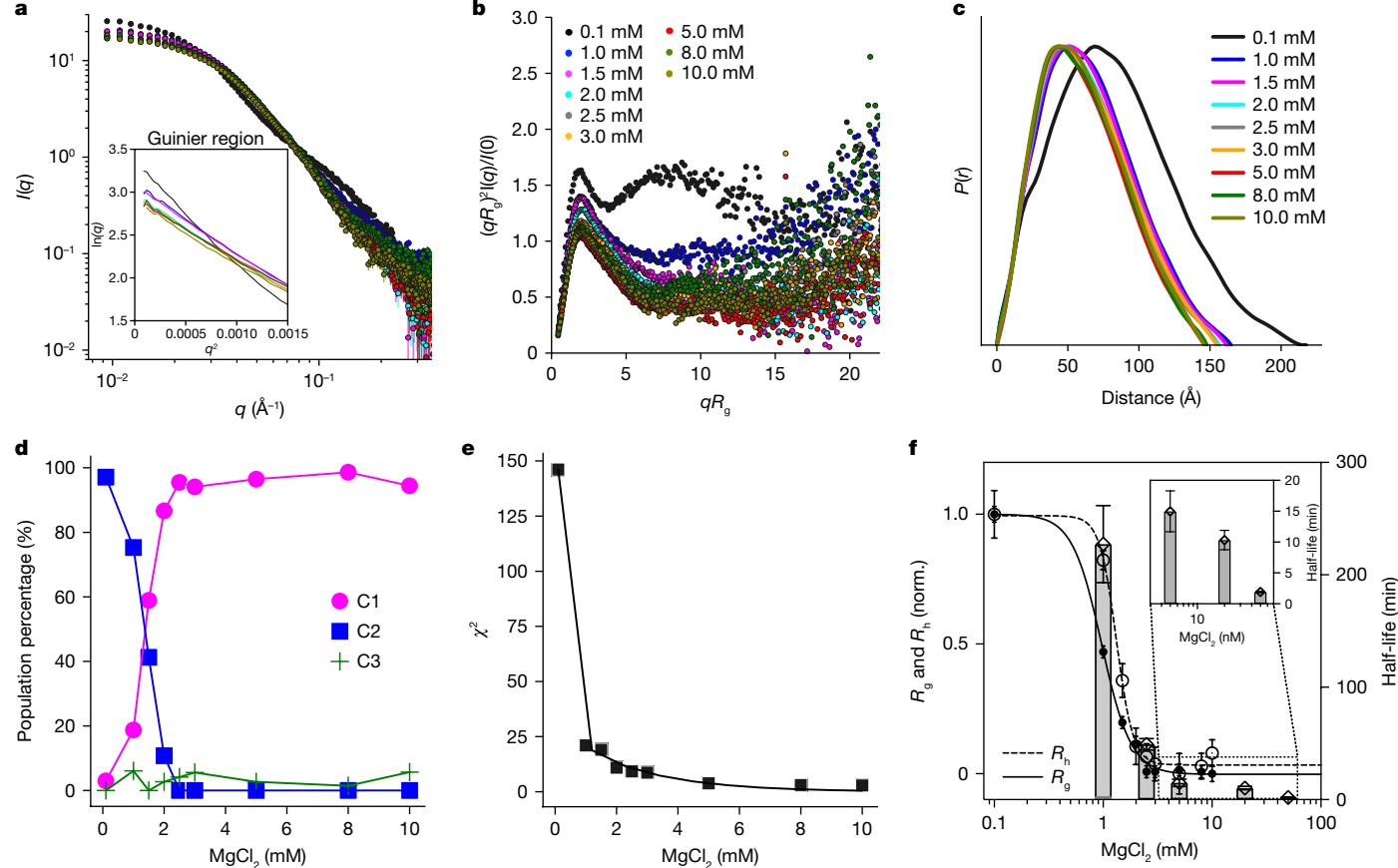

**Fig. 4 | Effect of Mg²⁺ concentration on conformation space and its correlation with enzymatic activity. a**, Comparison of the SAXS profiles and corresponding Guinier regions (inset) recorded at 0.1, 1.0, 2.0, 2.5, 3.0, 5.0, 8.0 and 10 mM Mg²⁺. **b**, Dimensionless Kratky plots. **c**, Pairwise distance distribution, $P(r)$, versus distance. **d**, SAXS-derived populations of the three main classes of conformers versus Mg²⁺ concentration. **e**, $\chi^2$ between the experimentally measured SAXS data and the back-calculated SAXS profile of the crystal structure (PDB 2A64) at each Mg²⁺ concentration (Extended Data Fig. 7).

**f**, Normalized hydrodynamic radius $R_h$ (open circles, dashed line), gyration radius $R_g$ (solid circles, solid lines) and reaction half-life derived from time-course experiments of hydrolysis of human pre-tRNA$^{Gln}$ (grey columns), plotted as a function of Mg²⁺ concentration. For clarity, reaction half-life at Mg²⁺ concentrations above 5 mM are magnified in the inset panel. See also Extended Data Table 1. Error bars are s.d. derived from nonlinear regression (Methods).

reaction is about 15 times shorter at 5 mM Mg²⁺ compared to that at 1 mM Mg²⁺ (Extended Data Table 1), indicating that the compact conformers apparently correlate with higher enzymatic activity. Notably, the Mg²⁺ concentration at which compactness and activity are most sensitive is once again just above 1 mM, the physiological cellular concentration of Mg²⁺ (refs. 39,40). Aside from its stabilizing effects, the requirement for Mg²⁺ concentrations greater than 1 mM for the highest activity may also be in part due to the low affinity of the catalytic Mg²⁺ ions[51].

## Sequence–flexibility correlation

Our method not only identifies the parts of the RNA that are structurally dynamic or have correlated motions, but also those that are rigid (Fig. 5). The analysis of the ellipsoid volume variance of the structure ensemble reveals 47 'anchor' residues whose relative positions are almost invariant among all 158 conformers (Fig. 5a, Extended Data Fig. 9 and Supplementary Table 2). These residues are mostly clustered within P4 and the junctions of P2–P19 and P2–P15.2 that comprise the highly conserved pseudoknot catalytic core (Fig. 5b,c). The coaxial stacking between P4 and P5 within the invariant core was previously reported to be stabilized by divalent cation interactions[38]. The most structurally invariant residues in P4 make direct contact with the elbow region of tRNAs[52–57]. Furthermore, residues 333, 339 and 387–388 in the invariant

core are at the binding interface with RnpA, the protein component of the bacterial RNase P holoenzyme[58,59] (Fig. 5b). It is understandable that, being a ribozyme, the RNase P RNA enzymatic activity requires a rigid core structure that precisely positions the substrate and catalytic residues. However, the observation that the RnpA binding interface is included in the structurally invariant core indicates that the rigidity of this region is also functionally important. Phylogenetic and structural comparisons of the RNase P RNA from different species reveal that the structure of the same invariant region is conserved regardless of whether the RNA is free or in complex with protein components (Extended Data Fig. 10).

The knowledge about RNase P RNA conformational space makes it feasible to examine the correlation between the sequence conservation score (SCS) and the 3D conformation conservation score (3DCS) derived from the per-residue r.m.s.f. (Fig. 5e and Methods). Our phylogenetic analysis using 114 type B RNase P RNA sequences shows that the overall SCS loosely correlates with 3DCS, with a Pearson correlation coefficient of 0.56 (Fig. 5f). The residues showing a relatively low SCS but high 3DCS (Fig. 5e, dashed line boxes) are involved in the A-minor (P8–P4) and kissing loop (P5.1–P15.1) interactions, and are thus probably driven by structural complementarity even though they are relatively less sequence conserved. On the other hand, the residues showing high SCS but low 3DCS (Fig. 5e, solid line boxes) may help to maintain the dynamical motions of the S-domain that sufficiently

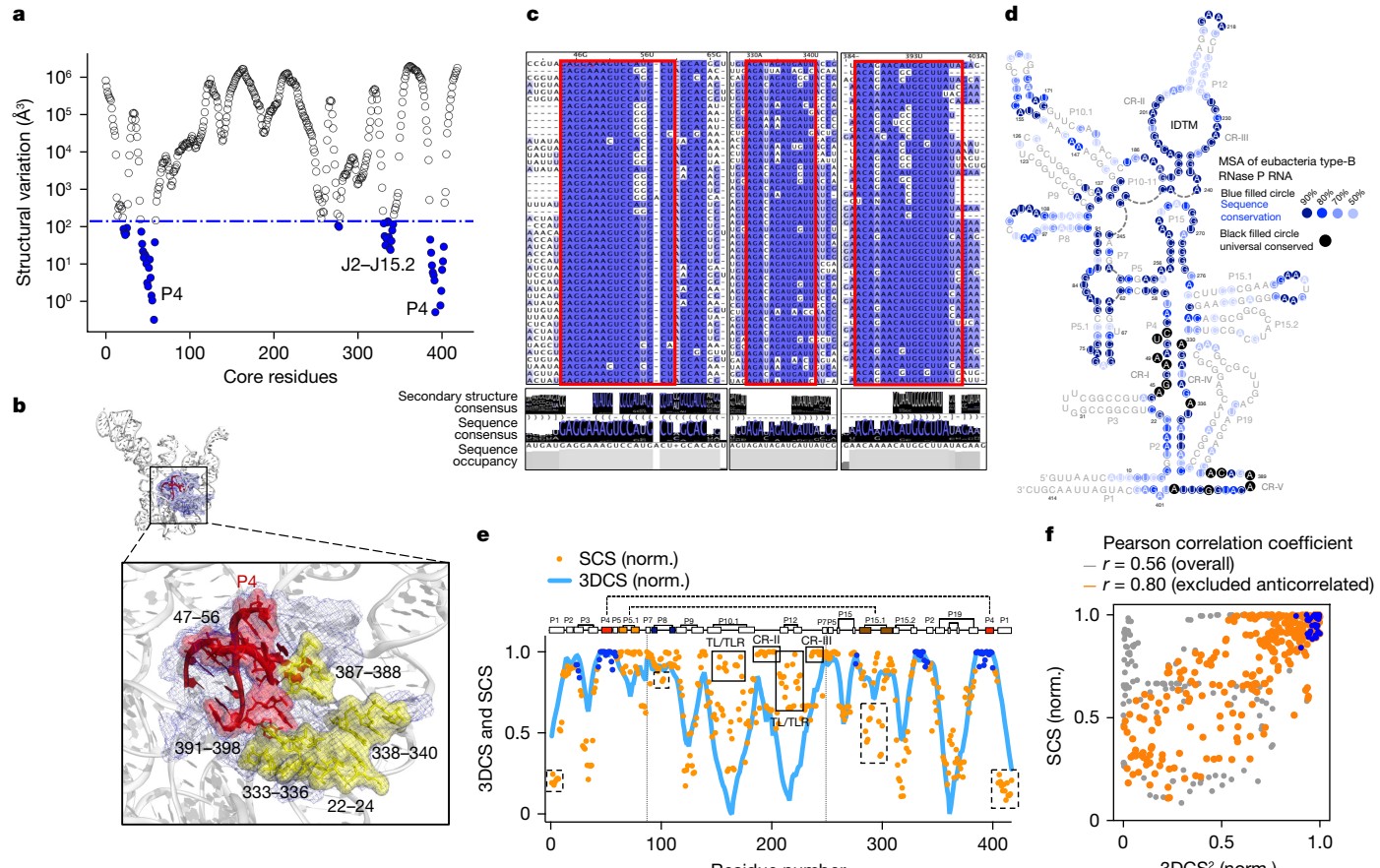

**Fig. 5 | Structurally invariant residues and correlation between sequence conservation and conformational dynamics. a**, Analysis of ellipsoid volumes for determining structurally invariant residues (blue) using Bio3d[46]. **b**, The RNase P RNA shown with structurally invariant residues (blue mesh) and residues in the tRNA- and protein-binding interfaces coloured in red and yellow, respectively. **c**, Phylogenetic analysis using multiple sequence alignment of 114 bacterial type B RNase P RNAs from Rfam[61] database (Methods). For clarity, only 46 sequences are shown. Sequentially invariant residues highlighted in red rectangular boxes coincide with structurally conserved residues shown in **a** and **b**. **d**, Sequence conservation mapped onto the secondary structure of *Gst* RNase P RNA[42]. **e**, SCS (orange) and 3DCS (blue). Solid black boxes: regions with high SCS but low 3DCS. Dashed black boxes: regions with low SCS but high 3DCS. **f**, Correlation plot between SCS and 3DCS with a scaling exponent of 2. Structurally invariant residues in **e** and **f** are displayed as blue dots.

sample conformations for a broad spectrum of substrates. These sequence-conserved but conformationally mobile residues are anti-correlated and located in P10.1 and P12, which form the TL–TLR interaction (Fig. 3e), and conserved regions II and III (CR-II and CR-III) of the S-domain (Fig. 5d), which form the interdigitated double T-loop motif and make direct contact with the elbow region of the tRNA substrate[58]. This suggests that the dynamic nature of these sequence-conserved regions may be functionally relevant for RNase P to recognize diverse RNA substrates, and is encoded in the primary RNA sequence. Excluding these anticorrelated residues, the Pearson correlation between SCS and 3DCS rose to 0.80 (Fig. 5f).

## Discussion

Our results demonstrate the conformational dynamics of this RNA that is consistent with, but far beyond, the known information obtained using indirect methods[10,13,29,60]. Such large motion amplitudes of conformational sampling in a multitude of directions at little energy cost explain the adaptability required to accommodate various types of substrates at the expense of catalytic efficiency. This conformational sampling is in contrast to the transition between thermodynamically distinct states. Furthermore, it has been revealed that the structurally invariant core that includes the tRNA- and protein-binding interfaces (Fig. 5b) is evolutionarily conserved in terms of both the primary sequence and known structures across all kingdoms of life (Extended Data Fig. 10), and is therefore of functional importance. Mutations in the pseudoknot core of human mitochondrial RNA processing RNase (RNase MRP) are linked to several human genetic diseases. The combination of highly flexible peripheral domains and an invariable core explains the paradoxical behaviour of substrate promiscuity and enzymatic precision of this ribozyme. Finally, the phylogenetic comparison of sequence conservation among all kingdoms of life and conformational conservation among all *Gst* RNase P RNA conformers provides an opportunity to examine possible links between sequence evolution and structural dynamics of this RNA, suggesting a strong correlation between its primary sequence conservation and 3D conformational conservation in the invariant core and the secondary structural regions, excluding the S-domain (Fig. 5). Interestingly, anticorrelation of some highly sequence-conserved regions in *Gst* RNase P RNA exhibited high mobility (Fig. 5e), implying an evolutionarily conserved conformational flexibility. Altogether, these correlations and anticorrelations in the RNase P RNA suggest that the primary sequence encodes information in the form of structure and conformational dynamics propensity, both of which have important functional roles. Furthermore, our results may stimulate discussion about the deep-rooted belief of 'one sequence one structure', which originated from the protein folding theory that still influences current thinking about RNA structure.

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

## Methods

### RNase P RNA sample preparation

RNase P RNA was transcribed in vitro in transcription buffer (20 mM potassium–HEPES buffer pH 7.5, 25 mM MgCl$_2$, 1 mM DTT) for 3 h with recombinant T7 bacteriophage RNA polymerase and double-stranded DNA template amplified by PCR from a synthesized plasmid pUC18. This plasmid encodes the full-length RNase P RNA sequence from bacterial strain *G. stearothermophilus* (GenBank access number M19021.1) with an upstream T7 RNA polymerase promoter sequence, GGATCCAGCTCG AAATTAATACGACTCACTATA. After in vitro transcription (IVT), the magnesium pyrophosphate precipitate was removed by centrifugation with a spin rate of 13,000 rpm for 10 min using a high-speed benchtop centrifuge. RNase-free DNase I (New England BioLabs) and the final concentration 5 mM CaCl$_2$ were added into the IVT supernatant, and the solution was incubated at 37 °C to fully digest the double-stranded DNA template for an additional 30 min. The final concentration of 200 mM sodium chloride (NaCl) was then added to the IVT solution for overnight refolding at 4 °C before further purification. The refolded RNA was subjected to fast protein liquid chromatography (GE Health-Care ÄKTA pure) and purified by a suggested non-denatured method[10] using a size-exclusion chromatography (SEC) column (HiLoad 16/600 Superdex 200 pg). The column was pre-equilibrated with the SEC elution buffer (25 mM Tris pH 7.5, 100 mM NaCl, 1 mM MgCl$_2$) before purification, and the monomeric RNase P RNA molecules were eluted with a flow rate of 1.0 ml min$^{-1}$ and separated from aggregation species. Eluting RNase P RNA was detected by absorbance at 280 and 260 nm; peak fractions were collected based on the SEC chromatogram (Supplementary Fig. 4a) and stored at 4 °C for a period of a few minutes (typically, less than 30 min) before the AFM visualization. A small aliquot of the RNA was taken from the elution fractions for checking the purity and folding using 8% native PAGE and electrospray ionization (ESI) mass spectrometry (Supplementary Fig. 4b,c).

### AFM experiment and image processing

All AFM experiments were performed in a physiologically relevant buffer solution using a Cypher VRS AFM (Asylum Research, Oxford Instruments) at 4 °C with amplitude-modulated dynamic a.c. mode, known as the tapping mode. To conduct RNA adhesion on an AFM mica surface, the mica supports were freshly treated with 1-(3-aminopropyl) silatrane (APS) (synthesized in-house). A 50 mM APS stock solution was diluted by 300-fold in ultrapure water just before use and coated on freshly cleaved muscovite mica (Highest Grade V1 mica discs, Ted Pella). After 30 min, the mica surface was rinsed with ultrapure water (Pico Pure Water system, Avidity) several times, and dried gently with filtered nitrogen gas. Then, 10 μl 8 nM RNase P RNA in the purified buffer (25 mM Tris pH 7.5, 100 mM NaCl, 1 mM Mg$^{2+}$) was deposited on the APS-functionalized mica surface for 20 min and washed with 500 μl AFM buffer (10 mM MES pH 6.8, 10 mM KCl and 1 mM MgCl$_2$) at least three times. FASTSCAN-D-SS AFM probes (Bruker, resonance frequency of 110 kHz in fluid and spring constant of 0.25 N m$^{-1}$) with tip apex radius of 1 nm were used for high-resolution imaging. A pulsed blue laser (BlueDrive) equipped with the AFM instrument was used for photothermal excitation in tapping mode, positioned at the rear of the cantilever, while a super-luminescent photodiode was positioned near the head of the cantilever to detect cantilever deflection. AFM images for particle cropping were collected with a scan size of 500 × 500 nm$^2$, 1,024 × 1,024 pixels$^2$ and a scan rate of 1.0 Hz. The AFM tip was carefully landed on the RNA molecular surface on a piezoelectric scanner driven by an initial setpoint voltage of 450 mV and a free amplitude of 500 mV. The setpoint voltage was reduced in a stepwise manner (10 mV per step) while the tip was approaching the surface and adjusted based on the image quality during imaging. For images used for 3D topological structure determination, the raw images were processed with Gwyddion[62] using the following steps: (1) plane flatness correction by applying second-order polynomial levelling to the particle-free region; (2) filtering by correct horizontal scans to remove string artefacts; and (3) fast Fourier transform analysis to remove high-frequency noise in Fourier transform space. Single-particle images were cropped from the processed images and converted into a text file with *X*, *Y*, *Z* topography information for structure recapitulation.

### Enzymatic assay of 5′ processing of precursor tRNA

Human pre-tRNA$^{Gln}$ was produced by in vitro transcription using T7 bacteriophage RNA polymerase. As DNA templates, to avoid the non-specific *N* + 1 activity of the T7 RNA polymerase, PCR products that bear two consecutive 2′-*O*-methyl modifications on the 5′ end of the template strand were used. The pre-tRNA$^{Gln}$ was purified using 8 M urea denaturing PAGE and eluted in 300 mM sodium acetate buffer (pH 5.3). The eluted pre-tRNA$^{Gln}$ was further refolded at 10 μM using a stepwise and temperature-ramped protocol (90 °C for 3 min to fully denature pre-tRNA then quickly ramped down on a PCR machine to 4 °C at a maximum rate of 5 °C s$^{-1}$). After the pre-tRNA$^{Gln}$ refolding, 2 μM of each in vitro-transcribed pre-tRNA was incubated with 2 μM purified RNase P complex at 37 °C in the presence of 25 mM Tris–HCl pH 7.5, 100 mM NaCl and with various MgCl$_2$ concentrations, ranging from 1 mM to 50 mM (Fig. 4f and Extended Data Fig. 8). Reactions were terminated by adding the denaturing gel loading buffer and incubated at 4 °C. Samples were analysed on an 8 M urea Tris–borate–EDTA denaturing preparative 10% polyacrylamide (29:1 acrylamide:bisacrylamide) gel and stained with SYBR Gold (Invitrogen by Thermo Fisher Scientific). The 5′ processing of pre-tRNA$^{Gln}$ via RNase P RNA produces the 76-nucleotide product, which can be separated from the remaining pre-tRNA$^{Gln}$ by denaturing PAGE (Extended Data Fig. 8b,c and Supplementary Fig. 5). Quantification was performed using ImageJ software and the data were analysed using GraphPad Prism 10. All the experiments were performed at least in duplicate.

### Dynamic light scattering

Before dynamic light scattering (DLS) measurements, the RNA samples were centrifuged (5 min, 10,000 rpm, 11,000*g*) in a refrigerated Sigma benchtop centrifuge. DLS experiments were performed at the Biophysics Resource facility, National Cancer Institute, using the DynaPro Plate Reader III Dynamic Light Scattering instrument (Wyatt Technologies) composed of a laser light source (830 nm laser diode), a plate reader cell, a detector placed at a fixed angle of 90°, a photomultiplier amplifying the signal and a correlator. A 20 μl portion of sample solution was loaded into the corresponding sample well of the 384-well microwell plate. The sample plate, placed pairwise on cushion plate holders of the four-place swinging-bucket rotor of a benchtop centrifuge, was centrifuged for 5 min at 1,500*g* to eliminate air bubbles in the sample well. Before the plate was placed into the instrument, its bottom surfaces were wiped gently with a sheet of soft lens-cleaning tissue (Olympus Optical). At least 20 successive DLS measurements were performed per sample after 1 min waiting time to enable the solutions to be at equilibrium. The translational diffusion coefficient ($D_T$), molecular weight ($M_w$) and hydrodynamic radius ($R_h$) were calculated from averaging of a series of autocorrelation profiles, at a fixed RNA concentration of 0.20 mg ml$^{-1}$.

### LC–ESI MASS experiments

Liquid chromatography–mass spectrometry (LC–MS) experiments were performed on a 6520 Accurate-Mass Q-TOF LC/MS system equipped with a dual electrospray source, operated in positive-ion mode. Samples included 2 μM RNase P RNA in RNase-free double-distilled water solution. Acetonitrile was added to all samples to a final concentration of 10%. Data acquisition and analysis were performed using a Mass Hunter Workstation (v.B.06.01). For data analysis and deconvolution of mass spectra, Mass Hunter Qualitative Analysis software (v.B.07.00) with Bioconfirm Workflow was used. The supernatant was transferred to polypropylene injection vials for LC–MS analysis. LC–MS was performed

with a TSQ Quantiva triple quadrupole mass spectrometer (Thermo Fisher Scientific) operating in selected reaction monitoring mode with positive ESI and with a Shimadzu 20AC-XR system using a $2.1 \times 50$ mm$^2$, 2.7 μm Waters Cortecs C18 column.

## Recapitulation and accuracy of topological structure

The AFM image of full-length RNase P RNA shows a broad number of distinct and heterogeneous particle shapes in terms of molecular surface. After image processing, as described above, we selected and evenly cropped isolated particles, that is, the particle does not overlap with any neighbour molecule, resulting in a total of 161 individual topological conformers. The individual 3D structures for the 161 AFM images were calculated and the structure quality and degree of accuracy were estimated using the HORNET software package: https://github.com/PNAI-CSB-NCI-NIH/HORNET (ref. 44). Three of these 161 particles were not converged within a reasonable computing time. These particles are apparently elongated, which could possibly be explained by partial unfolding, and thus were excluded from further analysis. Briefly, the native information and initial 3D folding were built using a combination of the crystal model (PDB 2A64), SimRNA[63] and Coot[64]; the last two packages were respectively applied to modelling the missing residues that were not resolved in the crystal model, followed by the structure refinement procedure. Then, the initial model was aligned against the AFM image, in which the optimized rotation and translation of the model give a maximum score of agreement between the experimental AFM image and the calculated image of the respective model orientation. Next, the optimized initial model orientation was saved, and a configuration file created for the dynamic fitting step applying CafeMol[65,66], for a trajectory with a total of 20 million frames (approximately 0.1 μs). Of note, the native structure information of local contact, stacking and base-pairing energies was scaled by a weighting factor of 5, 9 and 9, respectively. Finally, the trajectory and structure accuracy were analysed and evaluated by HORNET, in which the top models were selected. The same procedure was performed for all 158 particles (Supplementary Figs. 1–3).

## SAXS experimental data acquisition and ensemble analysis

The SAXS experiments were carried out using the in-house instrument at the NCI SAXS core facility (BioSAXS-2000, Rigaku) located at the National Cancer Institute. The photon energy used was 8.04 keV ($\lambda = 1.54$ Å). The combination of OptiSAXS optic, two-dimensional Kratky collimation and sample-to-detector distance of 0.484 metres enabled us to obtain a $q$ range of $0.0051 < q < 0.6767$ Å$^{-1}$, where $q$ is the magnitude of the momentum transfer, $q = (4\pi/\lambda)\sin\theta$, $2\theta$ is the scattering angle and $\lambda$ is the wavelength of the radiation. To minimize radiation damage and obtain a good signal-to-noise ratio, 8 image frames were captured for each sample at 0.55 mg ml$^{-1}$ (4 μM), in the same buffer conditions used for AFM imaging at various Mg$^{2+}$ concentrations, using a flow cell with an exposure time of 900 s per frame. The two-dimensional scattering patterns were collected using a Dectris PILATUS 100K detector and then converted to one-dimensional SAXS curves through radial averaging. The one-dimensional data from the 8 frames were subsequently averaged after per-curve evaluation of outliers using the software SAXSLab v.4.0.2 (Rigaku). To examine the effect of sample concentration on the scattering profiles, we repeated the SAXS experiments with samples at lower and higher concentrations: 0.38 and 0.76 mg ml$^{-1}$. The similarity between the SAXS intensity profiles obtained from the two datasets was assessed using CorMap[67] and the reduced $\chi^2$ values, neither of which showed any indication of concentration-dependent effects with statistical significance (Supplementary Fig. 6).

To orthogonally classify the conformers determined from HORNET and AFM (Fig. 2a), we applied SAXS experiments, which are non-correlated to AFM-derived results.

The SAXS data and standard analysis for all recorded scattering experiments at different Mg$^{2+}$ concentrations are available in the SAXSBDB public repository[68]; the accession codes can be found in the data availability section of this article. Structural models derived from HORNET and AFM experimental data collected at 1 mM Mg$^{2+}$ (Fig. 2a) were used to fit the SAXS experimental profile collected at each Mg$^{2+}$ concentration. SAXS and AFM present distinct yet complementary experimental approaches for structural investigation. SAXS provides averaged information over the entire ensemble of molecules, in a concentration range from μM to mM, in which the scattering signal is collated over a timescale of minutes. AFM, on the other hand, is a direct visualization of individual particles, at low nM sample concentrations, where each particle represents a snapshot of a single conformer of the ensemble at the time of immobilization. However, the structure calculations of thousands of particles observed by AFM is impractical in terms of both labour and computational resources. A complementary method, such as SAXS, therefore, is more suitable for characterizing the solution ensemble of models derived from AFM data.

As the population fraction of each conformer is unknown, including interconversion of species on various timescales, we assume that the experimental SAXS profile can be described by an ensemble of models where the total SAXS intensity ($I_{Total}$) is a linear combination of $n$ conformers. The contribution of each conformer is determined by the minimization of the discrepancy between the calculated profile and experimentally recorded data, that is:

$$I_{Total} = \sum_{i=1}^{n} v_i I_i(q) \qquad (1)$$

where $v_i$ represents the volume fraction of particle $i$ with a scattering intensity profile equal to $I_i(q)$. The 158 AFM-derived conformers were used as a pool of reference structures for SAXS profile fitting, and the synthesized profile of $I(q)$ for each particle was determined using CRYSOL[69]. The optimized volume fraction, $v$, for each component of the ensemble is obtained by minimizing the discrepancy between the back-calculated $I_{Total}$ and $I_{experimental}$ ($\chi^2$) curves using an in-house Python script that implements an iterative least-squares process[70] by applying a trust region reflective algorithm[71], with boundaries of $0 \le v_i \ge 1$.

The best fit to the experimental data ($\chi^2 = 1.7$) was obtained with an ensemble of 3 of the 158 conformers, with volume fraction percentages of 18% (S31), 76% (S69) and 6% (S53) (Fig. 2b,c). However, different combinations of different models could achieve a similar fit to the SAXS profile, with $\chi^2$ ranging from 1.8 to 15 (Fig. 2f). This finding is expected, given that the more populated conformers throughout the course of the data collection contribute more to the total scattered intensity, and our calculated particles from AFM images represent only a sampling of the billions of particles immobilized on the mica surface.

The three conformers that best fit the SAXS data exhibited different levels of compactness, from very compact (S31), to partially open (S69), to fully extended (S53). Based on this observation, we then classified the 155 models into 3 clusters (classes C1, C2 and C3) using the 3 representative structures as reference. For this task we partitioned the theoretical SAXS profile, respectively, for each of the 155 models using the cosine distance[72] among all variations of intensity as a function of $q$ ($I(q)$). A minimum threshold of 0.1 was set per similarity cluster. The largest group was C2 with 117 models (S69 as reference), which was also the largest volume fraction observed by SAXS, followed by C1 with 31 models (S31 as reference) and C3 with 7 models (S53 as reference).

Indeed, the classified particles based on the SAXS profile similarity to representative structures show similar topological features, as shown in Fig. 2d, overlaid with the reference structure for each class. This analysis hinges on global conformational similarity by ignoring local conformational fluctuation (Fig. 2d). The three topological classes of conformers are defined primarily by the relative orientations of the two modular domains of RNase P RNA, namely the substrate specificity (S-) and catalytic (C-) domains, which are linked by a flexible linker (Fig. 2g), giving $R_g$ values ranging from 46 to 58 Å (Fig. 2e).

To further investigate the variation of $\chi^2$ during ensemble fitting, we performed an optimization of the volume fraction settings using different combinations of the 158 particles. In this procedure, we start with the fitting of each of 158 conformers, independently, mapping the $\chi^2$ values, and then correlating them with the class (C1, C2 and C3) to which that particle belongs. Then, we add in permutations of a given number (1, 2, 3, 5, 10, 20, 50) of random conformers from each class, map the $\chi^2$ for 10,000 rounds, in which (1) assumes one structure from each class, (2) assumes two structures from each class, (3) assumes three structures from each class, and so on.

In terms of $\chi^2$ fluctuation, we observed that C2-like particles have better agreement with the experimental SAXS profile, as those particles indeed represent an intermediate topology between closed and open conformations with respect to the S- and C-domains. The combination of C2 and C1 can reach a $\chi^2$ as small as 2.1, but the combination of C1, C2 and C3 gives the best $\chi^2$ and the lowest standard deviation (Fig. 2f). Combinations including a greater number of conformers yield a better fit in terms of the $\chi^2$ mean ($\mu$), standard deviation ($\sigma$) and minimum value, but do not improve beyond 20 from each class ($\mu = 2.0$, $\sigma = 0.5$). C3 conformers show the largest deviation from the experimental SAXS data ($\mu = 31.4$), the largest $\chi^2$ range ($\sigma = 7.7$) and the largest minimum $\chi^2$ (24.2).

Given that every method has its limitations, we applied two additional independent methods to classify the 158 AFM-derived conformers: clustering in PC space[73] and the ensemble optimization method (EOM)[45,74]. The analysis of clustering in PC space makes use of orthogonal eigenvectors to describe the maximal variance of the space distribution among the 158 models. We observed that seven components were sufficient to cover more than 70% of the variance. Making use of these components, we clustered the 158 models into three main clusters (Extended Data Fig. 4c). The PC analysis and clustering were performed using the Bio3d package. SAXS data were analysed using the EOM package, applying the genetic algorithm (GAJOE) module, which uses a searching process to select a subensemble of models from a pool that is sufficient to describe the SAXS data. For this analysis, we used a maximum number of conformers per ensemble of 50, a number of ensembles per generation of 50 and a minimum number of models per ensemble of 1, with no curve repetition. The EOM performed the fitting for 100 cycles of repeated searching. The best fits achieved using EOM had $\chi^2$ values of 2.3 and 1.1, respectively, for SAXS data recorded at 1 mM and 5 mM $Mg^{2+}$. As the results in Extended Data Fig. 4d show, RNase P RNA presents three main distributions of $R_g$ with high frequency, the largest of which is around 51 Å, the next largest between 47.5 and 50 Å and the smallest population showing $R_g > 55$ Å. At 5 mM $Mg^{2+}$, the largest population shifts to smaller $R_g$ values, $R_g < 47.5$ Å, and no significant counts are observed for the extended conformers with $R_g > 52$ Å.

## Isothermal titration calorimetry

RNase P RNA (4 μM) was dialysed overnight at 4 °C against the isothermal titration calorimetry (ITC) buffer (20 mM HEPES pH 7.5, 100 mM NaCl, 0.1 mM $MgCl_2$) before ITC measurements. The dialysis buffer was used to dissolve $MgCl_2$ hexahydrate (Sigma-Aldrich) to a final concentration of 20 mM $MgCl_2$, which was used as titrant. Differential heat of the $Mg^{2+}$-induced compaction of RNase P RNA was monitored using a MicroCal PEAQ-ITC instrument (Malvern). After pre-equilibration at 37 °C and an initial delay of 180 s, 0.4 μl of titrant was injected, followed by 18 serial injections (2.0 μl each) with spacing of 720 s. Stirring speed was 750 rpm and the reference power was set at 8 μcal s$^{-1}$. Thermogram data were recorded as power (μcal s$^{-1}$) over time. Afterwards, the heat associated with each titration step was integrated and plotted against the molar ratio of $Mg^{2+}$ and the RNA. Each binding isotherm was calibrated for dilution effects by a corrected RNA concentration quantified using a NanoDrop after the ITC experiment.

The beet western yellow virus (BWYV) pseudoknot RNA was transcribed by IVT, followed by SEC purification (Supplementary Fig. 7) using a HiLoad Superdex 75 pg 16/60 prepacked column (Cytiva) with elution buffer (20 mM HEPES pH 7.5, 100 mM NaCl). The purified BWYV pseudoknot RNA was denatured by supplementing 200 mM EDTA into the buffer and heating at 80 °C for 10 min. The 2 μM denatured BWYV pseudoknot RNA was then dialysed overnight at 4 °C against buffer containing 20 mM HEPES pH 7.5. Titrant was prepared by dissolving $MgCl_2$ hexahydrate (Sigma-Aldrich) in dialysis buffer to a final concentration of 60 mM. Differential heat of the $Mg^{2+}$-induced refolding of BWYV pseudoknot RNA was monitored using the MicroCal PEAQ-ITC (Malvern). After pre-equilibration at 25 °C and an initial delay of 90 s, 0.4 μl of titrant was injected, followed by 22 serial injections (1.0 μl each) with spacing of 100 s. Stirring speed was 750 rpm and the reference power was set at 4 μcal s$^{-1}$.

For processing of ITC data, the raw thermogram (Supplementary Fig. 8) of each heat compensation profile was used to derive the isotherms (Fig. 2b). Data integration and background heat subtraction were done using the PEAQ-ITC analysis software suite (Malvern).

## Invariant core and directional motions

To assess the structural relationship among all 158 resolved conformers of RNase P RNA we applied the Bio3d package[46] with implemented function for determination of invariant core residues and PCA approach. The invariant core analysis addresses the rigid and 3D invariable region of a pool of models. In this procedure the atom displacements are quantified by an interactive alignment of all structural coordinates, where each round of superposition determines an ellipsoid volume that covers the variance of $X$, $Y$ and $Z$ coordinates among the structures, and residues with largest fluctuation are removed from the next superposition interaction[46,75]. After a sequence of rounds the remaining residues over a small ellipsoid volume define the core region. To define a cutoff of the ellipsoid volume that represents an invariable core we calculate the derivative of the ellipsoid volume as a function of the number of residues present in the remaining refined structure, and the minimum valley is reached with an ellipsoid volume of 150 Å$^3$ (Extended Data Fig. 9). Following the defined core residues, we disseminate the most important directional motions for all resolved 158 conformers using PCA. PCA is a method that is well suited to combine and disseminate similarities and differences of conformational space among different conformers of the same molecule. Briefly, the orthogonal eigenvectors, named as principal components (PC), describe the most variance of the structural data by reducing the dimensionality of features but maintaining the information[46] and mathematical description in HORNET[44]. The PCA was performed using the core as a reference, to first apply structure superposition of the structure models and, afterwards, the principal components were obtained for the pool of 158 structures. The number of components was determined using the variance and number of components plot (Extended Data Fig. 5a). The five and seven components cover more than 70% and 80% of total fluctuations, respectively.

## Correlation of sequence conservation and structural fluctuations

The SCS were calculated by taking into consideration sequence homology, secondary structure conservation and compensatory changes via long-distance tertiary interaction. Multiple sequence alignment, secondary structure consensus and sequence occupancy of 114 bacterial type B RNase P sequences were explicitly defined from the Rfam database[61] with accession number RF00011 (Data availability), the sequence aliment and visualization were performed using Jalview cross-platform[76]. Primary sequence conservation derived from multiple sequence alignment was further transformed and standardized in a probabilistic framework using the ConSurf server[77]. The 3D structural conservation scores are calculated on the basis of '1 minus fractional per-residue r.m.s.f.' of all 158 conformers that were normalized between 0 and 1 (Fig. 3a).

## Reporting summary

Further information on research design is available in the Nature Portfolio Reporting Summary linked to this article.

## Data availability

Data are available in the main project folder at https://home.ccr.cancer.gov/csb/pnai/data/conformational_space/Conf_space_RNasePRNA/. PCA and core analysis are available at https://home.ccr.cancer.gov/csb/pnai/data/conformational_space/Conf_space_RNasePRNA/scripts_analysis/PCA_core/. AFM data are available at https://home.ccr.cancer.gov/csb/pnai/data/conformational_space/Conf_space_RNasePRNA/AFM_images/. DNN data are available at https://home.ccr.cancer.gov/csb/pnai/data/conformational_space/Conf_space_RNasePRNA/DNN/. Sequencing analyses are available at https://home.ccr.cancer.gov/csb/pnai/data/conformational_space/Conf_space_RNasePRNA/multiple_sequence_alignment/. SAXS data are available at https://home.ccr.cancer.gov/csb/pnai/data/conformational_space/Conf_space_RNasePRNA/SAXS_data/ and the analysis is available at https://home.ccr.cancer.gov/csb/pnai/data/conformational_space/Conf_space_RNasePRNA/scripts_analysis/SAXS_ensemble/. All SAXS data are available at SAS data bank https://www.sasbdb.org/project/2201/b9y8c4b6qf with the SASBDB accession codes: SASDTA7, SASDTB7, SASDTC7, SASDTD7, SASDTE7, SASDTF7, SASDTG7, SASDTH7, SASDTJ7.

## Code availability

All codes for SAXS analyses, PCA and the structures respectively used and generated from this study are available at https://home.ccr.cancer.gov/csb/pnai/data/conformational_space/Conf_space_RNasePRNA/scripts_analysis/.

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

**Acknowledgements** We thank W. F. Heinz of the Optical Microscopy and Analysis Laboratory, Frederick National Laboratory for Cancer Research, Leidos Biomedical Research Inc., and J. Chen, Protein-Nucleic Acid Interaction Section, Center for Structural Biology, NCI for their assistance; B. Miller, S. Fellini and S. Chacko of the NIH HPC computing facility for allocating computing and storage resources for the project; S. G. Tarasov and M. Dyba of the Biophysics Core facility at the National Cancer Institute's Center for Structural Biology for assistance in data acquisition and processing of ITC, ESI mass spectrometry and in dynamic light scattering measurements. We also thank J. Strathern for his vision about RNA. This work is supported by the fund for the NIH/NCI Intramural Research Program (Y.-X.W. and J.Z.).

**Author contributions** Y.-X.W. conceptualized and designed the research. Y.-T.L. and P.Y. prepared the RNA sample. Y.-T.L. recorded all the AFM images of the RNase P RNA. Y.R.B., M.F.S.D. and H.F.D. wrote all computational scripts and programs. Y.-X.W., Y.-T.L. and M.F.S.D. performed calculations. Y.-X.W., Y.-T.L. and M.F.S.D. performed PCA analyses and interpretation. J.R.S. analysed the crystal structure; Y.-T.L., I.S. and J.Z. performed enzymatic assays of RNase P;. L.F., M.F.S.D. and Y.-T.L. collected SAXS data. M.F.S.D. processed and analysed SAXS data. Y.-X.W., Y.-T.L. and M.F.S.D. drafted the text manuscript, and all authors contributed to revisions.

**Competing interests** The authors declare no competing interests.

**Additional information**
**Correspondence and requests for materials** should be addressed to Yun-Xing Wang.

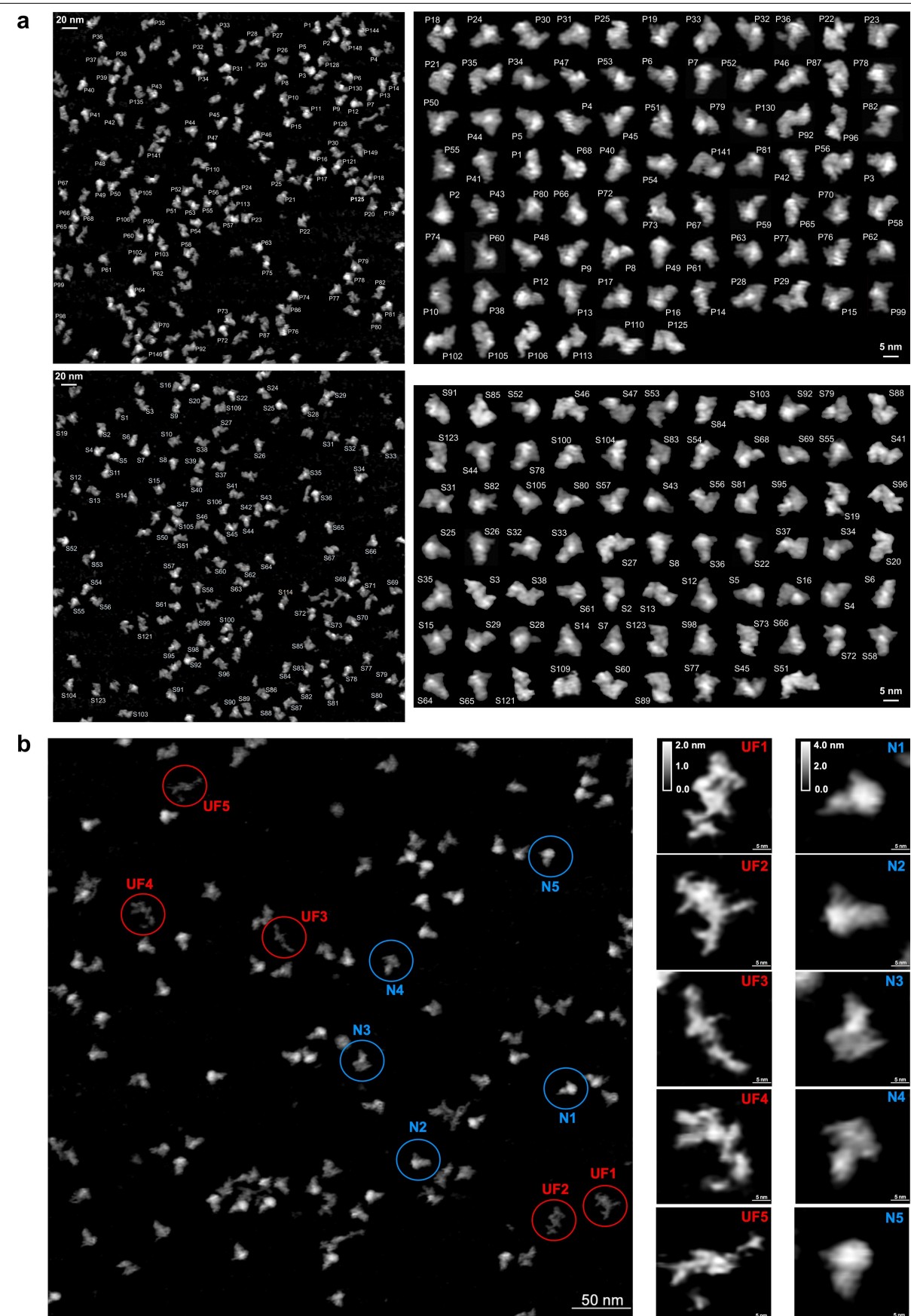

**Extended Data Fig. 1 | AFM images of individual particles. a**, Left: AFM images with labeled particles for single-molecule image cropping. Right: AFM topographies of the 158 individual selected particles that were utilized for structure recapitulation using HORNET. **b**, Left, visualization of the mixture of folded and unfolded molecules. Right: ~15x zoom-in views of molecules.

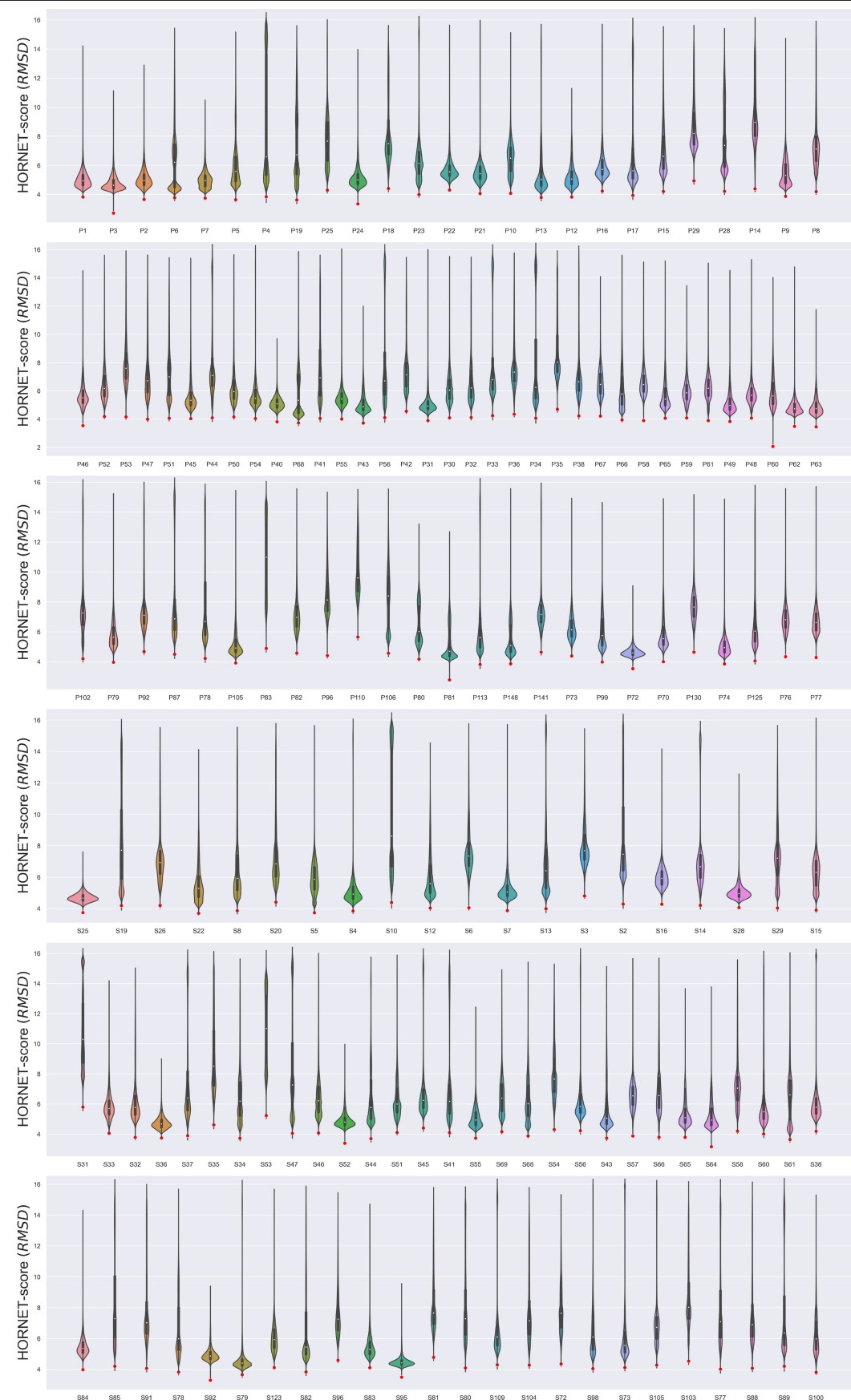

**Extended Data Fig. 2 | Violin convergency plots of the HORNET score for the 158 recapitulated RNase P RNA conformers.** The red dots represent the position of the top selected structure for each particle, with an estimated accuracy of 4.1 ± 0.4 Å RMSD on average. The table of minimum, mean, maximum and sample sizes of the 158 structures is provided in Supplementary Information.

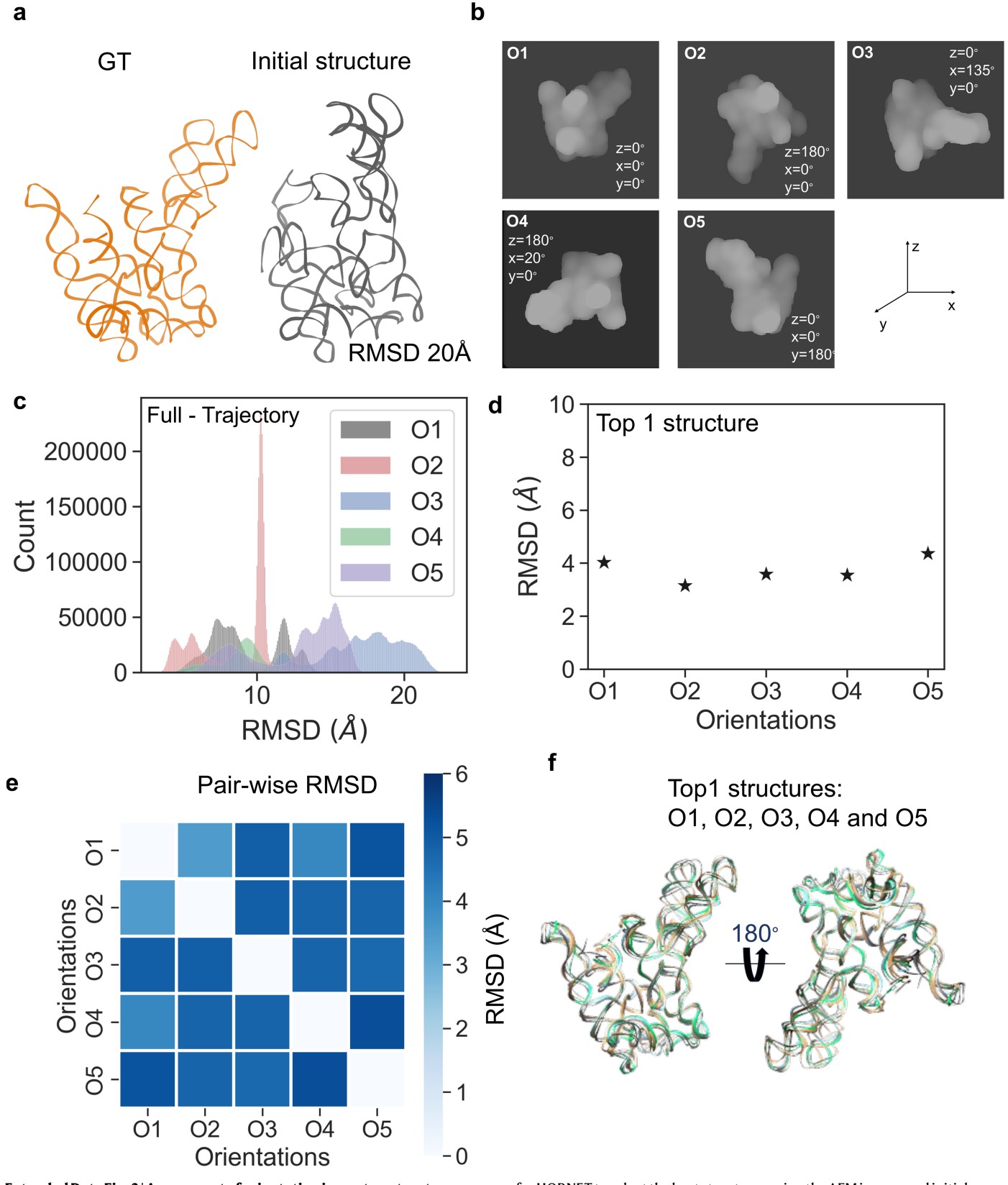

**Extended Data Fig. 3 | Assessment of orientation impact on structure recapitulation of RNase P RNA. a**, Structural recapitulation from an RNase P conformer with initial structure (grey color) that is 20 Å off deviated from a ground truth (GT) structure (orange color). **b**, Simulated AFM images using the GT structure oriented in five distinct poses (O1-O5). **c**, Calculation trajectories for HORNET to select the best structures using the AFM images and initial structure. **d**, RMSDs of the best structures compared to GT structure. **e**, Cross-correlation matrix among the five HORNET determined structures. **f**, Structural superimposition of the best structures recapitulated by HORNET.

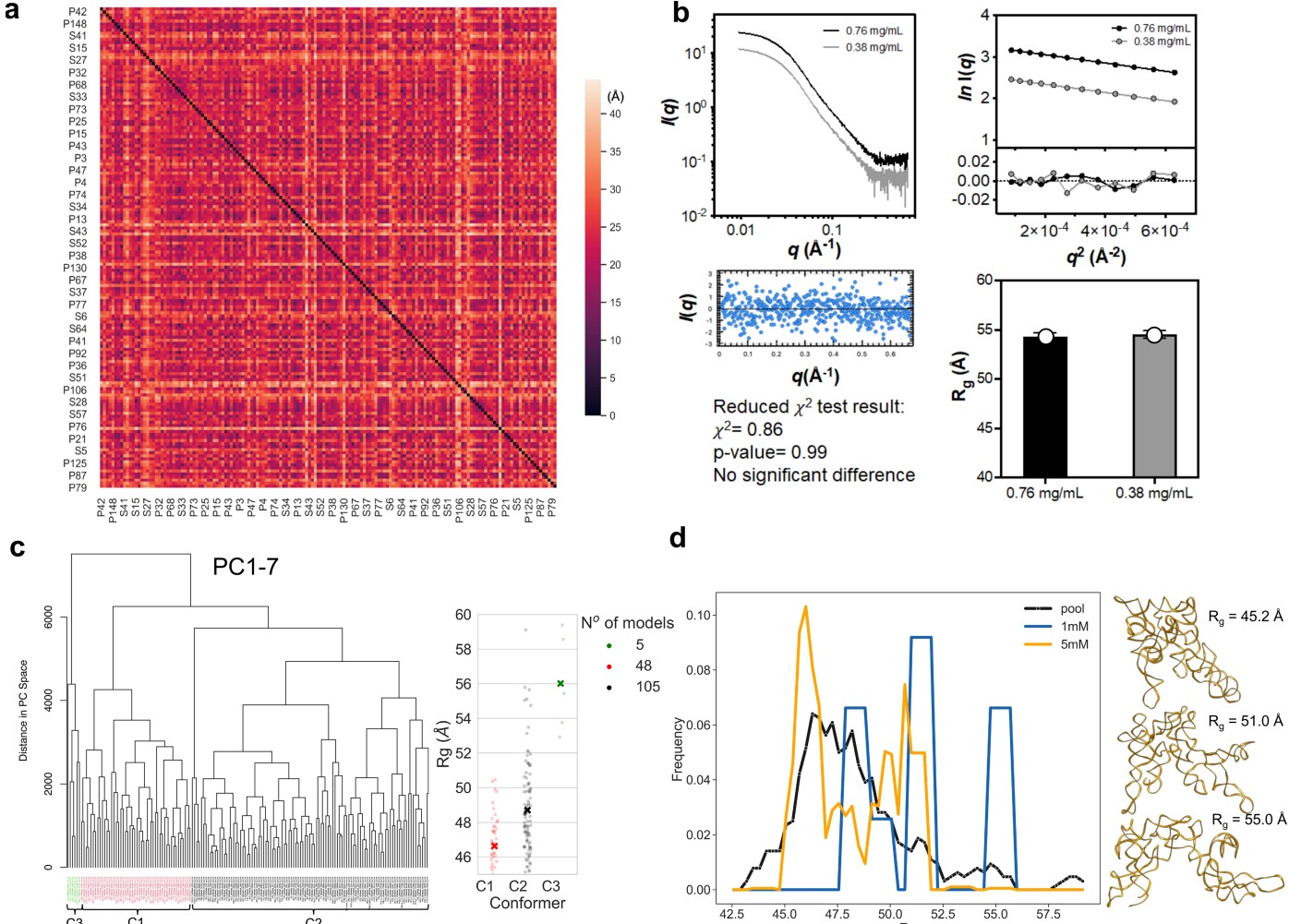

**Extended Data Fig. 4 | Characterization of the motion correlation and the ensemble of the conformers. a**, Pairwise RMSD cross-correlation of the 158 conformers. The axes are only partially labeled with residue numbers due to space congestion. **b**, from top left clockwise, superimposed SAXS profiles recorded at three different concentrations, comparison residual, statistical analyses, Rg values derived from three data set, Guinier regions and the SAXS profiles of the three concentrations without overlaying. **c**, Clustering (classification) analysis of the PCA components 1–7 derived from 158 models. Left, Dendrogram of clustering as a function of PCA distance, right, $R_g$

distribution for each cluster, where the cross symbol represents the average value. **d**, EOM analysis of SAXS intensity profile, left Frequency distribution as function of the ratio of gyration ($R_g$) in angstrom recorded for RNase P RNA at 1 mM Mg$^{2+}$ (blue) and 5 mM Mg$^{2+}$ (orange), using 158 models as a pool of library structure (black). The 5 mM Mg$^{2+}$ data was used here for comparison and consistency check. Right, Representative conformer structure models for 1 mM Mg$^{2+}$ based on the EOM analysis. These three models show similar structural characteristics as the three models shown in Fig. 2d and g.

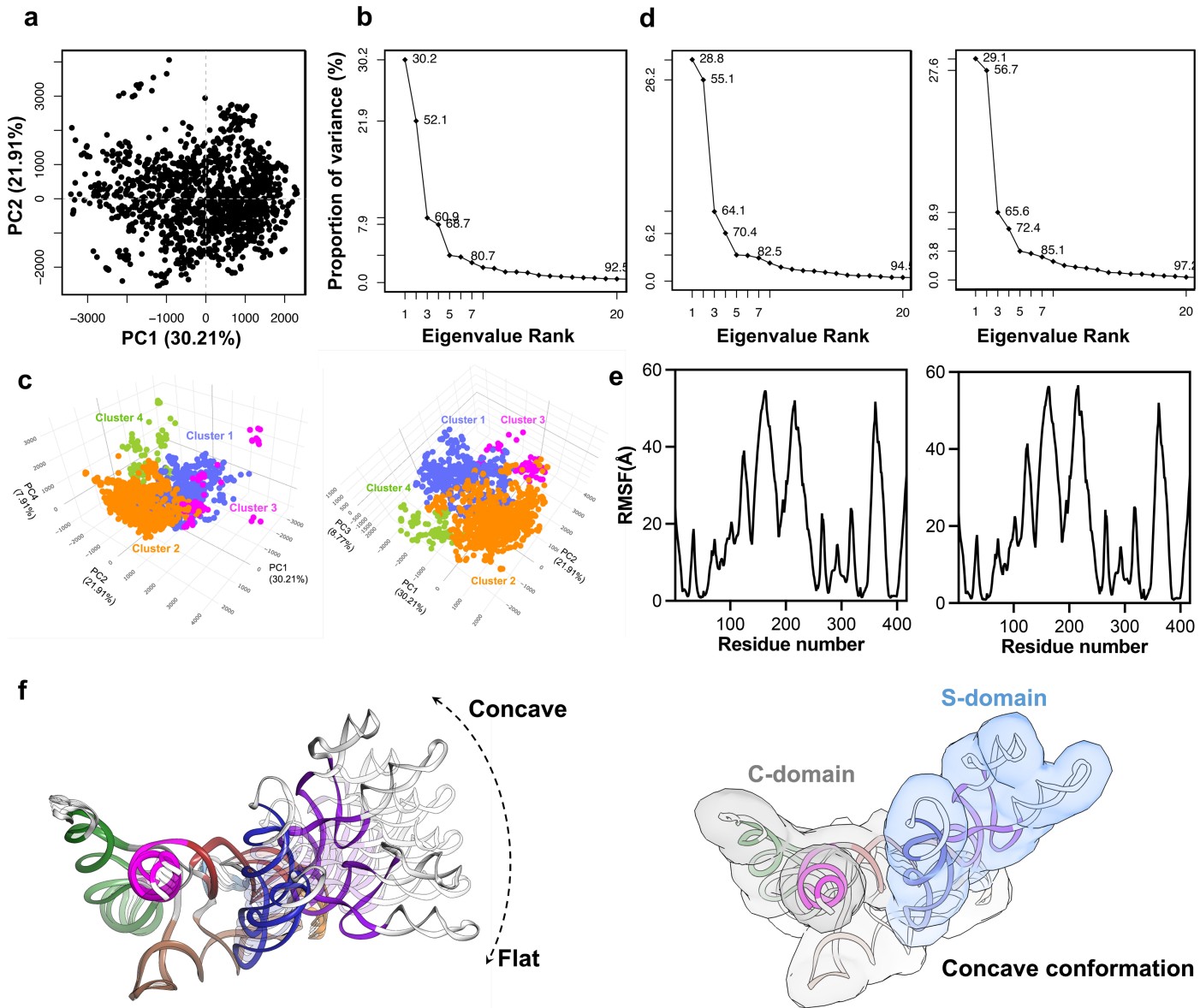

**Extended Data Fig. 5 | Principal component analysis (PCA) of the pool of the top 10 models from HORNET. a**, 2D correlation plot of PC1 and PC2 (left), which cover more than 50% of the conformational variance (right). **b**, Cumulative proportion variability of the data in each component. Five components cover around 80% of the total variance. **c**, 3D plots of PCA scores among the top four principal components (PC1, PC2, PC3, PC4) based on clustering analysis. **d**, proportional structure variances ranked by PCA eigenvalue using the conformers excluding those with Rg > ±1.0 Å (left) or ±0.5 Å (right) of the average Rg values of representative conformers. **e**, Root-mean-square fluctuation (RMSF) of the same pools as in **d**, respectively. **f**, motion between concave and flat conformations (left) and domain arrangement in concave conformation.

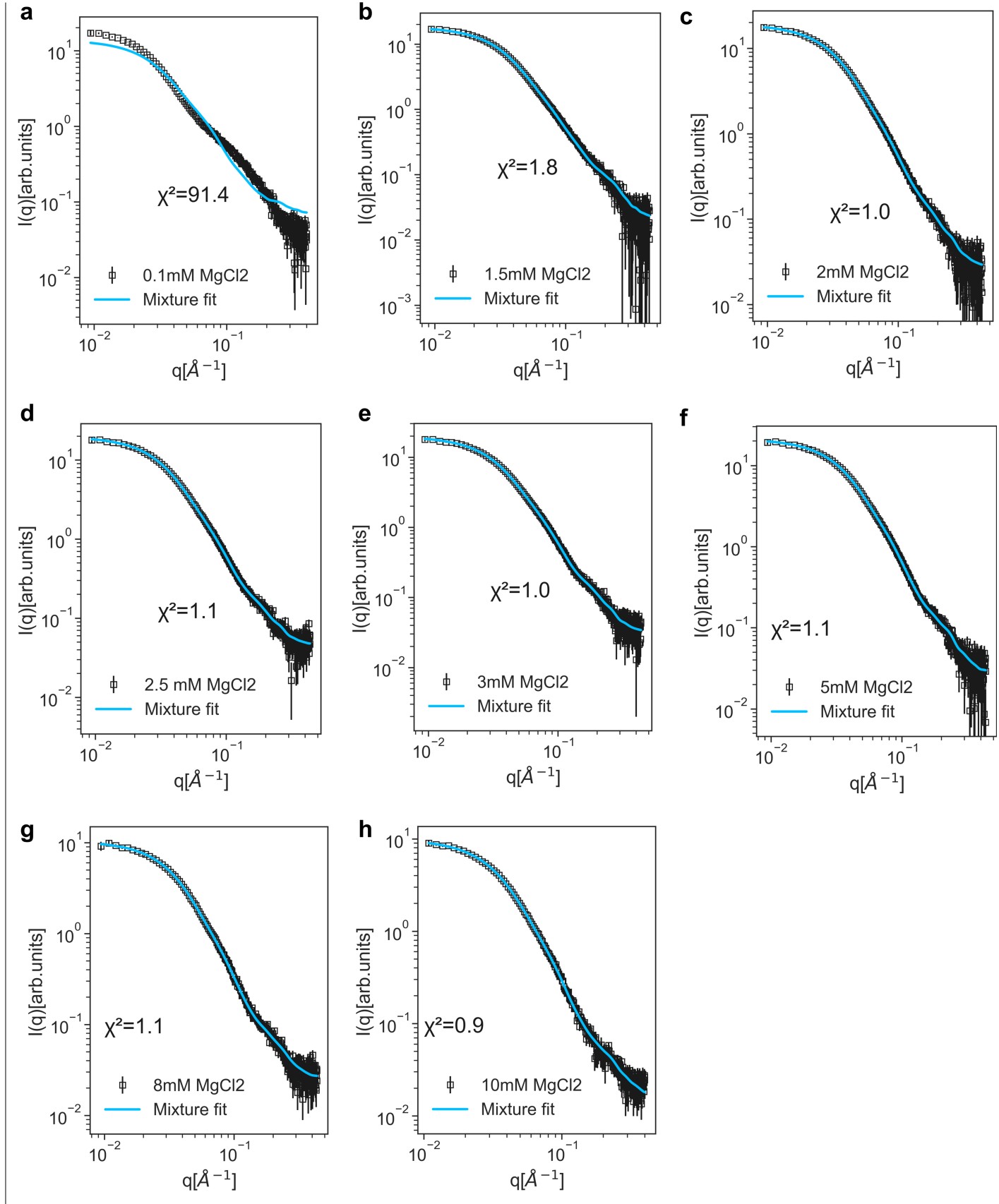

**Extended Data Fig. 6 | Ensemble analysis of SAXS data recorded at different Mg²⁺ concentrations. a–h**, Data recorded at different Mg²⁺ concentrations (black squares). Each SAXS profile is described as a linear combination of representative models at different volume fractions (Mixture fit, cyan), with goodness-of-fit ($\chi^2$) indicated.

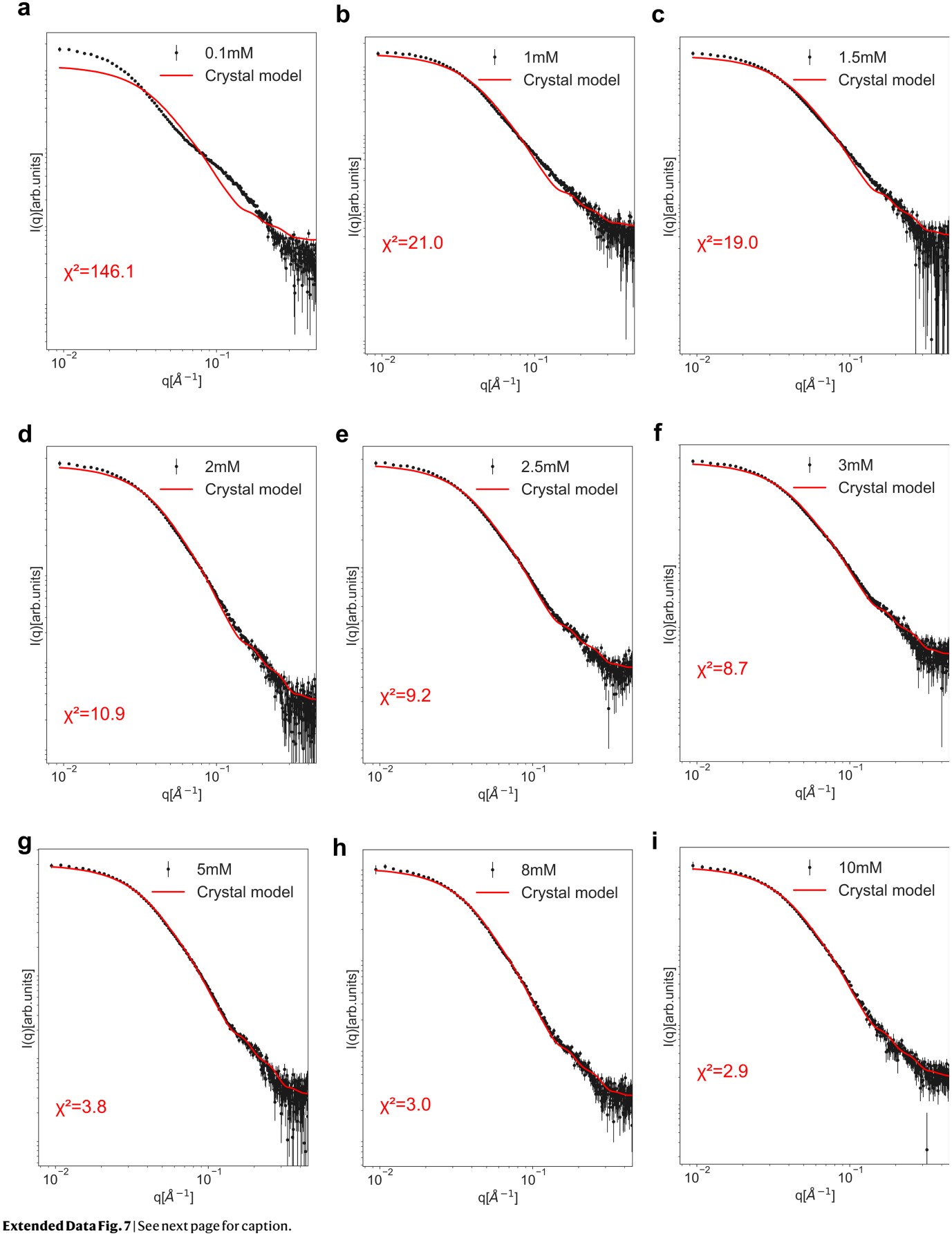

**Extended Data Fig. 7** | See next page for caption.

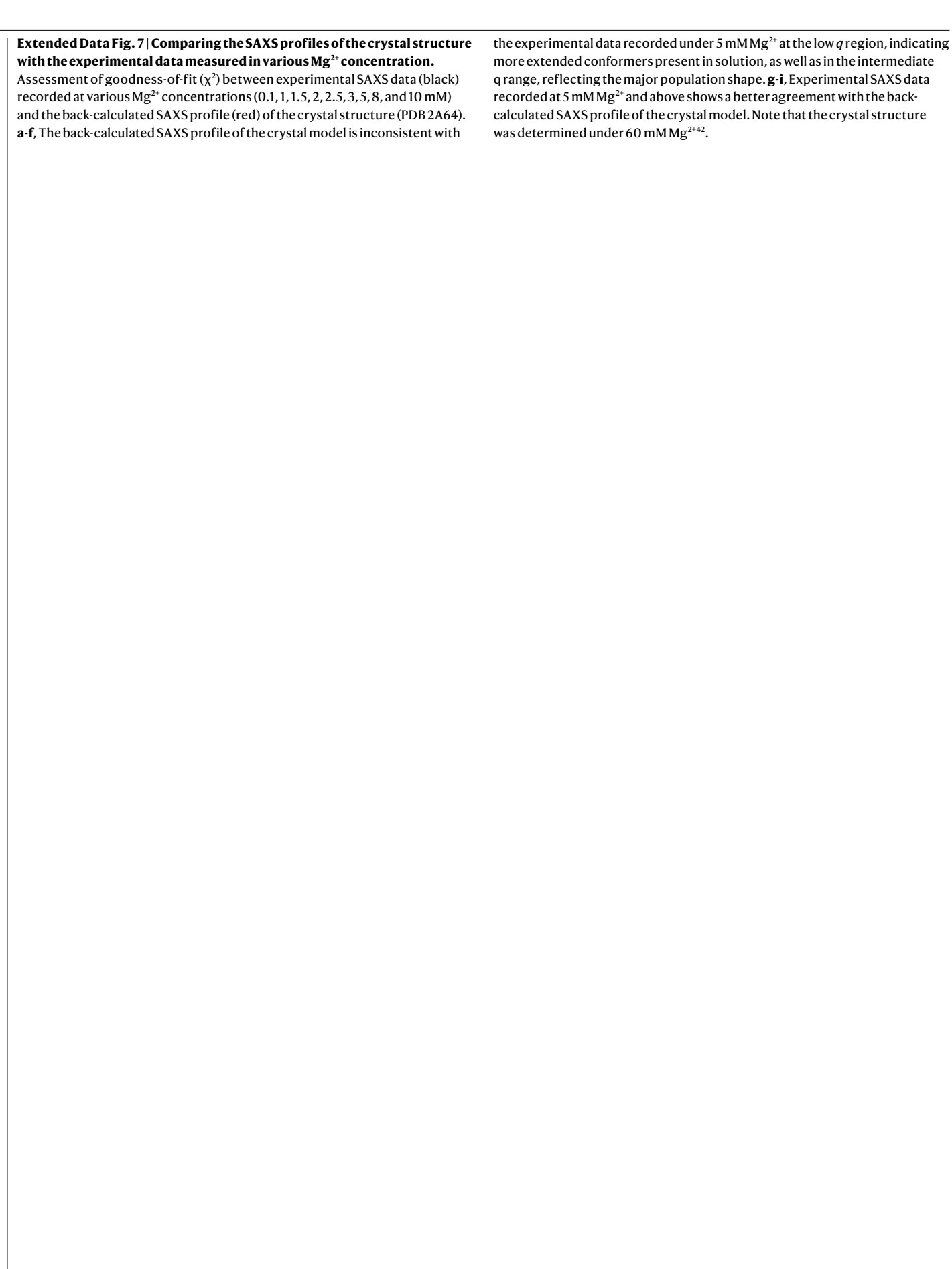

**Extended Data Fig. 7 | Comparing the SAXS profiles of the crystal structure with the experimental data measured in various Mg²⁺ concentration.** Assessment of goodness-of-fit ($\chi^2$) between experimental SAXS data (black) recorded at various Mg²⁺ concentrations (0.1, 1, 1.5, 2, 2.5, 3, 5, 8, and 10 mM) and the back-calculated SAXS profile (red) of the crystal structure (PDB 2A64). **a-f**, The back-calculated SAXS profile of the crystal model is inconsistent with the experimental data recorded under 5 mM Mg²⁺ at the low $q$ region, indicating more extended conformers present in solution, as well as in the intermediate q range, reflecting the major population shape. **g-i**, Experimental SAXS data recorded at 5 mM Mg²⁺ and above shows a better agreement with the back-calculated SAXS profile of the crystal model. Note that the crystal structure was determined under 60 mM Mg²⁺[42].

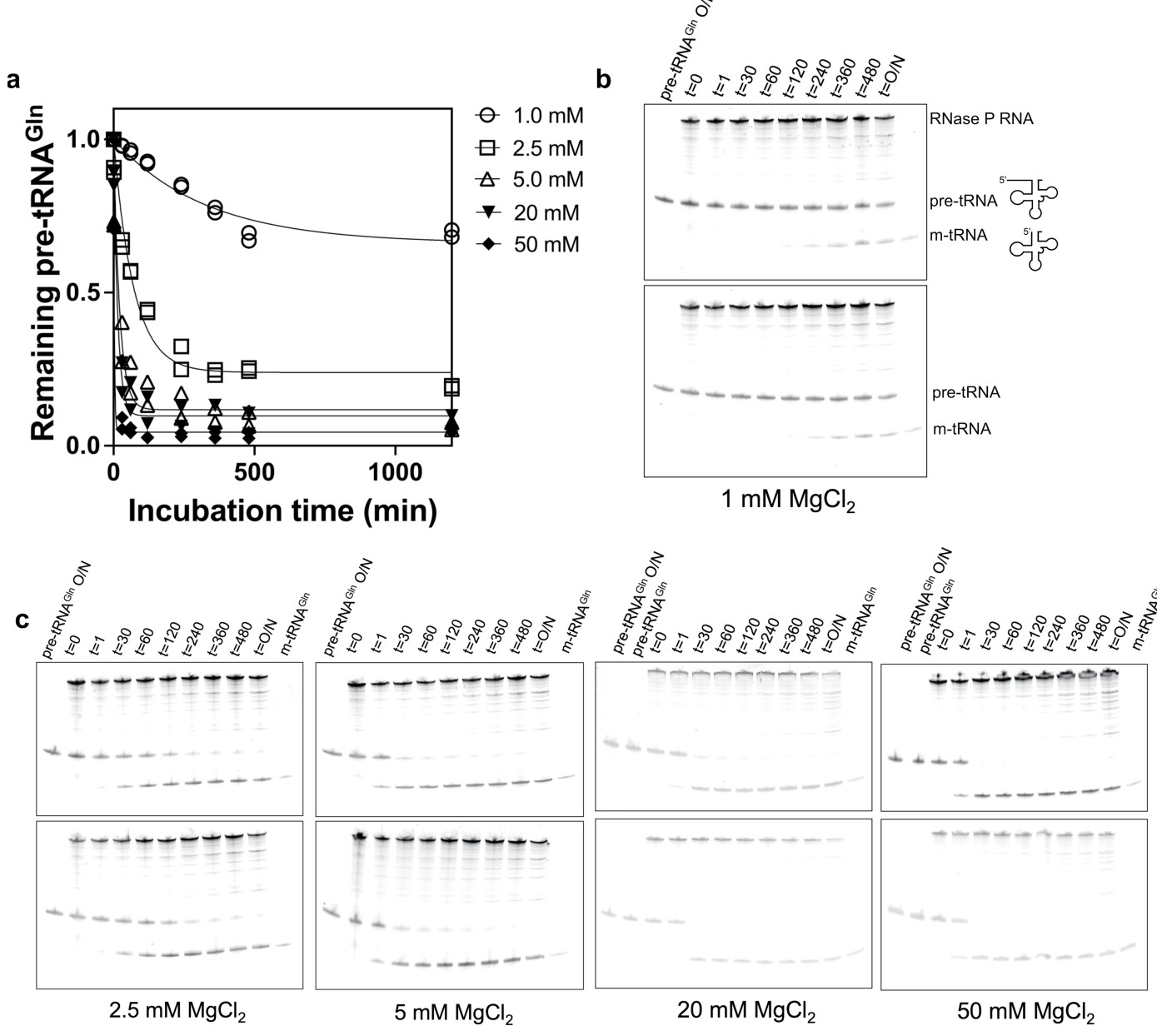

**Extended Data Fig. 8 | Mg²⁺ dependence of the nucleolytic activity of the *Gst* RNase P RNA using human pre-tRNA^Gln as substrate. a**, Time-course measurements of remaining substrate for the nucleolytic single-turnover reaction. Means and mean errors of $n = 2$ independent experiments. **b-c**, Hydrolysis of human pre-tRNA^Gln as a function of time at various concentrations of Mg²⁺ (1, 2.5, 5, 20, 50 mM), examined by denaturing PAGE (10% polyacrylamide/8 M urea) and quantified from the band intensities of the mature (m-) tRNA^Gln and remaining pre-tRNA^Gln substrate. Apparent kinetic rates at the various Mg²⁺ concentrations were derived from the single-exponential fitting of each curve in (**a**) (see Extended Data Table 1).

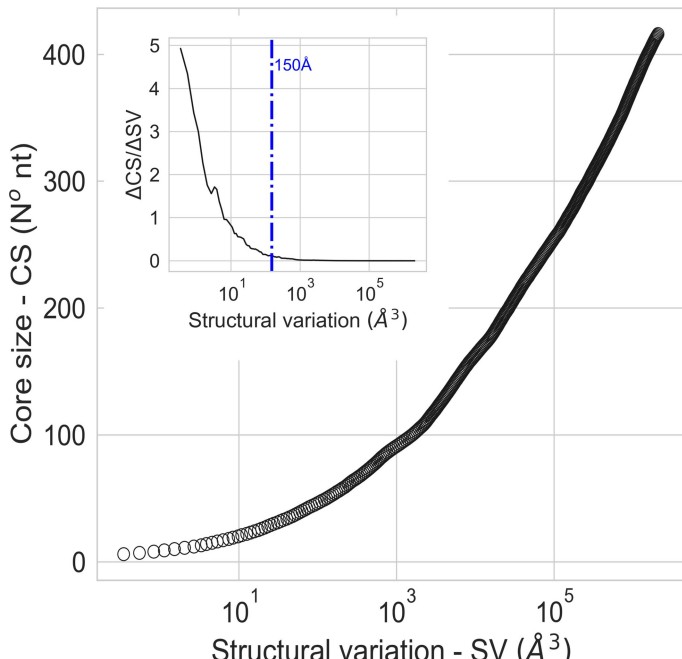

**Extended Data Fig. 9 | Structurally invariant (rigid) core analysis. Core size (CS) of residues as a function of the structural variation, given as ellipsoid volume.** The ellipsoid is built based on the eigenvalues of the Cartesian coordinates after iterative rounds of structure superposition. Atoms that are furthest outside the ellipsoid volume after each round are excluded in the next round of superposition. The plot shows that some residues are present only in the initial largest ellipsoid volume, and the number of included residues decreases exponentially to a minimum slope of variation. The inset plot shows the first derivative of the core size as a function of the ellipsoid volume, which plateaus at a volume of -150 $\text{Å}^3$, where two global minimums are defined (see Fig. 5a).

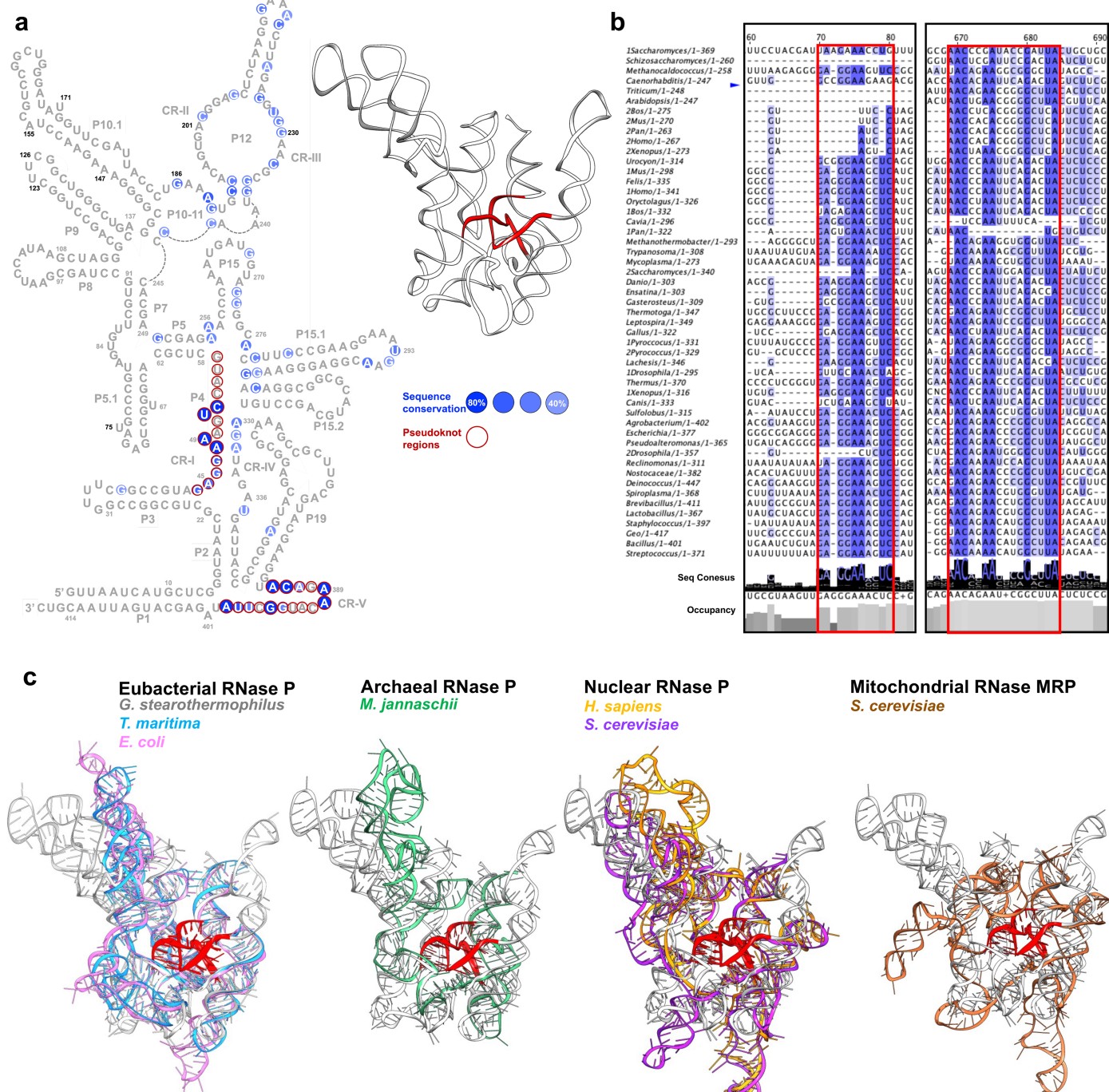

**Extended Data Fig. 10 | Phylogenetic and structural comparison of *Gst* RNase P RNA with RNase P RNAs from different species. a**, RNase P RNA sequence conservation across the three kingdoms of life (eubacteria, archaea and eukaryota). **b**, The multiple sequence alignment (filename: All_clustal.sto) of 51 RNase P sequences across the three kingdoms of life in panel b has been deposited on line in our share space: https://home.ccr.cancer.gov/csb/pnai/ data/conformational_space/. Conserved residues are shown in various shades of blue according to the level of conservation, and the pseudoknot core is indicated in red. **c**, 3D-structure comparison of bacterial, archaeal, eukaryotic nuclear, and mitochondrial RNase P RNAs. The structure of the *Gst* RNase P RNA (grey) from this study is shown in each superposition for reference. All structures are color-coded according to species and are aligned against the structurally invariant residues (red) identified in this study.

**Extended Data Table 1 | Volume fractions of conformer classes used for ensemble SAXS fitting. Note that the decimals are not rounded up**

**a**

| $[Mg^{2+}]$ (mM) | Class1 | Class2 | Class3 |
|---|---|---|---|
| 0.1 | 2.96 | 97.10 | 0 |
| 1 | 18.73 | 75.30 | 6.08 |
| 1.5 | 58.90 | 41.28 | 0 |
| 2 | 86.65 | 10.76 | 2.71 |
| 2.5 | 95.47 | 0 | 4.14 |
| 3 | 94.08 | 0 | 5.57 |
| 5 | 96.46 | 0 | 2.68 |
| 8 | 98.61 | 0 | 1.44 |
| 10 | 94.43 | 0 | 5.67 |

**b**

| $[Mg^{2+}]$ (mM) | $k_{app}$ (min$^{-1}$) |
|---|---|
| 1 | $3.26 \times 10^{-3} \pm 0.00052$ |
| 2.5 | $12.10 \times 10^{-3} \pm 0.0013$ |
| 5 | $38.37 \times 10^{-3} \pm 0.0076$ |
| 20 | $62.76 \times 10^{-3} \pm 0.0093$ |
| 50 | $362.70 \times 10^{-3} \pm 0.023$ |

**c**

| $[Mg^{2+}]$ (mM) | $R_h$ (Å) | $R_g$ (Å) |
|---|---|---|
| 0.1 | 67.50±0.09 | 61.6±0.04 |
| 1 | 65.74±0.04 | 54.6±0.03 |
| 1.5 | 61.12±0.07 | 51.0±0.03 |
| 2 | 58.60±0.07 | 49.9±0.04 |
| 2.5 | 58.20±0.05 | 48.5±0.03 |
| 3 | 57.92±0.07 | 48.5±0.03 |
| 5 | 57.55±0.08 | 48.6±0.03 |
| 8 | 57.84±0.05 | 48.5±0.03 |
| 10 | 58.34±0.05 | 48.4±0.04 |

# Reporting Summary

## Statistics

For all statistical analyses, confirm that the following items are present in the figure legend, table legend, main text, or Methods section.

| n/a | Confirmed | |
|---|---|---|
| ☐ | ☒ | The exact sample size ($n$) for each experimental group/condition, given as a discrete number and unit of measurement |
| ☐ | ☒ | A statement on whether measurements were taken from distinct samples or whether the same sample was measured repeatedly |
| ☒ | ☐ | The statistical test(s) used AND whether they are one- or two-sided<br>*Only common tests should be described solely by name; describe more complex techniques in the Methods section.* |
| ☐ | ☒ | A description of all covariates tested |
| ☐ | ☒ | A description of any assumptions or corrections, such as tests of normality and adjustment for multiple comparisons |
| ☐ | ☒ | A full description of the statistical parameters including central tendency (e.g. means) or other basic estimates (e.g. regression coefficient) AND variation (e.g. standard deviation) or associated estimates of uncertainty (e.g. confidence intervals) |
| ☒ | ☐ | For null hypothesis testing, the test statistic (e.g. $F$, $t$, $r$) with confidence intervals, effect sizes, degrees of freedom and $P$ value noted<br>*Give P values as exact values whenever suitable.* |
| ☒ | ☐ | For Bayesian analysis, information on the choice of priors and Markov chain Monte Carlo settings |
| ☒ | ☐ | For hierarchical and complex designs, identification of the appropriate level for tests and full reporting of outcomes |
| ☐ | ☒ | Estimates of effect sizes (e.g. Cohen's $d$, Pearson's $r$), indicating how they were calculated |

*Our web collection on statistics for biologists contains articles on many of the points above.*

## Software and code

Policy information about availability of computer code

| Data collection | Asylum atomic force microscope was used to acquire AFM images, and CafeMol 3.02 was used for generating the Dynamic Fitting data.<br>The SAXS data were recorded using the in-house instrument at the NCI SAXS core facility (BioSAXS-2000, Rigaku) in the Center for Structural Biology.<br>Dynamic light scattering were collected at NCI Biophysics Resource facility in the Center for Structural Biology using DynaPro Plate Reader III Dynamic Light Scattering instrument (Wyatt Technologies).<br>Data acquisition of ESI-MASS spectrum were performed on 6520 Accurate-Mass Q-TOF LC/MS system equipped with a dual electro-spray source at NCI Biophysics Resource facility in the Center for Structural Biology. |
|---|---|

| Data analysis | Novel software for data analyses in this study:<br>HORNET  - Holistic RNA Structure Determination (v1.0.0). Zenodo. https://doi.org/10.5281/zenodo.10637777<br>Script for PCA and SAXS ensemble fitting and chi2 mapping can be found at https://home.ccr.cancer.gov/csb/pnai/data/conformational_space/Conf_space_RNasePRNA/scripts_analysis/.<br><br>Other existing softwares used in this study:<br>Gwyddion 2.65 for AFM data processing.<br>ATSAS 3.2.1 software package was used for SAXS data analysis.<br>ImageJ 1.54h was used to quantify the pre-tRNA digested band intensity on PAGE gel.<br>Mass Hunter Qualitative Analysis software (version B.07.00) was used for ESI-MASS data analysis and deconvolution of mass spectra.<br>JalView (2.11.2.7) was used for phylogenetic analysis of multiple sequence alignments of RNase P RNA.<br>PyMol 2.5.4 and UCSF Chimera 1.16 were used to visualize and display RNA structure for figure preparation.<br>COOT (0.9.8.95.EL) was used for RNA structure regularization. |
|---|---|

For manuscripts utilizing custom algorithms or software that are central to the research but not yet described in published literature, software must be made available to editors and reviewers. We strongly encourage code deposition in a community repository (e.g. GitHub). See the Nature Portfolio guidelines for submitting code & software for further information.

## Data

Policy information about availability of data

All manuscripts must include a data availability statement. This statement should provide the following information, where applicable:
- Accession codes, unique identifiers, or web links for publicly available datasets
- A description of any restrictions on data availability
- For clinical datasets or third party data, please ensure that the statement adheres to our policy

Data available for public access at: https:/home.ccr.cancer.gov/csb/pnai/data/HorNet/ and https://home.ccr.cancer.gov/csb/pnai/data/conformational_space/Conf_space_RNasePRNA/scripts_analysis/

Rfam database for RNase P RNA phylogenetic analysis: https://rfam.org/family/RF00011

SAXS data are available at SAS data bank https://www.sasbdb.org/project/2201/b9y8c4b6qf
and the SASBDB accession codes:  SASDTA7, SASDTB7, SASDTC7, SASDTD7, SASDTE7, SASDTF7, SASDTG7, SASDTH7, SASDTJ7.

## Research involving human participants, their data, or biological material

Policy information about studies with human participants or human data. See also policy information about sex, gender (identity/presentation), and sexual orientation and race, ethnicity and racism.

| Reporting on sex and gender | N/A |
|---|---|
| Reporting on race, ethnicity, or other socially relevant groupings | N/A |
| Population characteristics | N/A |
| Recruitment | N/A |
| Ethics oversight | N/A |

Note that full information on the approval of the study protocol must also be provided in the manuscript.

# Field-specific reporting

Please select the one below that is the best fit for your research. If you are not sure, read the appropriate sections before making your selection.

☒ Life sciences  ☐ Behavioural & social sciences  ☐ Ecological, evolutionary & environmental sciences

For a reference copy of the document with all sections, see nature.com/documents/nr-reporting-summary-flat.pdf

# Life sciences study design

All studies must disclose on these points even when the disclosure is negative.

| Sample size | 161 AFM single-molecule images for illustration of RNase P RNA conformational space. following PCA validates the sample size is sufficient to cover all covariances. The resulting 158 recapitulated structures are thus used for SAXS analysis.<br>SAXS and dynamic light scattering profiles were recorded at 9 different Mg2+ concentrations.<br>Enzymatic duplicates assays for RNase P RNA were conducted at 5 different Mg2+ concentrations. |
|---|---|

114 RNase P RNA sequences with well-defined secondary structure from different bacterial species were used for multiple sequence alignment.

**Data exclusions**
AFM particle images that were overlapped with others or aggregate were not used for structural determination and 3 particles that are not converge in reasonable computational time were excluded.

**Replication**
Enzymatic assays for RNase P and SAXS data were duplicated and repeated after 24 hours. The results were successful obtained.

**Randomization**
Randomized analyses were performed to observe the convergence of SAXS data fitting using the 158-particle ensemble.

**Blinding**
This approach was not applicable in this study where had no need of performed blind validation since AI model was trained, which is not amenable to blinding.

# Reporting for specific materials, systems and methods

We require information from authors about some types of materials, experimental systems and methods used in many studies. Here, indicate whether each material, system or method listed is relevant to your study. If you are not sure if a list item applies to your research, read the appropriate section before selecting a response.

## Materials & experimental systems

| n/a | Involved in the study |
|-----|----------------------|
| ☒ | Antibodies |
| ☒ | Eukaryotic cell lines |
| ☒ | Palaeontology and archaeology |
| ☒ | Animals and other organisms |
| ☒ | Clinical data |
| ☒ | Dual use research of concern |
| ☒ | Plants |

## Methods

| n/a | Involved in the study |
|-----|----------------------|
| ☒ | ChIP-seq |
| ☒ | Flow cytometry |
| ☒ | MRI-based neuroimaging |

