## [Peer Review File · Nature]

The conformational space of RNase P RNA in solution

Corresponding Author: Dr Yun-Xing Wang

Version 0:

Reviewer comments:

Referee #1

In this study, the authors utilise solution atomic force microscopy (AFM) in conjunction with deep neural networks to explore the diverse conformations of RNase P RNA, revealing a stable core surrounded by dynamic peripheral elements. Additionally, the study demonstrates that heightened Mg²⁺ concentration fosters RNA compaction and enhances enzymatic activity, likely attributable to reduced conformational variability. Crucially, this research establishes a nexus between spatial flexibility and sequence conservation, illuminating the intrinsic functional roles encoded within the RNA's primary sequence. This insight significantly advances our comprehension of the enzymatic precision and substrate diversity of RNase P RNA.

Nevertheless, the deep learning approach in this study, HORNET, presented in this article originates from the authors' previous work, "Determining Structures of Individual RNA Conformers Using Atomic Force Microscopy Images and Deep Neural Networks" (10.21203/rs.3.rs-2798658/v1). This earlier study describes the application of HORNET to map the three-dimensional structures of various large RNAs, including RNase P RNA. In the current article, the HORNET approach is utilised to analyse the diverse conformations of RNase P RNA, along with their biological implications. Therefore, this paper can be considered as complementary to the authors' previous work, further elaborating the conformation of RNase P using the HORNET method, providing insights into the complexity of its structure and its functional implications. Therefore, I would suggest that synthesising these two articles into one would significantly benefit both the AFM and RNA structure communities, enhancing the overall impact and relevance of the research.

The manuscript is commendably written, with its language being both articulate and accessible. The authors have structured their arguments in a logical and coherent manner, making the complex subject matter readily comprehensible to readers. There are also a few minor errors like figure notes and codes that I've listed below.

Minor comment:

1. The right plot in Ext. Data Fig. 1 totals 161 particles, which does not match the 158 described in the figure notes. A similar quantity problem is found in Ext. Data Figs. 2,3 and 4, which contain 165 images but are labeled 158 in figure legend. Also, Ext. Data Fig. 5 has 154 violin plots (158 in figure legend). Please check.
2. Ext. Data Fig. 11 contains 3 subfigures and figure legend mismatches. There are three subfigures a,b,c but only a and b in the figure legend.
3. I checked the code section. The HORNET section is out of the scope of this review. Other scripts such as PCA and SAXS data processing scripts, although there is introduction to their main functions, but due to the lack of corresponding test files so can not be tested through.

Referee #2

In "The conformational space of RNA in solution" by Yun-Tzai Lee, Maximilia F. S. Degenhardt, Ilias Skeparnias, Hermann F. Degenhardt, Yuba R. Bhandari, Ping Yu, Jason R. Stagno, Lixin Fan, Jinwei Zhang, Yun-Xing Wang, work is described that uncovers the structural flexibility of RNase P RNA.

The work is quite technical - in a good way - as it opens up a novel avenue of research that enables RNA structural biology.

The main enabling technology is high-resolution atomic force microscopy (AFM) combined with machine learning algorithms, principal component analysis, SAXS, and sequence analysis.

In brief, RNA structural biology remained challenging as current structural biology methods (X-ray diffraction of 3D crystals) only provided snapshots, which, especially in the case of RNA-based enzymes with high flexibility and functional promiscuity, may miss important aspects, namely how conformational heterogeneity is an inherent and necessary component of this class of molecules.

The authors present high-resolution AFM images from which they extract molecular observations that they then compare to structures, and model the structural variance using computational methods. The author make a compelling case that X-ray crystallography can only derive a single conformer of the molecule under investigation, which in addition might be biased by the crystal contacts and crystallization conditions. So far, cryo-EM has not been very much used to analyze RNA enzyme structures, likely due to their small size. In contrast, solution AFM offers itself as an alternative to study structural variability in liquid at ambient temperature and pressure and close-to-physiological conditions.

They find that RNase P RNA has a structural core that is quite constant and features peripheral elements that are highly flexible and undergo large scale fluctuations. Interestingly, conserved regions are comprised in the firm core as well as in the flexible parts, interpreted that the primary sequence encodes flexibility too, which I find an interesting aspect to think deeply about. While this reviewer thinks that the authors should use a more specific title - as we do not yet know how general their findings are ("The conformational space of RNA in solution" is clearly a much too general title), I do appreciate the technical quality and focus of the work, as I appreciate research breakthroughs that may open fields rather than reports on specific findings on a specific molecule. I am in principle excited about the possible impact and reach of this work, but I have major concerns regarding the orientating vs conformational variability, as I detail below.

Major:

AFM is a surface technique. As such the molecules that are surface adsorbed can adopt an almost infinite number of possible orientations on the surface.

1) This implies the first major question: how does surface adsorption impact structure, and how can we make sure we interpret the surface structures correctly towards 3D structures?

2) The second major question concerns the in principle infinite number of orientations. The identical molecular conformation could in principle result in a huge number of topographies dependent on observation orientation. In this context, the 158 conformers, which I think the authors refer to as major observation classes (they have more observations, Ext. Data Fig. 1) could thus report conformational variability or orientational variability. It is unclear to me how the authors unambiguously tackle this problem without a priori knowledge of the conformation. This is a major drawback, and I suspect the authors might need more data. It remains unclear how the authors separate orientational from conformational variability and how very different observation orientations are pooled into the same conformer using the described method, vs the coincidental emergence if similar observations from different conformations in different orientations.

Minor Comments:

Page 1, Line 1: Change the title to a more specific one. Researchers will understand the general aspect of your efforts.

Page 3, Line 4: Name what you call single molecule approaches. I can only assume that the authors refer to optical methods. It is however a pity, because the authors themselves do single molecule research, and the (in my view erroneous) dominant use of "single molecule" for certain techniques is not a positive tendency for all other single molecule research.

Page 3, Line 18: The sentence is not understandable for anyone who does not work on precisely that problem.

Page 4, Line 23: Not only "conformational differences"--there are also conformational similarities. Indeed, Fig. 1 panels c,d actually reveal notable similarities.

Page 6, Line 2: While molecules in crystals can explore some conformational space, this is only very limited (comprised in the B-factor) but only one structure is the result. Thus the phrasing "sample a much wider" is confusing as the other technique (crystallography) literally does not sample at all.

Page 6, Lines 11, 17, etc, and Fig. 2 caption: The writing is very confusing with regard to the 158 conformers. I did not understand how the authors came to that number; clearly they have many more observations (Ext. Data Fig. 1). I assume they pool observations in conformer classes. In addition there might be identical conformers but different views. Provide more details about observations, orientations, conformations, conformers, etc (see major concern).

Page 10, Line 29 and following: Since the manuscript is quite technical, I would appreciate more details regarding the PCA in the main text.

Page 11, Line 11: While "motions of grass on the sea floor" is poetical, I would prefer a more scientific description that does not appeal to each individuals vacation background for a better understanding of the results.

Referee #3

This paper describes the application of a new method for determining the structures of RNA in an ensemble, by interpreting solution AFM data using a new deep neural network technique that is detailed in a companion manuscript. Here, the focus is on determining the 'functional' motions of a particular RNA enzyme. The level of detail, specifically the information about the conformational degrees of freedom of the RNA, is quite impressive and represents a needed advance to the field. However, I have a number of questions/concerns about the interpretation of the experiment. Given that the methodology will be published elsewhere, it seems that the 'science' story is central to this work, so author responses will be critical in assessing the impact of the scientific story.

An alternative interpretation of the experimental results are as follows. What if they are measuring the folding of the RNA, not its functional motions (which I interpret to mean the motions of the folded enzyme that catalyze reactions). Fig. 4 shows the Mg-dependence of the conformations (from SAXS data). Fig. 4d closely resembles the equilibrium curves from a folding assay. In this case, C2 represents unfolded conformations (meaning conformations which are extended because the electrostatic compensation by the ions is not complete). Although functional folding motions are in their own right interesting, the assumption in the paper is that the molecule is in its physiologically active state at 1 mM Mg. What if it is in a superposition of unfolded and folded states, so it can partially carry out its function? It is impossible to know if the molecule is physiologically folded from these measurements, despite the statement that 1 mM Mg is physiological. Yes, that is true, but the other salt in the buffer is below physiological levels (100 mM NaCl, for example, not to mention that other ions contribute to overall ionic strength), so it is possible that 1 mM Mg is just not enough to fold the RNA. If the motions do indeed represent essential conformational dynamics for function, why does the enzyme become more efficient above the point where it is fully folded (5 mM Mg)?

Furthermore, key experimental details are not given in the paper. What is the [RNA] for the experiments, is the Mg just added or are the samples dialyzed (in other words, is the 1 mM Mg a 'free' or 'total' ion concentration)?

It is also concerning that, given their arguments about the importance of conformational variations, a *single* conformer is sufficient to represent C1, C2 and C3. This seems to contradict the main point of the manuscript, that multiple, distinct conformations are present. Additionally, from the information presented in the paper it is unclear if all of the atoms are present in the conformers. For example, C1 should be close to the crystal structure (I assume this by comparing data from Ext. Data Fig. 6 to the coefficients of Fig. 2d; at 10 mM Mg, the chi-square value for the fit to the crystal structure looks acceptable, and it looks like the population is nearly 100% C1.) This strikes me as odd because the crystal structure is *missing* multiple nucleotides, so how can the profiles be so similar? Are all of the nucleotides included in C1? This is not at all clear from what is presented in the paper and the pdb files for the C1, C2 and C3 structures do not appear to be available.

My last major point regards the use of crysol for predicting the SAXS profiles. Although crysol works reasonably well for proteins, it is not the best to use for RNA as it does not correctly capture the solvent effects. This, together with the use of only three states, diminishes enthusiasm for the fitting procedure. Please see Bernetti et al. (Eur. Phys. J. B (2021) 94:180) for a discussion about the benefits of using different packages to simulate SAXS profiles.

Other less critical points

It is also disappointing that the references to conformational dynamics mostly reference work done many years ago. The authors seem to have missed a small, but significant amount of literature modeling SAXS data with multiple conformers (for two examples, see Bernetti et al., NAR, 2021 gkab459, and He et al., NAR, 2023 gkad809).

I find the repeated emphasis on their measurements of the 'motions' to be overstated. If you are measuring motions, you MUST have a time axis or a time scale. What is measured are 'ranges of motion', or better yet, the full structural ensemble. Interpolation between the different structures can suggest the pathways for motion (as they do), but they do not directly report on motion itself. In fact, they have it exactly right in the abstract; they are mapping the conformational space, NOT determining the motions.

Version 1:

Reviewer comments:

Referee #1

I appreciate the authors' feedback on my comments. I have no further comments to add.

Referee #2

The manuscript, "The conformational space of RNase P RNA in solution", by Lee et al. is the revised version of a former paper entitled "The conformational space of RNA in solution". I have been quite positive about the paper in my initial assessment, and I still am, but I do not think that the paper was improved in the revision cycle.

The authors seemed reluctant to consider the referees' comments, where it mattered most.

1. I concur with referee #1 that the separation of method and biological finding in two individual manuscripts is somewhat unfortunate. Clearly, given that the authors have the two manuscripts in parallel in consideration, indicates that referee #1 is right that they are somewhat interdependent. At least to the point that this manuscript is difficult to follow and appreciate without reading entirely the methods paper, while the methods paper contains to a minor extent some aspects presented here. The authors argue that space and authorship warrants for two independent manuscripts and I understand them. On the other hand they might run the risk that readers might turn away from these findings as they may not find the entire information that leads from conceptualization over analysis to result. This referee thinks that the modern publishing format with extended data figures and co-first authorship would allow for such a combined manuscript, and that such a combined paper would have stronger impact than the two manuscripts alone, I can only recommend consideration.

2. I concur with referee #3 that the use of the word 'motion' is inaccurate, actually wrong. Even if the authors now define "motion as spatial motion without a temporal aspect", the word just doesn't match the findings. The findings have no information about motion, neither temporal nor directional, nor correlated motions, etc. The authors have plenty of words at hand like conformational space, conformational variability, etc, that are not words that represent an action. No action, no dynamics is reported here.

3. My own major criticism concerned the multiplicity of conformations and the theoretical multiplicity of adsorption orientations, and that multiple conformations in different orientations could potentially have similar topographies, while the same conformation in multiple orientations could be represented by multiple different topographies. The authors answer that the RNA molecules, unlike proteins, have one or a few flat surfaces and therefore their assumption is that they are looking at conformational variability of molecules that are all adsorbed on the surface in very similar fashion. Maybe. I tend to believe the authors, but think this is a significant potential drawback. The authors do not mention this in the manuscript, while I think the readers should be made aware of this question. As the methods of cryo-EM advance the concepts of orientation and conformational variability will become only more common and the authors have an interest to detail precisely what they think and analyze. Also, I commented about the somewhat confusing use of particles, observations, orientations, conformations, conformers, etc. Initially I thought that there were 158 classes (conformations, conformers) and that each class contained tens of particles (molecules, observations). From the explanations in revision, I seem to understand that there are only 158 particles and each particle is an individual conformation. This confuses me a lot. Each of the images in Figure 1a,b alone has more than 100 particles. I assume the authors have collected many such images (how many, I could not find in the Methods). These images look great. Most, >90% of the particles are nicely isolated and well resolved, so how do the authors only come up with 158 particles? Do I understand this correctly? If yes, how were they selected, using what criteria? In Fig. 2, the authors show a gallery of the structural models, and then gather them in three classes (f) or conformers (c). Which models in (a) are in class/conformer C1, C2 or C3, and based on what are they grouped (I understand that these three classes are enough to precisely fit the SAXS curve)?

In summary, I still think this is an interesting paper with high quality primary data and an interesting analysis method/concept and outcome. However, while the revision tackled the details, it did not resolve some of the major questions.

Minor:

82: I still do not understand what conformational space molecules packed in a crystal lattice sample.

50: When I commented that the sentence was not understandable for anyone who does not work on precisely that problem, I didn't mean for the authors to delete it or tell me that they keep the sentence. As an intermediate, the authors could rephrase the sentence in a way that readers from slightly outside the field could also understand it.

Referee #3

Unfortunately, I do not concur with the authors' response to my question about whether or not the RNA is fully folded at 1 mM magnesium, based on the SAXS data. The data shown in Fig. 4f, along with the Kratky plots (4b) and the P(r) (4c) all clearly indicate that it is not. RNA folding does not have to be 2-state, and in fact most definitely is not. There are usually any number of compact intermediates. This is well documented in the literature, in both equilibrium and time-resolved studies of magnesium dependent RNA folding.

For reference, look at Mg²⁺-Dependent Compaction and Folding of Yeast tRNAPhe and the Catalytic Domain of the B. subtilis RNase P RNA Determined by Small-Angle X-ray Scattering by Fang et al. (Biochemistry, 39:11107-13, 2000). Figure 4 shows the Mg-dependent folding of the catalytic domain of the Bacillus subtilis RNase P RNA (not the same as the molecule studied here, but a good lesson in identifying when an RNA is folded in a magnesium series). The native state, meaning folded, occurs when the Rg reaches its lowest value, not at the midpoint of the curve. In fact, Feng et al. show the existence of an intermediate, partially folded state, which may be comparable to what is detected here.

I stand by my original comments: I am not convinced the molecule is in its folded state under (at least some of) the conditions of these studies. Unless they can make the case that this is the physiological form of the molecule (so the partially folded state is actually found in the cell), I cannot agree that conclusions reached at 1 mM magnesium are physiologically relevant, and these are central to the main point of the paper.

In their exchange with the Reviewers, the Authors have made two points that I find somewhat contradictory. For one of them they argue that surface immobilization of RNase P in the course of AFM measurements has no impact on its sampled space of conformations since SAXS, as an independent solution technique, fully corroborates the AFM findings. For the other, they argue that conformational variability of RNase P and its sensitivity to changes in Mg²⁺ concentration at near-physiological conditions are central to understanding its function.

I agree with the Authors' second point on flexibility being the central aspect of RNA's functional behavior at physiological free Mg²⁺ concentrations. Indeed, as shown in the Extended Data Table 1, the change in the Mg²⁺ concentration from 1mM to 2.5mM changes the fraction of the compact crystal structure-like state from 19% to 95%. At 1mM Mg²⁺ RNase P appears to exhibit the highest response to that variable, suggesting an ensemble of conformations. However, if we assume that to be true, the Authors' view that immobilization of a flexible RNA on a coated mica surface has no impact at all doesn't seem entirely plausible. In addition to the Mg²⁺ concentration, a pool of presumably transiently sampled and inter-converting conformations could be sensitive to a wide range of factors, such as temperature, RNA concentration, or the presence of a foreign surface they are forced to interact with, as is the case with AFM. In that respect, I don't find the Authors' argument regarding SAXS perfectly corroborating AFM entirely convincing. Strict equivalence between the hypothetical solution-state (SAXS) and the surface-immobilized (AFM) ensembles would have implied that the experimental SAXS data matches structure-modeled data averaged over the entire 158-member ensemble recovered by AFM, without any need for modification of the AFM ensemble weights. However, the Authors have clearly adjusted the ensemble member weights to fit the SAXS data. The Authors could have stated that they expect only the populations and not the conformations themselves to be affected by the change from the solution to the surface-associated state. However, a much stronger claim of complete equivalence is being made, and I find that harder to defend. The Authors are reporting that an optimally weighted ensemble of 3 conformations out of the AFM-derived 158 satisfactorily fits the data. This is plausible, in line with the Authors' prior results such as those reported in reference 23. The Authors then state on page 31: "classification based on quantitative partitioning using scattering intensity traits ... shows that the three classes are sufficient to describe the ensemble of 158 particles". In other words, the Authors are claiming that scattering intensity profiles predicted from the three best-fitting conformations are indistinguishable from those of the remaining 155 conformations. However, the evidence for that and the numerical criteria for considering equivalence are not presented. Even if we disregard the common appearance of discrepancies at increasing q in the scattering data predicted from two different structures, such claim would have to hold for the lowest- q data that define the gyration radii. The entire set of 158 AFM-derived models would need to have the distribution of their predicted radii of gyration tightly clustered around the 3 values corresponding to those of the 3 best-fitting conformers. On page 6 the Authors report standard deviations of the radii of gyration for the conformers belonging to the three classes as 2A, 2A, and 3A. Considering that these are standard deviations, the respective distributions are falling well outside of what could be considered an achievable precision of a SAXS-extracted gyration radius (0.5-1A). Therefore, it appears that many of these conformers should not have been ascribed to any of the (C1, C2, C3) classes, and may thus represent possible differences between the surface-immobilized and solution-state ensembles, which brings us back to the argument of the impact of AFM immobilization.

I suspect that the SAXS data at 1mM Mg²⁺ could be similarly fitted with a 2-component ensemble, as the weight of Class 3 is only 6%. I did not find the χ^2 value for the best-fitting N=2 ensemble, but I am estimating it to be close to $\chi^2=2.0$ based on the Figure 2. It is difficult to visualize the significance of that fit improvement relative to $\chi^2=1.7$ of the N=3 ensemble, or the precision with which this very minor component could be defined structurally. I also note that ED Fig 6 shows that perfect $\chi^2=1$ fits are only observed for Mg²⁺ at or above 2mM. Therefore, fit results at 1mM Mg²⁺ are likely associated with some residual data/model discrepancies and the improvement in the fit from $\chi^2=2$ to $\chi^2=1.7$ may not be the final detail in the overall picture.

It is clear from the Authors' data that something resembling the crystal structure becomes an increasing fraction of the solution ensemble with increasing Mg²⁺ concentration. The question is how confident can one be regarding the ensemble composition. Even if we reject the possibility of an impact of surface immobilization, the set of 158 AFM-derived conformations is unlikely to be exhaustive and SAXS data are known to be degenerate with respect to the 3D structure, with this effect even more pronounced with the relatively limited q_{max} of 0.35A⁻¹ and sizeable data noise at higher q . It is clear that SAXS data at 1mM Mg²⁺ point to the presence of conformations with the gyration radii in excess of 50A. I also find it plausible that the presence of a number of states differing from the "crystal structure-like" could be inferred from such data. I am less confident that these AFM-processed and SAXS-fitted classes are equivalent to actual structures present in solution, due to caveats of both SAXS and AFM, as noted above. The results of the study essentially suggest that the amount of information contained in the SAXS and AFM data is sufficient for derivation of a best-fitting solution with up to 3 "states" with each "state" representing, at a several nm resolution, a heterogeneous ensemble of potentially diverse conformations, subject to the q -range and the signal/noise profiles of the measured SAXS data. Therefore, I consider the "high resolution" part of the manuscript results primarily localized in the section titled "Conformational space of the RNase P RNA" to be somewhat speculative.

I note that the Authors' protocols do not appear to include any aspects of validation, which would be useful to detect over-interpretation of the data.

As a separate point, I was not able to find in the Methods section the RNA concentration(s) that SAXS data were recorded at and what constitutes the fitted data, and thus, how confident are the Authors that the SAXS-extracted gyration radius and the lowest- q data in general, are free from the concentration effects. This is an important point as determination of the less-

compact states would require high confidence in the lowest-q data.

Version 2:

Reviewer comments:

Referee #2

In the prior review cycle, I was somewhat disappointed with the authors' reluctance to address my initial comments, and pointed out 3 points that I wished the authors to address:

1. The possible merging of the methods and the more applied manuscript (this was only a recommendation).
2. The wrong use of the word 'motion'.
3. The multiplicity of conformations vs the multiplicity of adsorption orientations, and how they selected the 158 particles.

In response, the authors:

1. Let us know that the methods paper is now accepted and that they hold publication for back-to-back appearance with this paper. I have no problem with this.

2. The authors let me know that this was addressed in the former review cycle, meaning that they dismissed the comment. The authors say "A pool of discrete snapshot structures has been used to derive molecular motions in proteins^{1,2}, and we aren't the first to do so.". Do the authors not see that this statement is wrong? Snapshots of structure do not allow to derive motions. It doesn't really matter if the authors are not the first to do so or not. Readers with background in physics will disagree with it.

3. The authors let me know "We have addressed this question in the previous response. In short, we selected isolated particles based on image quality free of damages caused by the AFM probe, or image artifacts caused by instrument random instability." But they do not let me know how they discern between these experimental effects and conformational variations. Also, they let me know "It took about 24 million CPU hours (10,000 CPU for 100 days nonstop) to compute the structures of 158 conformers, not to mention the time spent optimizing the computational protocols.", which was not what I wanted to know, but hoped to learn if each conformer was an aggregate of many particles.

Overall, I do not find that I get into a fruitful scientific exchange with the authors in the review experience.

I congratulate the authors for the beautiful data and the ground-breaking achievement of extracting statistical ensemble distributions from AFM data, and regret their stubbornness to engage meaningfully into the review process and change/amend their work in the process beyond their original thoughts.

Referee #4

My main critique focused on the possibility of mica surface immobilization impacting the AFM-sampled ensemble of conformers. The authors cite references 25-30 as prior evidence for the lack of such effects. I was not able to locate those references in either the edited main text, where refs. 25-30 cover NMR and FRET, or the rebuttal letter, where references end at 26 and refs. 25-26 also have nothing to do with AFM. I thank the authors for supplementing these references with the link to the plain-language document for the Bruker-sponsored white paper. The linked brochure includes evidence of preservation of the major and minor grooves in DNA upon surface immobilization. The authors highlight that point in their response stating that the same interactions are responsible for formation of the fully base paired double helices as those driving the complete process of RNA folding. I agree that the hydrogen binding or base stacking interactions are universal. However, I would argue for differences between the energetics of a cooperatively formed Watson-Crick paired helices and structural elements containing just a handful of hydrogen bonds such as those that may be present in the inter-helical loops. Therefore, the fact that completely base-paired helices roughly retain their groove parameters upon immobilization, within the 10-15Å resolution of the AFM images, does not prove that the same preservation applies to weaker interactions that may be determining the overall architecture of flexible RNAs such as the one studied here.

The authors have modified the text stating the AFM ensemble members "represent the sampling of the RNA's conformational space at the time the molecules were immobilized on the mica surface" (p 4), covering "the nearly full if not complete sampling of the conformational space" (the top of p 7) and representing "a conformational sampling of the solution ensemble at 1 mM Mg²⁺" (the beginning of the Discussion section on p 14). While interaction of the RNA with the mica surface is likely non—instantaneous and may include some conformation reorganization, the authors have provided new evidence to support their claims based on the PCA analysis, the comparison to cryo-EM data at 10mM Mg²⁺, and the SAXS data. The results of both the PCA analysis and the cryo-EM data comparison appear compelling. The details on the SAXS data analysis included in the rebuttal letter also corroborate the authors' claims as the appropriately weighted AFM ensemble provides an excellent for to the SAXS data. The caveat here is the relatively low resolution of the scattering data

and the relatively high noise at high q . However, these issues are nearly universal in the field and these results are as good as can be expected.

In summary, the authors have provided a sizeable amount of new details and analysis to more thoroughly back up the main claims of the paper, which are indeed quite remarkable. I feel that the manuscript can be accepted for publication in its edited form.

Referee #1:

In this study, the authors utilise solution atomic force microscopy (AFM) in conjunction with deep neural networks to explore the diverse conformations of RNase P RNA, revealing a stable core surrounded by dynamic peripheral elements. Additionally, the study demonstrates that heightened Mg²⁺ concentration fosters RNA compaction and enhances enzymatic activity, likely attributable to reduced conformational variability. Crucially, this research establishes a nexus between spatial flexibility and sequence conservation, illuminating the intrinsic functional roles encoded within the RNA's primary sequence. This insight significantly advances our comprehension of the enzymatic precision and substrate diversity of RNase P RNA.

Nevertheless, the deep learning approach in this study, HORNET, presented in this article originates from the authors' previous work, "Determining Structures of Individual RNA Conformers Using Atomic Force Microscopy Images and Deep Neural Networks" (10.21203/rs.3.rs-2798658/v1). This earlier study describes the application of HORNET to map the three-dimensional structures of various large RNAs, including RNase P RNA. In the current article, the HORNET approach is utilised to analyse the diverse conformations of RNase P RNA, along with their biological implications. Therefore, this paper can be considered as complementary to the authors' previous work, further elaborating the conformation of RNase P using the HORNET method, providing insights into the complexity of its structure and its functional implications. Therefore, I would suggest that synthesising these two articles into one would significantly benefit both the AFM and RNA structure communities, enhancing the overall impact and relevance of the research.

Response:

We thank the reviewer for recognizing the fundamental importance of our study.

About merging the two manuscripts into one, we appreciate the suggestion aimed at enhancing the overall impact. The HORNET manuscript tackles one of the greatest challenges in RNA structural biology. The significance, novelties, amount of work and complexity of the results indeed require a full paper to disseminate. As a result, both of the manuscripts would be compromised, defeating the purposes of enhancing impact if merged into one.

The manuscript is commendably written, with its language being both articulate and accessible. The authors have structured their arguments in a logical and coherent manner, making the complex subject matter readily comprehensible to readers. There are also a few minor errors like figure notes and codes that I've listed below.

We thank the reviewer for the comments.

Minor comment:

1. The right plot in Ext. Data Fig. 1 totals 161 particles, which does not match the 158 described in the figure notes. A similar quantity problem is found in Ext. Data Figs. 2,3 and 4, which contain 165 images but are labeled 158 in figure legend. Also, Ext. Data Fig. 5 has 154 violin plots (158 in figure legend). Please check.

Three out of 161 particles were not converged within a reasonable computing time for reasons that could be attributed to partial unfolding based on their elongated shapes. We have added a sentence to clarify it in **Methods**. In the data in Ext. Data Figs. 2, 3 and 4, some of the particles were separated into two groups, one of them requiring longer time of dynamic fitting to converge. They were labeled as X_2 in the original submission. In the current submission, to avoid confusion, a new plot combining all results in to one plot per particle.

We thank the reviewer for catching the missing 4 violin plots in Ext. Data Fig5 and have added the missing plots.

2. Ext. Data Fig. 11 contains 3 subfigures and figure legend mismatches. There are three subfigures a,b,c but only a and b in the figure legend.

We thank the reviewer for pointing them out. We have corrected the problems.

3. I checked the code section. The HORNET section is out of the scope of this review. Other scripts such as PCA and SAXS data processing scripts, although there is introduction to their main functions, but due to the lack of corresponding test files so can not be tested through.

We thank the reviewer for the comments. We have included the relevant codes and files for the reviewers and users to test (https://home.ccr.cancer.gov/csb/pnai/data/conformational_space/Conf_space_RNasePRNA/scripts_analysis/).

Referee #1 (Remarks on code availability):

I have reviewed the code section. The HORNET component falls outside the purview of this review. Regarding other scripts, such as those for PCA and SAXS data processing, while their main functions are introduced, the absence of corresponding test files precludes thorough testing. I recommend that the authors provide a more comprehensive and user-friendly computational package, including test files, to facilitate use and validation by others in the field.

We thank the reviewer for the comments. We have included the relevant codes and files for the reviewers and users to test (see the link in the response to the last comments). We also thank the reviewer for the detailed comments/suggestions.

Referee #2:

In "The conformational space of RNA in solution" by Yun-Tzai Lee, Maximilia F. S. Degenhardt, Ilias Skeparnias, Hermann F. Degenhardt, Yuba R. Bhandari, Ping Yu, Jason R. Stagno, Lixin Fan, Jinwei Zhang, Yun-Xing Wang, work is described that uncovers the structural flexibility of RNase P RNA.

The work is quite technical - in a good way - as it opens up a novel avenue of research that enables RNA structural biology. The main enabling technology is high-resolution atomic force microscopy (AFM) combined with machine learning algorithms, principal component analysis, SAXS, and sequence analysis.

In brief, RNA structural biology remained challenging as current structural biology methods (X-ray diffraction of 3D crystals) only provided snapshots, which, especially in the case of RNA-based enzymes with high flexibility and functional promiscuity, may miss important aspects, namely how conformational heterogeneity is an inherent and necessary component of this class of molecules.

The authors present high-resolution AFM images from which they extract molecular observations that they then compare to structures, and model the structural variance using computational methods. The author make a compelling case that X-ray crystallography can only derive a single conformer of the molecule under investigation, which in addition might be biased by the crystal contacts and crystallization conditions. So far, cryo-EM has not been very much used to analyze RNA enzyme structures, likely due to their small size. In contrast, solution AFM offers itself as an alternative to study structural variability in liquid at ambient temperature and pressure and close-to-physiological conditions.

They find that RNase P RNA has a structural core that is quite constant and features peripheral elements that are highly flexible and undergo large scale fluctuations. Interestingly, conserved regions are comprised in the firm core as well as in the flexible parts, interpreted that the primary sequence encodes flexibility too, which I find an interesting aspect to think deeply about. While this reviewer thinks that the authors should use a more specific title - as we do not yet know how general their findings are ("The conformational space of RNA in solution" is clearly a much too general title), I do appreciate the technical quality and focus of the work, as I appreciate research breakthroughs that may open fields rather than reports on specific findings on a specific molecule. I am in principle excited about the possible impact and reach of this work, but I have major concerns regarding the orientating vs conformational variability, as I detail below.

We thank the reviewer for recognizing the importance and complexity of our work. We agree with the reviewer and have changed the title with a narrower descriptor "*The conformational space of RNase P RNA in solution*". We also thank the reviewer for the detailed comments/suggestions below.

Major:

AFM is a surface technique. As such the molecules that are surface adsorbed can adopt an almost infinite number of possible orientations on the surface.

1) This implies the first major question: how does surface adsorption impact structure, and how can we make sure we interpret the surface structures correctly towards 3D structures?

The reviewer raised two questions and we address these two questions separately as follows.

1. *how does surface adsorption impact structure?*

There has been abundant literature discussing the effects of various immobilization methods on mica surfaces for nucleic acids. One of the latest discussions of the subject can be found in a review by Lipiec et al., (Molecules, 2021; 26(21):6476 and references cited therein). It is known that immobilization using various divalent ions may affect the structures and conformation of nucleic acids on a mica surface or in solution. The immobilization using silane agents is much milder and do not alter the physiological conformation (Japaridze, et al., Macromolecules, 2016; 49:643–652). However, ultimate proof will have to come from alternative methods that can directly visualize the individual conformation of known structure (as opposed to a structure obtained via averaged signals from methods, such as NMR, crystallography, or cryo-EM). Given there is no such a proven direct-visualization method other than AFM itself, the closest orthogonal method is to examine and compare the synthesized ensemble behaviors, such as measurements using SAXS or ITC, based on structures of individual conformers and the population tallies from AFM direct visualization. This is what we have demonstrated in our previous publication (Ding, et al., Nat. Comm., 2023), where we show the synthesized behaviors of the cobalamine RNA corroborate the direct observation of very heterogeneous conformation populations. We have shown similar corroborative results in the current manuscript. Should the immobilization process significantly affect the topological structures, we would expect inconsistency between orthogonal data sets. Importantly, those heterogeneous conformations aren't induced by absorption on mica surfaces, because we have observed similar heterogeneity using cryo-EM where no such process is involved (Ding et al., NAR, 2023), although for the very reason of requiring signal-averaging over a large number of particles of "identical" conformations, high-resolution structures of each individual conformers could not be obtained by the cryo-EM approach, the ultimate limitation of all current technologies relying on signal-averaging.

2. *how can we make sure we interpret the surface structures correctly towards 3D structures?*

This question, which is one of the key functions of HORNET, has been thoroughly addressed in the supporting manuscript, which has been provided.

2) The second major question concerns the in principle infinite number of orientations. The identical molecular conformation could in principle result in a huge number of topographies dependent on observation orientation. In this context, the 158 conformers, which I think the authors refer to as major observation classes (they have more observations, Ext. Data Fig. 1) could thus report conformational variability or orientational variability. It is unclear to me how the authors unambiguously tackle this problem without a priori knowledge of the conformation. This is a major drawback, and I suspect the authors might need more data. It remains unclear how the authors separate orientational from conformational variability and how very different observation orientations are pooled into the same conformer using the described method, vs the coincidental emergence of similar observations from different conformations in different orientations.

The general shapes of RNA molecules, unlike globular proteins, are usually highly asymmetric with one or a few sides relatively "flat", as we have shown in our previous publications and in the supporting manuscript. As a result, they have preferential orientations on a mica surface where the orientations are highly anisotropic and the preferred orientations are those with maximized or near maximized contact surfaces. Because of the preferential orientations, the total number of orientations is not infinite, but quite the opposite, very limited. This behavior differs from the number of possible infinite orientations of particles that molecules could adopt on a cryo-EM grid. Furthermore, isotropic

orientations of particles are necessary for 3D reconstruction in the cryo-EM approach but unnecessary for HORNET. A detailed description of how HORNET recapitulates 3D topological structures from AFM images is fully disseminated in the supporting manuscript. Nevertheless, we provide a sample calculation here to illustrate HORNET can faithfully recapitulate the 3D topological structure with accuracy (uncertainty) around 5 Å as we stated in the HORNET manuscript regardless of the orientations. In the case of RNase P RNA, the accuracy (uncertainty) is mostly around 4 Å (**Extended Data Fig. 2** and **Supplementary Table 1**). In the following illustration, we show the recapitulation of 3D topological structures of the same conformer in five different orientations with RMSDs of the recapitulated structures relative to the GT structure under 5 Å (**Figure 1**). Note that the same initial structural model, which differs by ~20 Å from the GT structure in terms of RMSD, was used for all calculations. The accuracy could be improved if an extensively long calculation is performed.

Figure 1. Recapitulating 3D topological structures from particles of the same conformers but in five different orientations. **a.** The GT and initial structures differ from each other by 20 Å in terms of RMSD. **b.** Five different orientations. **c.** Counts vs. RMSDs of the full trajectory of the five orientations from dynamic fittings. **d.** The RMSDs of top 1 structures relative to the GT structure. The top 1 structures are identified by HORNET. **e.** Pair-wise RMSD comparison matrix among five structures recapitulated from the five orientations. **f.** The superimposed five structures for comparison.

The second and related question is whether the differences among the 158 conformers determined by HORNET are indeed greater than the differences between different orientations of the same conformer. To answer this question, we performed a pairwise comparison among all 158 conformers. With the exception of a single pair (RMSD=6.0 Å), the structural comparison among conformers shows an RMSD range of 7.8 to 43.1 Å, with a mean RMSD of 24.4 +/- 2.9 Å. Given the limited resolution of 5 Å in general and 4 Å in the RNase P RNA case, we recognize HORNET would not be able to distinguish the difference if the difference among conformers is less than the uncertainty.

Figure 2. The full matrix of the pairwise RMSD comparison of all 158 conformers. Only 40 conformer labels are shown because of the space limit in the graph. The smallest RMSD between two conformers is ~6 Å. The vertical color scale on the right shows RMSD in Å.

Both Figures 1&2 are added to the **Extended Data** figures.

Minor Comments:

Page 1, Line 1: Change the title to a more specific one. Researchers will understand the general aspect of your efforts.

We thank the reviewer for the suggestion and changed it to a more specific title, “*The conformational space of RNase P RNA in solution*”.

Page 3, Line 4: Name what you call single molecule approaches. I can only assume that the authors refer to optical methods. It is however a pity, because the authors themselves do single molecule research, and the (in my view erroneous) dominant use of "single molecule" for certain techniques is not a positive tendency for all other single molecule research.

We aren't totally sure what the reviewer meant here. The definition and use of “single molecule” are debatable. We see the use of the term very loosely and liberally, which could mean “single molecule” behaviors derived from ensemble measurements aided with computation techniques, or from direct visualization of individual molecules. Strictly speaking in our view, the cryo-EM is not a single molecule approach because it uses signal averaging over a large number of particles, all of which may adopt a more or less the same conformation. However, we're in no position to set a clear definition and we believe the structural biology field will itself eventually come to a consensus on the definition. Thus, in the cited references of “single molecule” approaches, they included both cases.

Page 3, Line 18: The sentence is not understandable for anyone who does not work on precisely that problem.

We will keep the sentence because of its relevance to cancer, the clear importance of which needs no further elaboration in this context.

Page 4, Line 23: Not only "conformational differences"--there are also conformational similarities. Indeed, Fig. 1 panels c,d actually reveal notable similarities.

We thank the reviewer's suggestion and added “or similarities”.

Page 6, Line 2: While molecules in crystals can explore some conformational space, this is only very limited (comprised in the B-factor) but only one structure is the result. Thus the phrasing "sample a much wider" is confusing as the other technique (crystallography) literally does not sample at all.

The reviewer is correct here that crystallographers usually conclude with only one structure even though the B-factors suggest otherwise. What we meant here is comparing what really happens in a confined lattice space compared to those in free solution and under 1 mM Mg²⁺, regardless if one or a few crystal structures are used to best fit the electron densities.

Page 6, Lines 11, 17, etc, and Fig. 2 caption: The writing is very confusing with regard to the 158 conformers. I did not understand how the authors came to that number; clearly they have many more observations (Ext. Data Fig. 1). I assume they pool observations in conformer classes. In addition there might be identical conformers but different views. Provide more details about observations, orientations, conformations, conformers, etc (see major concern).

We've indicated in the manuscript that we selected those particles that are well resolved, free of overlapping and with a good image quality. Regarding particles of the same conformation but different

orientations, we have addressed the question in response to one of the previous comments. Specifically, they are the molecules of the 158 different conformers, which could be grouped into three classes, C1, C2 and C3 (Fig. 2 and the relevant text).

Page 10, Line 29 and following: Since the manuscript is quite technical, I would appreciate more details regarding the PCA in the main text.

We added a description to the **Methods** section, in addition to including all in-house software used for the analysis (see the link below):

https://home.ccr.cancer.gov/csb/pnai/data/conformational_space/Conf_space_RNasePRNA/scripts_analysis/

Page 11, Line 11: While "motions of grass on the sea floor" is poetical, I would prefer a more scientific description that does not appeal to each individuals vacation background for a better understanding of the results.

We try to use the metaphor to describe the motions of RNA peripheral structural elements in solution in a way a layman's reader could understand without further explanation. We have deleted the language.

Referee #3:

This paper describes the application of a new method for determining the structures of RNA in an ensemble, by interpreting solution AFM data using a new deep neural network technique that is detailed in a companion manuscript. Here, the focus is on determining the ‘functional’ motions of a particular RNA enzyme. The level of detail, specifically the information about the conformational degrees of freedom of the RNA, is quite impressive and represents a needed advance to the field. However, I have a number of questions/concerns about the interpretation of the experiment. Given that the methodology will be published elsewhere, it seems that the ‘science’ story is central to this work, so author responses will be critical in assessing the impact of the scientific story.

We thank the reviewer for her/his appreciation of the work. In the following, we will address the raised questions.

An alternative interpretation of the experimental results are as follows. What if they are measuring the folding of the RNA, not its functional motions (which I interpret to mean the motions of the folded enzyme that catalyze reactions). Fig. 4 shows the Mg-dependence of the conformations (from SAXS data). Fig. 4d closely resembles the equilibrium curves from a folding assay. In this case, C2 represents unfolded conformations (meaning conformations which are extended because the electrostatic compensation by the ions is not complete). Although functional folding motions are in their own right interesting, the assumption in the paper is that the molecule is in its physiologically active state at 1 mM Mg. What if it is in a superposition of unfolded and folded states, so it can partially carry out its function? It is impossible to know if the molecule is physiologically folded from these measurements, despite the statement that 1 mM Mg is physiological. Yes, that is true, but the other salt in the buffer is below physiological levels (100 mM NaCl, for example, not to mention that other ions contribute to overall ionic strength), so it is possible that 1 mM Mg is just not enough to fold the RNA. If the motions do indeed represent essential conformational dynamics for function, why does the enzyme become more efficient above the point where it is fully folded (5 mM Mg)?

Regarding the process of folding/unfolding vs. functional motion. We appreciate the reviewer for bringing up this point, which is understandable. The reviewer’s comments could be boiled down to whether one could tell with certainty the difference between folded vs unfolded RNase P RNA molecules. In the past, the folding/unfolding processes have been studied using conventional arrays of biophysical methods, all of which are indirect and the interpretation of data could be circumstantial and ambiguous. AFM does provide a direct visualization, where unfolded molecules can be visible and identified and there is little ambiguity here. In the figure below, unfolded or partially folded RNase P RNA molecules can clearly be identified and distinguished from folded molecules (we have revised the manuscript and added this image to **Extended Data Fig. 1b**). This sample was prepared after denaturing but before proper folding. In contrast, we did not observe any significant populations of unfolded RNase P RNA molecules under our specified condition. It is noteworthy that AFM is very useful in monitoring sample prep (see one of our recent publications on the subject: *Ding. J., et al., RNA heterogeneity visualized under AFM in Integrative Structural Biology, in Integrative Structural Biology, pp388-269, Royal Soc. Chem; 2023*). We do take careful steps and develop RNA sample preparation protocols aided by AFM imaging. As an example to illustrate how carefully we fold RNA samples, the reviewer may look into an elaborate scheme that was developed for folding a more challenging RNA (**Supplementary Figure 1** in *Ding et al., Nat. Comms., 2023*).

Unfolded (UF_n) vs. folded (N_n) RNase P RNA visualized under AFM.

Regarding the effect of monovalent ions, we also compared the activities and SAXS profiles of the RNase P RNA in solutions containing 1 mM Mg²⁺ and either 100 and 150 mM NaCl and observed little difference (see the figures below) by changing the NaCl concentrations, in contrast to the marked effect of Mg²⁺ (Fig. 4). Thus these results together strongly support the well-known notion that Mg²⁺ is critical for the folding and activity of this RNA. The Kratky plots simply suggest the RNA is folded and flexible in both concentrations. One last point, although our solution conditions are the most relevant to physiological conditions, especially the physiologically relevant Mg²⁺ concentration,

compared to those used in crystallography and cryo-EM for RNA, we do not claim our solution condition as a whole is THE physiological condition.

Regarding the RNA's behavior at 5 mM Mg^{2+} , we believe that the RNA is driven to and adopt a narrower and more compact conformational space under 5 mM Mg^{2+} and becomes more crystal-structure-like, as shown in **Fig. 4e**. Furthermore, it is known that there is a low-affinity site for a Mg^{2+} ion that directly involves in the catalysis. We suspect the compaction of the conformational space, which could be driven by non-physiologically high Mg^{2+} concentrations such as 5 mM, may facilitate the binding of the catalytic Mg^{2+} . Protein RnpA, the protein subunit of this RNase P, is more effective by 5-fold in facilitating the binding of the low-affinity catalytic Mg^{2+} binding without alteration of the core structure (where we refer to the invariant structural core in our manuscript) (Kurz, J. C. & Fierke, C. A. *Biochemistry* 36 41, 9545-9558, 2002). We and others believe that RnpA binding limits the sampling of conformational space by the RNA component.

Furthermore, key experimental details are not given in the paper. What is the [RNA] for the experiments, is the Mg just added or are the samples dialyzed (in other words, is the 1 mM Mg a 'free' or 'total' ion concentration)?

The details about the sample preparations have been provided in the **Methods** section. All samples were freshly FPLC-purified in buffers containing 100 mM NaCl and indicated Mg^{2+} concentrations before experiments.

*It is also concerning that, given their arguments about the importance of conformational variations, a *single* conformer is sufficient to represent C1, C2 and C3. This seems to contradict the main point of the manuscript, that multiple, distinct conformations are present. Additionally, from the information presented in the paper it is unclear if all of the atoms are present in the conformers. For example, C1 should be close to the crystal structure (I assume this by comparing data from Ext. Data Fig. 6 to the coefficients of Fig. 2d; at 10 mM Mg, the chi-square value for the fit to the crystal structure looks acceptable, and it looks like the population is nearly 100% C1.) This strikes me as odd because the crystal structure is *missing* multiple nucleotides, so how can the profiles be so similar? Are all of the nucleotides included in C1? This is not at all clear from what is presented in the paper and the pdb files for the C1, C2 and C3 structures do not appear to be available.*

We do not suggest that a single conformer is sufficient to represent the ensemble behavior shown by experimental SAXS data. Rather, we believe that the three conformers are representative of each of the three distinct classes, as one knows that SAXS describes only ensemble behavior not a single particle. We added a figure and a table (**Fig. 2e; ED Table 1**) showing more conformers give better fits and plateaued after a certain number of permutative interactions. We also added description in **Methods** about ensemble fitting. We have reworded the language to avoid any confusion. The computational script performing the fitting is also provided at https://home.ccr.cancer.gov/csb/pnai/data/conformational_space/Conf_space_RNasePRNA/scripts_analysis/

The missing residues in the crystal structure (2A64) were modeled as described in the **Methods** section (see sub-section *Recapitulation and accuracy of topological structure* in **Methods**). Thus, all atoms of 417 residues are present in the structural models of the full-length RNase P RNA. In the meantime, we have reworded the section to avoid confusion.

My last major point regards the use of crysol for predicting the SAXS profiles. Although crysol works reasonably well for proteins, it is not the best to use for RNA as it does not correctly capture the solvent effects. This, together with the use of only three states, diminishes enthusiasm for the fitting procedure. Please see Bernetti et al. (Eur. Phys. J. B (2021) 94:180) for a discussion about the benefits of using different packages to simulate SAXS profiles.

The reviewer raised an interesting issue. The contribution to scattering profiles by hydration layers is an active sub-area of investigation in small-angle scattering. We re-examined relevant literature. Here we provide our views on this issue. There have been a large number of computation programs developed in an attempt to accurately address the hydration issue, including Crysol/Crysol3, AquaSAXS, Zernike, Fast-SAXS-pro, hySAS, and FoXS, in addition to the programs that the reviewer mentioned. All those programs appear to perform well in the cases that were demonstrated. One important factor for us to consider is what the impact of hydration layers is in terms of R_g values modeled using various programs. Bernetti and Bussi (Bernetti and Bussi, *E. Phys. J. B.*, 2021) show R_g varies from 16.1 to 16.8 Å for the 57-nt GAC RNA. Note both WAXSiS and Crysol return with the same R_g value of 16.7 Å. The relatively small difference is less than one-tenth of the range among the three classes of RNase P RNA conformers in our study, with R_g ranging from 46 Å to 58 Å (Fig. 2d). Moreover, Bernetti et al. showed that, in terms of the full SAXS profile up to 0.35 \AA^{-1} , the distance (a parameter related to χ^2) between that of Crysol and that under the experimental Mg^{2+} (1 mM) (Bernetti et al., *NAR*, 2021) is ~ 0.2 , similar to that by other programs (Bernetti and Bussi, *E. Phys. J. B.*, 2021), suggesting a similar performance under 1 mM Mg^{2+} . Although one might obtain somewhat slightly different percentages of conformer populations based on the back-calculated SAXS profiles with different approximations of hydration layers, we do not expect significant changes in those percentages, especially when the differences among those methods are negligible compared to R_g differences among the three classes. One might agree that all conformers are subject to similar hydration layers under the same solution condition thus the volume fractions of all conformers would remain approximately the same. As a result, we believe that it is reasonable to use an implicit hydration layer when approximating the ensemble behavior of widely heterogeneous conformers that sample a broad conformational space such as RNase P RNA in our study, as previously suggested (Bernetti and Bussi, *E. Phys. J. B.*, 2021).

Nevertheless, we used WAXSiS to perform the analysis on all 158 conformers using an explicit hydration layer. It is noteworthy to mention, however, that using an MD simulation to generate an explicit hydration layer, a procedure required by all programs, alters conformation and thus cannot be used for the validation purpose in our study. As an example, using WAXSiS software and the suggested 27 ps simulation, we generated a 7 Å standard hydration layer. Even such a short simulation resulted in significant structural perturbations with RMSD of about 13 Å from the HORNET-determined original structure (see the figure below). This change is substantially larger than those observed for GAC RNA (RMSD ranging from ~ 1.5 to 3 Å) (Bernetti et al., *NAR*, 2021). Most likely, this is because the initial structure used in the GAC RNA case was the crystal structure determined at 90 mM Mg^{2+} (in which 10 ordered Mg^{2+} ions were modeled). In contrast, the 158 RNase P RNA conformers were determined under 1 mM Mg^{2+} , and the starting structures used for explicit solvation in each case contained no ions. Therefore, we suspect that the minimization steps of MD simulation modeled based on the GAC RNA case (Bernetti et al., *NAR*, 2021, Fig. 3) would result in far greater structural perturbation in our structure in this study. The crystal structure of the RNase P RNA (PDB: 2A64), which was recorded at 70 mM Mg^{2+} , contains no modeled Mg^{2+} ions thus we could not model them explicitly. In summary, as the use of SAXS in this study is for an orthogonal method to validate the structures determined by HORNET, such MD-altered structures generated upon explicit solvation cannot be used for our validation.

Other less critical points

It is also disappointing that the references to conformational dynamics mostly reference work done many years ago. The authors seem to have missed a small, but significant amount of literature modeling SAXS data with multiple conformers (for two examples, see Bernetti et al., NAR, 2021 gkab459, and He et al., NAR, 2023 gkad809).

Given the limits on the number of references allowed, we mostly cited the references directly related to this study. But we agree with the reviewer and cited suggested references in the **Methods**.

I find the repeated emphasis on their measurements of the 'motions' to be overstated. If you are measuring motions, you MUST have a time axis or a time scale. What is measured are 'ranges of motion', or better yet, the full structural ensemble. Interpolation between the different structures can suggest the pathways for motion (as they do), but they do not directly report on motion itself. In fact, they have it exactly right in the abstract; they are mapping the conformational space, NOT determining the motions.

We refer to the motion in this study as spatial motion. The temporal aspect of motion is not within the scope of this study. We have revised the manuscript and added a statement "Throughout the text, we refer to motion as spatial motion without a temporal aspect".

Referee #1:

I appreciate the authors' feedback on my comments. I have no further comments to add.

Referee #1 (Remarks on code availability):

From the computational standpoint, I have verified that the authors have diligently updated the corresponding test files and documentation in response to the suggestions provided, ensuring the code's usability and reproducibility.

We appreciate the reviewer's meticulous review and verifying some important details of our work.

Referee #2:

The manuscript, "The conformational space of RNase P RNA in solution", by Lee et al. is the revised version of a former paper entitled "The conformational space of RNA in solution". I have been quite positive about the paper in my initial assessment, and I still am, but I do not think that the paper was improved in the revision cycle.

The authors seemed reluctant to consider the referees' comments, where it mattered most.

1. I concur with referee #1 that the separation of method and biological finding in two individual manuscripts is somewhat unfortunate. Clearly, given that the authors have the two manuscripts in parallel in consideration, indicates that referee #1 is right that they are somewhat interdependent. At least to the point that this manuscript is difficult to follow and appreciate without reading entirely the methods paper, while the methods paper contains to a minor extent some aspects presented here. The authors argue that space and authorship warrant for two independent manuscripts, and I understand them. On the other hand, they might run the risk that readers might turn away from these findings as they may not find the entire information that leads from conceptualization over analysis to result. This referee thinks that the modern publishing format with extended data figures and co-first authorship would allow for such a combined manuscript, and that such a combined paper would have stronger impact than the two manuscripts alone, I can only recommend consideration.

We understand the reviewer's concern. At the time this response is being written, the other manuscript has been accepted for publication but we are holding it till the review of the current manuscript is complete, hoping for back-to-back publications. This arrangement may address the reviewer's concern.

2. I concur with referee #3 that the use of the word 'motion' is inaccurate, actually wrong. Even if the authors now define "motion as spatial motion without a temporal aspect", the word just doesn't match the findings. The findings have no information about motion, neither temporal nor directional, nor correlated motions, etc. The authors have plenty of words at hand like conformational space, conformational variability, etc, that are not words that represent an action. No action, no dynamics is reported here.

This issue was addressed in response to the suggestion made by Reviewer 3, who did not object to our revisions and response. A pool of discrete snapshot structures has been used to derive molecular motions in proteins^{1,2}, and we aren't the first to do so. What might be underappreciated here is the large number of conformational samplings (snapshots) that cover both ends of the conformational spectrum and the space in between. We thank the reviewer for acknowledging the usefulness of our analytical approach in the review comments.

3. My own major criticism concerned the multiplicity of conformations and the theoretical multiplicity of adsorption orientations, and that multiple conformations in different

orientations could potentially have similar topographies, while the same conformation in multiple orientations could be represented by multiple different topographies. The authors answer that the RNA molecules, unlike proteins, have one or a few flat surfaces and therefore their assumption is that they are looking at conformational variability of molecules that are all adsorbed on the surface in very similar fashion. Maybe. I tend to believe the authors but think this is a significant potential drawback. The authors do not mention this in the manuscript, while I think the readers should be made aware of this question. As the methods of cryo-EM advance the concepts of orientation and conformational variability will become only more common and the authors have an interest to detail precisely what they think and analyze. Also, I commented about the somewhat confusing use of particles, observations, orientations, conformations, conformers, etc. Initially I thought that there were 158 classes (conformations, conformers) and that each class contained tens of particles (molecules, observations). From the explanations in revision, I seem to understand that there are only 158 particles, and each particle is an individual conformation. This confuses me a lot. Each of the images in Figure 1a,b alone has more than 100 particles. I assume the authors have collected many such images (how many, I could not find in the Methods). These images look great. Most, >90% of the particles are nicely isolated and well resolved, so how do the authors only come up with 158 particles? Do I understand this correctly? If yes, how were they selected, using what criteria? In Fig. 2, the authors show a gallery of the structural models, and then gather them in three classes (f) or conformers (c). Which models in (a) are in class/conformer C1, C2 or C3, and based on what are they grouped (I understand that these three classes are enough to precisely fit the SAXS curve)?

We thank you the reviewer for bringing up this question, and we apologize for not fully understanding the concerns behind the question in the last round of the review. The reviewer's concerns could at least partially be attributed to the lack of clarity and detailed dissemination of information. The reviewer's concerns can be summarized into three points.

1. How were the 158 conformers/particles selected and why not more since all images look great?

We have addressed this question in the previous response. In short, we selected isolated particles based on image quality free of damages caused by the AFM probe, or image artifacts caused by instrument random instability. We agree with the reviewer that we could have picked even more particles. However, our goal was to collect enough particles that were statistically significant to compute the conformational space and illustrate the conformational heterogeneity. It took about 24 million CPU hours (10,000 CPU for 100 days nonstop) to compute the structures of 158 conformers, not to mention the time spent optimizing the computational protocols. Sampling more particles, although feasible, is neither necessary nor computationally practical. This concern is also related to the one in the next section.

2. Are the 158 conformers representative of the full conformational space?

We do not claim that the pool of the 158 conformers fully encompasses or is equal to all possible conformations in solution. Instead, the pool represents 158 samplings, which already far exceed what is needed for PCA analysis, as it has been well demonstrated in studies of protein conformational space^{1,2}. We would like to point out, if we may, the fact that determining topological structures of 158 conformers in solution using our novel method is in itself a significant advance, and this fact appears to be under-appreciated in the review. Since this pool covers the conformational spectrum from very compact to very extended conformers, with a large number of discrete conformers in between, the PCA analysis results in the mapping of motion trajectories, where many conformers, observed or not by AFM, fall into. These trajectories are derived in the form of motion direction and amplitude. The proposal of sampling an “exhaustive” larger pool is neither necessary nor computationally practical. Moreover, currently, there isn’t any experimental or theoretical approach to solve the RNA conformational space question and our study represents a very significant advance in the field.

3. How were they classified into three classes?

The classification of the particles based on SAXS data is described in detail in the SAXS analysis section in **Methods**, as well as in the main text (**pp 5-6 and pp 9-10 in the manuscript**). Briefly, the synthesized SAXS intensity profile is calculated for all 158 particles; next, all these profiles are used with no pre-assumption to optimize the volume fractions of each conformer to best fit the experimental SAXS data using a computational script. The more conformers used, the better the fit to the experimental SAXS data (**Fig. 2e in the manuscript**). In addition, we found that all conformers could be classified into three classes based on the distance between the substrate specificity and catalytic domains (**Fig.2f in the manuscript**). This distance defines the relative positioning/orientation of the S- and C-domains, and is key to the promiscuity of RNase P towards various substrates differing in size and sequence. In the revised manuscript, we have also applied PCA analysis³ and the commonly used EOM method⁴ to further illustrate and examine the 158 conformer pool. Both analyses result in the three classes, even though they use different sets of criteria. For more detailed discussion, please see the response to Reviewer #4’s comments.

In summary, I still think this is an interesting paper with high quality primary data and an interesting analysis method/concept and outcome. However, while the revision tackled the details, it did not resolve some of the major questions.

We have provided all the necessary information in the revision and in the response to address the reviewer’s concerns.

Minor:

82: I still do not understand what conformational space molecules packed in a crystal lattice sample.

We believe that our revision based on the reviewer’s suggestion is clear to scientists as well as layman readers. Macromolecular crystals can have solvent contents ranging

from 30–80%, averaging ~50%, with RNA crystals trending toward the higher end of the spectrum (>60%). Thus, crystal lattice contacts in RNA crystals tend to be relatively weak and few in number, which is why they tend to diffract poorly. Conformational heterogeneity (disorder) within a given lattice is the exact reason why some regions of the molecule show little or no electron density. Therefore, albeit limited by lattice constraints, these molecules do exhibit some level of conformational space within the crystal, which can range from small loop regions to entire domains, such as the S-domain of the RNase P RNA, which was not observable in the crystal structure (PDB: 2A64).

50: When I commented that the sentence was not understandable for anyone who does not work on precisely that problem, I didn't mean for the authors to delete it or tell me that they keep the sentence. As an intermediate, the authors could rephrase the sentence in a way that readers from slightly outside the field could also understand it.

In response to the reviewer's comment in the last round of revision, the authors have removed the original metaphor and conveyed the conformational flexibility of various structural elements whenever possible throughout the manuscript without the need to include the sentence.

Referee3:

Unfortunately, I do not concur with the authors' response to my question about whether or not the RNA is fully folded at 1 mM magnesium, based on the SAXS data. The data shown in Fig. 4f, along with the Kratky plots (4b) and the $P(r)$ (4c) all clearly indicate that it is not. RNA folding does not have to be 2-state, and in fact most definitely is not. There are usually any number of compact intermediates. This is well documented in the literature, in both equilibrium and time-resolved studies of magnesium-dependent RNA folding.

*For reference, look at Mg^{2+} -Dependent Compaction and Folding of Yeast $tRNA^{Phe}$ and the Catalytic Domain of the *B. subtilis* RNase P RNA Determined by Small-Angle X-ray Scattering by Fang et al. (Biochemistry, 39:11107-13, 2000). Figure 4 shows the Mg^{2+} -dependent folding of the catalytic domain of the *Bacillus subtilis* RNase P RNA (not the same as the molecule studied here, but a good lesson in identifying when an RNA is folded in a magnesium series). The **native state**, meaning folded, occurs when the R_g reaches its lowest value, not at the midpoint of the curve. In fact, Fang et al. show the existence of an intermediate, partially folded state, which may be comparable to what is detected here.*

I stand by my original comments: I am not convinced the molecule is in its folded state under (at least some of) the conditions of these studies. Unless they can make the case that this is the physiological form of the molecule (so the partially folded state is actually found in the cell), I cannot agree that conclusions reached at 1 mM magnesium are physiologically relevant, and these are central to the main point of the paper.

In the last round of review, this reviewer mainly raised three concerns or questions. 1) The folding/unfolding status of our sample; 2) Relationship between the three classes of conformers and three specific conformers; 3) Back-calculated SAXS curves with or without including the consideration of hydration layers and its impact on the calculation of the synthesized SAXS curves. We have addressed those concerns and questions in detail and the reviewer did not object to our explanations and illustrations for the last two, but remains unconvinced about the first concern. Regarding the first concern about folding/partial folding/unfolding, it is noteworthy that the reviewer did not raise any questions about the images of the direct visualization by AFM that shows the clear differences that can be identified between the folded, partially folded and unfolded molecules. These images provide axiomatic evidence without any ambiguity or interpretation bias. It is now clear to us that the reviewer is conflating conformational heterogeneity with states of folding. The assertion that the formation and deformation of weak and transient tertiary interactions (which are ubiquitous among RNAs) are equivalent to the folding and unfolding of an RNA molecule is unfounded. It is well known that RNA, particularly in cellular environments or in near-physiologically relevant in vitro conditions, exists as an ensemble of molecular conformations. In many cases (e.g., riboswitches and RNase P RNA), such fluctuations play a critical role in sampling the structural-chemical environment for the recognition of ligands/substrates. It is true that upon binding of ligand/substrate (or in a solution containing an elevated non-

physiologically high concentration of Mg^{2+}), the RNA can be stabilized in one or more energetically favorable compact conformations. However, by no means does this imply that every other conformation the RNA can adopt in the absence of ligand/substrate is “unfolded” or “physiologically irrelevant.” Moreover, all particles we observed by AFM show completely intact secondary structure, and a combination of tertiary interactions, some of which are transient. Here, we would like to address the reviewer’s statement on several fronts in greater detail and with referenced examples.

The conceptual difference between “unfolded vs. partially folded vs. folded” and conformational heterogeneity. The former is the simplistic description of distinct thermodynamic/kinetic and structural states, which originate from studies of protein folding and unfolding. This concept should not be confused with heterogeneous conformations of RNA where they co-exist in physiologically relevant solutions or in cells⁵⁻¹⁰. In fact, virtually no ncRNA exists in a single native conformation but rather samples many different conformations in a rugged energy landscape⁵. Often, the functional form may not be thermodynamically favored over other conformers¹¹. For the simplest example, the HIV-1 TAR RNA, a small RNA hairpin with only ONE bulge, is flexible and able to sample through considerable conformational space in solution and in cells^{5,12}. One could imagine the vastly broader conformational space that the RNase P RNA could sample. Indeed, it consists of two large domains, connected by a highly flexible linker, TEN bulges and internal loops (**Fig 5d in the manuscript**), and many other flexible structural elements. Even at 70 mM Mg^{2+} , the conditions under which the crystal structure of *Gst* RNase P RNA was determined, the electron density of the whole S-domain was not observable¹³, exemplifying RNA conformational flexibility and dynamics. Thus, the use of the terms folded/partially folded to describe the heterogeneous conformations of the RNase P RNA, which consists of two domains linked by a flexible linker, is incorrect.

To further illustrate the difference between conformational compaction and folding-unfolding processes, we performed isoenergetic experiments using ITC, where we titrated the RNase P RNA solution with Mg^{2+} from 0.1 to 5 mM while monitoring the energy exchange (**Fig. 1, this response**). Note that the RNase P RNA was never unfolded during the sample preparation process. We repeated the same experiment with the beet western yellow virus pseudoknot RNA to monitor the energy change during the process of unfolding/folding. After appropriate background subtraction, we obtained the net energy exchange during the titration (**Fig. 1, this response**). Even though BWYV pseudoknot is 15 times smaller than RNase P RNA, the energy change of refolding of BWYV pseudoknot RNA is more than 10 fold of the RNase P RNA compaction. The small energy exchange, seen in the RNase P RNA curve, < 0.4 kcal/mol, is likely attributed to hydration changes associated with both RNA and Mg^{2+} ions, or a trace amount of unfolded RNA. The reviewer stated in his/her comment: *I am not convinced the molecule is in its folded state under (at least some of) the conditions of these studies. We hope that our ITC data, which clearly shows no major folding events from 0.1 mM to 5 mM Mg^{2+} convinces the reviewer that there are no unfolded or partially folded species analyzed or presented in this study.*

Fig. 1. Comparison of isothermal titration calorimetry (ITC) analyses of Mg^{2+} -dependent compaction of RNase P RNA and refolding of beet western yellow virus (BWYV) pseudoknot RNA. Left, ITC thermogram of Mg^{2+} -dependent compaction of RNase P RNA (417 nucleotides). Middle, Thermogram of Mg^{2+} -dependent refolding of BWYV pseudoknot RNA (28 nucleotides). Right, isotherm comparison between the compaction of RNase P RNA and refolding of BWYV pseudoknot RNA. Even though BWYV pseudoknot is 15 times smaller than RNase P RNA, the energy change of refolding of BWYV pseudoknot RNA is more than 10 fold of the RNase P RNA compaction. The sample purification of the BWYV pseudoknot RNA and the duplicates of the isothermal titration calorimetry thermograms are provided in **Suppl. Info. Fig. 7 and 8**, respectively.

Now, let's discuss some of the known differences between the environments relevant to RNase P RNA folding and activity in vitro and in cells. There is no study done on the conformational heterogeneity of RNase P RNA of any type in cells. However, the environmental differences that impact RNA folding are known. They are co-transcriptional folding, binding of proteins, small molecule metabolites and ions, in particular Mg^{2+} ¹¹. The impact of co-transcriptional folding is directly related to the transcriptional speed. The studies by two independent groups show the folding of bacterial RNase P RNA is not affected by transcriptional speed regardless of the type of RNA polymerase (bacterial POL or phage POL) used^{14,15}.

Furthermore, like the RNase P from other bacterial strains, the RNase P holoenzyme from *Geobacillus stearothermophilus* (*Gst* RNase P) used in our study consists of an accessory protein, rnpA (14 kDa). Extensive studies have been done showing unambiguously that the binding of the protein component of the holoenzyme does not facilitate the folding of the P RNA component, but rather enhances the binding of substrate RNA and catalytic Mg^{2+} ions¹⁶⁻¹⁹. The enhancement of the substrate binding is realized via the direct interaction between the central cleft of the protein and the single-stranded 5' leader sequence of pre-tRNA (4 – 8 nucleotides away from the cleaved phosphodiester bond).²⁰ Pre-tRNA binding studies examining the dependence of affinity on the length of the 5' leader indicate that a 4- or 5-nucleotide leader is required to achieve the enhanced affinity conferred by the protein component²⁰.

Further cross-linking experiments also confirm the stabilizing interaction between the protein component and the RNA substrate ²¹⁻²³.

[REDACTED]

While the RNA structural biology and biophysics communities have been studying RNA structures under elevated Mg^{2+} for many justifiable reasons, please do not lose sight of the actual picture of RNA conformational heterogeneity under the physiologically relevant Mg^{2+} concentration.

[REDACTED]

One can imagine (and appreciate), then, how much more structural variability exists at 1mM Mg^{2+} .

[REDACTED]

About the study cited by the Reviewer (Fang et al., Biochemistry), we would like to point out the following key facts. There are several critical differences between the samples used in our study and theirs. These differences are important when comparing the results of the two studies. First of all, the RNase P RNA construct used in the Fang et al. study consists of only the catalytic domain (i.e., without the highly flexible substrate specificity domain), and it is also artificially permuted at the 5' and 3' ends. Thus, its biological activity could not be assayed in relationship to the “folding” states. In contrast, our sample is the full-length (417-nt) RNase P RNA containing both domains and the catalytic activity can be assayed using the standard protocol. Secondly, Fang et al. prepared the RNA sample starting from the completely unfolded (U) state (no secondary structure), denatured using 7 M urea, followed by heating at 85-90°C and cooling for renaturation. In comparison, our RNA sample was prepared under native (non-denaturing) conditions throughout and was never subjected to denaturants such as urea, nor to high temperatures. As we mentioned in our response in the revision, we have tested numerous protocols to ensure the RNA stays in the folded state. In our hands, once this RNA, or any other large RNA with complex fold, is denatured, it is extremely difficult if not impossible to obtain the native sample without contamination of misfolded, unfolded, or partially unfolded molecules, as illustrated by direct visualization under AFM (**Extended Data Fig. 1b in the manuscript**) (see also one of our recent publications on the subject: *Ding. J., et al., RNA heterogeneity visualized under AFM in Integrative Structural Biology, in Integrative Structural Biology, pp388-269, Royal Soc. Chem; 2023*). Thus, preparing the full-length RNase P RNA under non-denaturing conditions is absolutely required and critical for this study. We are puzzled by the reviewer's claim that our sample contains a mixture of folded and partially unfolded molecules, despite our direct visualization result (**Fig. 1; Extended Data Fig. 1b in the manuscript**), which the Reviewer has never challenged.

Returning to the concepts of “folded vs. partially folded vs. unfolded” and “conformational heterogeneity” under physiologically relevant Mg^{2+} . The difference between these two concepts is not simply a matter of terminology. Fang et al. observed clear differences between the named intermediate and unfolded states only when an extra 4-8 M urea was added to the sample. Unfolded and intermediate states showed a $P(r)$ function with a vast number of extra oscillations and a Kratky plot with linear behavior at low q , indicative of unfolding behavior. The reviewer made a surprising assertion that “*The data shown in Fig. 4f, along with the Kratky plots (4b) and the $P(r)$ (4c) all clearly indicate that it is not (folded)*”, as this assertion is directly contradicted by the SAXS data profiles. We kindly ask the reviewer to re-examine our results (**Figs. 4a, b and c in the manuscript**); there is no observation of such unfolding in our data. In contrast, our data shows a classical behavior of folded particles, a well-defined peak in the Kratky plot whose maximum does not show a shift at different Mg^{2+} concentrations, and a well-known $P(r)$ profile of folded particles, indicative of a compact average shape (**Fig. 4 in this response**). For the basic theory and practice of SAXS, please review the literature (Kikhney and Svergun, A practical guide to small angle X-ray scattering (SAXS) of flexible and intrinsically disordered proteins, FEBS Letters, Volume 589, Issue 19, Part A, 2015, <https://doi.org/10.1016/j.febslet.2015.08.027>).

Fig. 4. Simulation of the SAXS curve, Kratky plot and $P(r)$ function of three conformers of RNase P RNA: a highly compact conformer (S105), a model derived from the crystal structure (FLRNaseP), and an extended conformer (S53) where the S- and C-domains are open via the flexible linker between them. In each case, there is a distinct peak in the Kratky plots indicative of folded particles. The longest dimension of each particle can be seen in the $P(r)$ plots where S53 is the most extended among the three. The open conformation should not be confused with “unfolded.”

Lastly, we would like to respond to the statement “*Unless they can make the case that this is the physiological form of the molecule (so the partially folded state is actually found in the cell), I cannot agree that conclusions reached at 1 mM magnesium are physiologically relevant, and these are central to the main point of the paper*”. Given the minimal impact of the cellular environmental factors, such as cotranscription^{14,15}, the accessory protein¹⁶⁻²³ and monovalent ion concentrations (**Fig. 4, this response**) on the folding of this RNA, as well as the axiomatic visual evidence and rigorous analysis

disseminated throughout this manuscript, the authors would kindly ask if anyone could draw any other reasonable scientific conclusion other than what we have presented. Our finding is consistent with the increasing number of literature reports about RNA conformational heterogeneity in vitro and in cells^{5,6,8-10,12,24}. Furthermore, the authors kindly ask if the reviewer could provide explicit information indicating possible other known cellular factors affecting the RNase P RNA folding, without which the authors believe a separate study to probe the heterogeneity in cells is unnecessary as well as unreasonable. Lastly, the authors are curious if the reviewer has evidence that the cellular Mg²⁺ is not near 1 mM but much higher despite the consensus in the literature. The authors would also like to caution against the notion that observations of molecular structure made in vitro are not valid unless cellular data are presented. Such an assertion would trivialize the significance of the whole body of structural biology studies done in vitro, which we believe is not the reviewer's intention.

Reviewer #4:

In their exchange with the Reviewers, the Authors have made two points that I find somewhat contradictory. For one of them they argue that surface immobilization of RNase P in the course of AFM measurements has no impact on its sampled space of conformations since SAXS, as an independent solution technique, fully corroborates the AFM findings. For the other, they argue that conformational variability of RNase P and its sensitivity to changes in Mg²⁺ concentration at near-physiological conditions are central to understanding its function.

The authors agree that these are two main points presented in this study, but we fail to see any contradiction or logical inconsistency between them. We would restate our conclusions as follows. 1) Our results demonstrate the physiological significance of 1 mM Mg²⁺ in RNase P activity. 2) Our AFM images of surface-immobilized particles sample the RNA's conformational space at 1 mM Mg²⁺, revealing a wide range of conformationally heterogeneous structures, 158 of which were recapitulated using HORNET and used to describe the conformational space through PCA. 3) The use of SAXS as an orthogonal method corroborates our AFM observations.

I agree with the Authors' second point on flexibility being the central aspect of RNA's functional behavior at physiological free Mg²⁺ concentrations. Indeed, as shown in the Extended Data Table 1, the change in the Mg²⁺ concentration from 1mM to 2.5mM changes the fraction of the compact crystal structure-like state from 19% to 95%. At 1mM Mg²⁺ RNase P appears to exhibit the highest response to that variable, suggesting an ensemble of conformations...

We appreciate that the reviewer agrees with us on the central aspect of the biological relevance of conformational heterogeneity in 1 mM Mg²⁺ solution. We believe that the discoveries presented in this manuscript are very significant in light of recent developments in RNA biology and the historical misconceptions about RNA 3D structure.

We now address the reviewer's remaining points and comments in the following paragraphs. First, however, we thank the reviewer for the thoughtful and constructive comments about the conformational ensemble and the analysis of the SAXS data. The reviewer's comments led us to identify the sources of the confusion. We would like to emphasize that the reviewer's comments have, in our view, resulted in a better-articulated and stronger manuscript overall. We have revised the manuscript by expanding relevant sections and adding new data in both the main text and Ext Data to address the reviewer's comments. Since the reviewer's remaining points are distributed and repeated in different places throughout the review, we have summarized them into two main concerns for clarity:

1. The completeness of the conformation sampling by the pool of 158 discrete conformers and the impact of immobilization.

This concern is regarding whether the pool of 158 conformers faithfully represents the complete solution ensemble. We believe that this concern might be caused by confusion due to, in part, the lack of clarity in the original manuscript, including the lack of a more detailed description of how and why the 158 conformers were selected from the AFM images, and why they are sufficient to approximate the full conformational space. Moreover, as the reviewer pointed out, the impact of the immobilization could lead to impeding the interconversion between transient conformers, or alteration of the native structures of conformers, thus distorting the final results. When raised, we address those concerns point-by-point below.

2. Analysis of SAXS data and the conformation ensemble.

The comments related to the SAXS analysis and interpretation are helpful for us in identifying the sources of confusion, and we thank the reviewer for those comments. In the following response, we address these concerns by clarifying data in the original manuscript, and by providing additional data here and in the revised manuscript.

Point-by-point responses to specific comments related to the above two concerns:

... the Authors' view that immobilization of a flexible RNA on a coated mica surface has no impact at all doesn't seem entirely plausible. In addition to the Mg²⁺ concentration, a pool of presumably transiently sampled and inter-converting conformations could be sensitive to a wide range of factors, such as temperature, RNA concentration, or the presence of a foreign surface they are forced to interact with, as is the case with AFM...

First of all, regarding the statement "*In addition to the Mg²⁺ concentration, a pool of presumably transiently sampled and inter-converting conformations could be sensitive to a wide range of factors, such as temperature, RNA concentration, or the presence of a foreign surface they are forced to interact with, as is the case with AFM...*", Yes. It is known and we agree with the reviewer that RNA conformation can be affected by a wide range of factors. The focus of this study is to capture conformations under 1 mM Mg²⁺ solution condition.

About the impact of immobilization. The consensus in the AFM field is that immobilization of nucleic acids on a mica surface causes minimal perturbation to their native structures²⁵⁻³⁰. One unequivocal line of evidence is the direct visualization of double-stranded DNA and RNA, where the geometry of major and minor grooves is similar to the known values³¹ determined by crystallography. For technical details described in plain language, please see link <https://www.azom.com/article.aspx?ArticleID=12183>. Of note, the same forces, such as H-bonds and base-stacking interactions drive the formation of duplexes as well as RNA folding. Thus, any significant alteration of structures would have been observed

in duplexes, not just in topological folds. We do not rule out small local-scale structural perturbations whose detection is beyond the intrinsic resolution limits of the method. In the end, AFM is a low-resolution technique.

The other aspect of the reviewer's comment is about the "sensitivity of transiently sampled interconverting conformations" to factors such as surface immobilization. We stated in the manuscript that our method is a "shotgun" approach by capturing the topography of hundreds of individual particles at once, each of which may exist in a distinct conformation and represent a sampling at the time the molecules were deposited onto the mica surface. We agree with the reviewer that the immobilization may interfere with interconversion and conformational equilibrium, and thus may have overlooked transient conformational species. However, the PCA analysis of the discrete conformers captured, in principle, derives motion trajectories that faithfully cover the conformational space sampled by the RNA, including those of transient species, provided both the maximum and minimum variances in the conformational space spectrum and sufficient number of conformers in between are known. The pool of 158 conformers exhibits a broad and well-distributed range of R_g (~43–59 Å), defined by the relative positions of the S- and C-domains with respect to one another. The full-length model of the crystal structure (R_g ~46 Å) falls on the lower end of the spectrum, and the biological relevance of extended conformers has been implicated previously³². The completeness of the PCA-derived conformational trajectories can be estimated based on the population of variance (70% and 80% with the top 5 and 7 eigenvalues, respectively) (**Extended Data Fig. 5 in the manuscript**) and the total number of the discrete conformers (158) in the basis set (**Extended Data Fig. 4 in the manuscript**). For comparison, a much smaller pool was used in two reported protein studies: only 53 conformers of transducin¹ and 37 conformers of the kinesin motor domain².

Our study of the conformational space of RNase P RNA in 1 mM Mg^{2+} represents a pioneering breakthrough in the field. While there are **neither absolute experimental nor theoretical methods known to us** to ascertain the complete conformational space of an RNA in solution more accurately than what we have presented here, there are correlation metrics one can apply to check for internal consistency among some of the conclusions drawn regarding the conformational space. First, the high Pearson correlation factor of 0.8 between the RMSF derived from the PCA analysis of the 158 conformer pool and the crystallographic thermal B factor (**Figs. 3a&b in the manuscript**) implies the consistency between the PCA-derived conformational space and thermal motion as a function of residue/domain. The RMSF is also highly correlated in 3D with the local resolution cryo-EM map (**Fig. 1, this response**). The uncertainty in positions of those peripheral structural elements and the S-domain is clearly illustrated and they are dynamic even at 10 mM Mg^{2+} (**Fig. 1, this response**). Perhaps precisely for this reason, the conformer structures in 1 mM Mg^{2+} could not be determined using conventional signal-averaging methods such as Cryo-EM or crystallography, which signifies the importance and urgency of our work.

Fig. 1. Comparison of cryo-EM local resolution map and RMSF (*unpublished data*). Cryo-EM local resolution of the RNaseP RNA structure obtained at 10 mM Mg²⁺ (blue) and the RMSF of the 158 HORNET conformers (red), plotted as a function of residue. The Pearson correlation factor is ~0.74.

In that respect, I don't find the Authors' argument regarding SAXS perfectly corroborating AFM entirely convincing. Strict equivalence between the hypothetical solution-state (SAXS) and the surface-immobilized (AFM) ensembles would have implied that the experimental SAXS data matches structure-modeled data averaged over the entire 158-member ensemble recovered by AFM, without any need for modification of the AFM ensemble weights. However, the Authors have clearly adjusted the ensemble member weights to fit the SAXS data. The Authors could have stated that they expect only the populations and not the conformations themselves to be affected by the change from the solution to the surface-associated state...

We do not claim the equivalence of the 158 conformers and what is in solution, particularly with respect to populations. The captured conformers are 158 samplings of the solution ensemble. The purpose of using SAXS is to characterize the pool of the conformers and approximate them into representative classes that can sufficiently describe the scattering profile. The reviewer is correct that the populations of conformer classes, which are unknown by experimental means, were obtained by fitting, and a detailed description of the fitting procedure was given in **Methods**. Of note, this fitting procedure differs from what we reported in a previous study (Ding et al., Nat. Comm., 2023), where the populations of each type of conformer could be approximated by tallies of their distinct topologies. The reviewer might agree with us that the low-resolution SAXS data are useful in approximating ensemble behaviors, even when no explicit coordinates of all conformers are available because of degeneracy, as the reviewer pointed out. We have revised the manuscript to avoid the confusion.

*However, a much stronger claim of complete equivalence is being made, and I find that harder to defend. The Authors are reporting that an optimally weighted ensemble of 3 conformations out of the AFM-derived 158 satisfactorily fits the data. This is plausible, in line with the Authors' prior results such as those reported in reference 23. **The Authors then state on page 31: "classification based on quantitative partitioning using scattering intensity traits ... shows that the three classes are sufficient to describe the ensemble of 158 particles". In other words, the Authors are claiming that scattering intensity profiles predicted from the three best-fitting conformations are indistinguishable from those of the remaining 155 conformations.** However, the evidence for that and the numerical criteria for considering equivalence are not presented. Even if we disregard the common appearance of discrepancies at increasing q in the scattering data predicted from two different structures, such claim would have to hold for the lowest- q data that define the gyration radii. The entire set of 158 AFM-derived models would need to have the distribution of their predicted radii of gyration tightly clustered around the 3 values corresponding to those of the 3 best-fitting conformers.*

We thank the reviewer for the comments and the concern. Before we get to the details, we do not claim the equivalence between the three best-fit conformers and the rest of the 155 conformers or their SAXS profiles. Our purpose is to characterize the ensemble by grouping them into classes based on conformational characteristics and SAXS data. We partitioned the theoretical scattering profiles respectively for each of 158 models using both R_g and cosine distance²⁵ among all variations of intensity as a function of q . A more detailed description of the procedure, which was inadvertently omitted during the manuscript preparation, is added in **Methods**. Given that SAXS is a low-resolution technique, and that the 158 conformers are not exhaustive, it makes sense to classify the derived conformations into a minimal set that can sufficiently describe the solution ensemble. However, this is very different than saying that three individual conformers are equivalent to or indistinguishable from the remaining pool of conformers. We never state this nor imply this.

To further address the reviewer's concern about the classification, we have also applied PCA analysis³ and the commonly used EOM method⁴ to further illustrate and examine the 158 conformer pool. The PCA analysis without using SAXS data as an input classifies the conformer classes solely based on conformational trajectory analysis (the PCA distance as a function of clusters/classes). The PCA components clusterize into the three classes, with the extended structures as the smallest group, the compact conformers in the next and the semi-compact (or semi-open) conformers making up the largest class (**Fig. 2, this response**).

Fig. 2. Clustering (classification) analysis of the PCA components 1-7 derived from 158 models. **(a)** Dendrogram of clustering as a function of PCA distance, **(b)** R_g distribution for each cluster, where the cross symbol represents the average value.

The EOM analysis was performed using the 158 conformers as the input to feed the algorithm. For this analysis, we used a maximum of 50 conformers per ensemble, a number of ensembles per generation of 50, and a minimum number of models per ensemble of 1, with no curve repetition. The EOM performed the fitting for 100 cycles of repeated search. The best representative fit achieved has χ^2 of 2.3 and 1.1, respectively for SAXS data recorded at 1 mM and 5 mM Mg^{2+} . The results (**Fig. 3 in this response**) indicate that RNase P RNA presents three main distributions of R_g with high frequency, where the largest class is centered around $R_g \sim 51$ Å, the compact class with R_g ranged between 47.5 – 50 Å, followed by a more open class of conformers with $R_g > 55$ Å. On the other hand, at 5 mM Mg^{2+} , the majority of the population shifts to smaller R_g values, $R_g < 47.5$ Å, with no significant counts sampled with $R_g > 52$ Å.

Fig. 3. EOM analysis of SAXS intensity profile, **(a)** Frequency distribution as function of the ratio of gyration (R_g) in angstrom recorded for RNase P RNA at 1mM Mg^{2+} (blue) and 5mM Mg^{2+} (orange), using 158 models as a pool of library structure (black). The 5 mM Mg^{2+} data was used here for comparison and consistency check. **(b)** Representative conformer structure models for 1mM Mg^{2+} based on the EOM analysis. These three models show similar structural characteristics as the three models shown in **Fig. 2c** and **f**.

Thus, we now have three separate analyses that lead to the same semi-quantitative conclusion, i.e., there are roughly three classes of conformers present in solution at 1 mM Mg^{2+} , although the details might be different because different sets of criteria were used to classify the classes in each approach (**Fig. 4, this response**). We elect to present the result using our algorithm for the classification because it takes into consideration both the characteristics associated with the flexible RNA conformational ensemble as well as experimental SAXS data. The PCA and EOM classification results will be provided in the Extended Data as supporting analyses.

Fig. 4. Populations of three classes derived using EOM (left) and PCA (right).

On page 6 the Authors report standard deviations of the radii of gyration for the conformers belonging to the three classes as 2A, 2A, and 3A. Considering that these

are standard deviations, the respective distributions are falling well outside of what could be considered an achievable precision of a SAXS-extracted gyration radius (0.5-1Å). Therefore, it appears that many of these conformers should not have been ascribed to any of the (C1, C2, C3) classes, and may thus represent possible differences between the surface-immobilized and solution-state ensembles, which brings us back to the argument of the impact of AFM immobilization.

We understand the reviewer's concern. While the accuracy of our Rg measurements is within the expected range in our hands, we believe that the reviewer is comparing the ensemble behaviors in terms of the measured Rg with that calculated based on individual structural models, whose Rg may fall outside of the measurement accuracy range. Those structural models that fall outside of the range may share structural characteristics similar to their classes and/or with low populations based on the criteria used in the calculation.

Let's examine the conformational pool where some of the structural models fall outside of the measured Rg ranges, which could be caused by artifacts such as immobilization. The presence of a large number of artifact structures could distort the derived conformational space. Thus, we re-examined the conformational space using only structural models with Rg of precision ± 0.5 or ± 1.0 Å. **Fig. 5 in this response** shows that the conformational trajectories in terms of RMSF remain similar to those derived from the full 158 conformer pool. This result suggests that our pool of 158 conformers is neither under-sampled nor contains a significant number of structures outside of the motion trajectories caused by artifacts. The key elements in the basis set are the conformers that sufficiently cover the full spectrum at both ends as well as the space in between. Note that 1580 structural models, top 10 from each of 158 conformer calculations, were used to compute the conformational space shown in **Fig. 3 in the manuscript**.

Fig. 5. Root-mean-square fluctuation (RMSF) and eigen-ranked structure variance derived from principal component analysis (PCA). **a**, RMSF (upper panel) and

proportional structure variances ranked by PCA eigenvalue (lower panel) among all 158 resolved structures (top 1 for each conformer calculation). **b**, PCA analysis using the same conformers selected from the three classes (**Figs. 2d,g in the manuscript**) but excluding conformers whose R_g values are $> \pm 1.0 \text{ \AA}$ from the values of the three represented conformers (**Figs. 2b,g in the manuscript**). **c**, Same analysis but excluding the conformers whose R_g values are $> \pm 0.5 \text{ \AA}$ from the values of the three represented conformers.

I suspect that the SAXS data at 1mM Mg2+ could be similarly fitted with a 2-component ensemble, as the weight of Class 3 is only 6%. I did not find the chi2 value for the best-fitting N=2 ensemble, but I am estimating it to be close to chi2=2.0 based on the Figure 2. It is difficult to visualize the significance of that fit improvement relative to chi2=1.7 of the N=3 ensemble, or the precision with which this very minor component could be defined structurally. I also note that ED Fig 6 shows that perfect Chi2=1 fits are only observed for Mg2+ at or above 2mM. Therefore, fit results at 1mM Mg2+ are likely associated with some residual data/model discrepancies and the improvement in the fit from chi2=2 to chi2=1.7 may not be the final detail in the overall picture.

In terms of χ^2 , we have shown that class C2-like particles have a good agreement with the experimental SAXS profile (**Fig. 2d in the manuscript**). This is simply because the C2 class is by far the largest class consisting of 117 conformers. The C2 conformers have conformational characteristics where the S and C domains are neither fully closed nor fully open. We think this is the most plausible conformer class populated in solution, as the two domains are neither restricted (as in the compact class) nor extended (as in the C3 class). The combination of C2 and C1 can reach χ^2 as small as 2.13 but the combination of C1, C2 and C3 gives the best fit with a smaller minimum and standard deviation after 1000 iterations (**Fig. 2e in the manuscript**).

Regarding the statement “*Therefore, fit results at 1mM Mg2+ are likely associated with some residual data/model discrepancies and the improvement in the fit from chi2=2 to*

Fig. 6. Ensemble fitting of the SAXS data using three (blue) or two classes (red), and their residuals.

chi²=1.7 may not be the final detail in the overall picture”, we are not sure what the reviewer meant here. We are assuming the viewer is asking for the information about the fit and residual plots using two vs. three conformers, the former of which suggests underfitting in the low-q area without the contribution from C3 (Fig. 6 in this response).

It is clear from the Authors' data that something resembling the crystal structure becomes an increasing fraction of the solution ensemble with increasing Mg²⁺ concentration. The question is how confident can one be regarding the ensemble composition. Even if we reject the possibility of an impact of surface immobilization, the set of 158 AFM-derived conformations is unlikely to be exhaustive and SAXS data are known to be degenerate with respect to the 3D structure, with this effect even more pronounced with the relatively limited q_{max} of 0.35Å⁻¹ and sizeable data noise at higher q. It is clear that SAXS data at 1mM Mg²⁺ point to the presence of conformations with the gyration radii in excess of 50Å. I also find it plausible that the presence of a number of states differing from the “crystal structure-like” could be inferred from such data. I am less confident that these AFM-processed and SAXS-fitted classes are equivalent to actual structures present in solution, due to caveats of both SAXS and AFM, as noted above.

The maximum q of our SAXS data is 0.45 Å⁻¹. The reviewer questions the completeness of the conformational pool and thus is less confident that the SAXS-derived conformers are equivalent to what is actually present in solution. As discussed in the response to the previous comments, we do not claim nor indicate that the pool of 158 conformers is (or is equivalent to) all possible conformations in solution. Instead, the pool represents 158 samplings, which already far exceed what is needed for PCA analysis and what has been used in studies of protein conformational space^{1,2}. As we have pointed out to another reviewer, the topological structure determination of 158 conformers in solution using our novel method is in itself a significant advance, and should not be overlooked. Since this pool covers the range of very compact to very extended conformations, the majority of which lies in between, the PCA of those conformers generates a reliable mapping of motion trajectories (direction and amplitude), which would include interpolated structures that fall within those trajectories but may not have been directly captured/observed. Therefore, we believe our analysis to faithfully represent the conformational space, despite not having captured every possible conformation. The prospect of sampling an “exhaustive” larger pool (e.g., an entire mica surface containing over a hundred billion molecules in 20 ul of 10 nM solution) is neither necessary nor computationally practical. It took about 24 million CPU hours (10,000 CPU for 100 days nonstop) to compute the structures of 158 conformers, not to mention the time spent optimizing the computation protocol. Moreover, currently, there isn't any experimental or theoretical approach to solve the RNA conformational space question and our study represents a very significant advance in the field.

The results of the study essentially suggest that the amount of information contained in the SAXS and AFM data is sufficient for derivation of a best-fitting solution with up to 3 “states” with each “state” representing, at a several nm

resolution, a heterogeneous ensemble of potentially diverse conformations, subject to the q-range and the signal/noise profiles of the measured SAXS data. Therefore, I consider the “high resolution” part of the manuscript results primarily localized in the section titled “Conformational space of the RNase P RNA” to be somewhat speculative.

We caution against the use of “states” rather than “classes.”. By the word “states,” in conventional thermodynamics, one implies that the molecules in different groups are energetically distinct and separated. The conformers in this study are energetically equivalent and the change from one conformer to another is almost isoenergetic. In the RNase P RNA case, the net energy change associated with “compaction” due to an increase in Mg^{2+} is the equivalent of less than one hydrogen bond. In contrast, the energy involved in the transition between the two states, for example, a small 28-nt BWYV pseudoknot RNA, is over 4 kcal/mol (**Fig. 7, this response**). This result suggests that conformational sampling in 1 mM Mg^{2+} is quasi-isoenergetic. In contrast to transitions between thermodynamically well-defined states.

Fig. 7. Comparison of isothermal titration calorimetry (ITC) analyses of Mg^{2+} -dependent compaction of RNase P RNA and refolding of beet western yellow virus (BWYV) pseudoknot RNA. Left, ITC thermogram of Mg^{2+} -dependent compaction of RNase P RNA (417 nucleotides). Middle, Thermogram of Mg^{2+} -dependent refolding of BWYV pseudoknot RNA (28 nucleotides). Right, Isotherm comparison between the compaction of RNase P RNA and refolding of BWYV pseudoknot RNA. Even though BWYV pseudoknot is 15 times smaller than RNase P RNA, the energy change of refolding of BWYV pseudoknot RNA is more than 10 fold of the RNase P RNA compaction. The data were recorded at 37 °C.

Regarding the statement “*Therefore, I consider the “high resolution” part of the manuscript results primarily localized in the section titled “Conformational space of the RNase P RNA” to be somewhat speculative*”, we are not sure what the reviewer means by “high resolution” to characterize the description of the conformational space in the section “*Conformational space of the RNase P RNA*,” or anywhere else in the manuscript. The motions discussed in the section are on the magnitude of over 10 Angstroms, and no discussion about structures and motions at 3 Angstroms or lower are presented. *The only place where resolution is mentioned is in reference to AFM images (10–15 Å) in the section “Visualizing conformational*

heterogeneity.” If referring to the estimated accuracy of recapitulated structures by HORNET, this is covered in detail in the other manuscript, which has already been accepted for publication. Either way, we do not claim nor imply “high resolution” of structural information in any way. Furthermore, we do not believe that the reviewer would challenge the validity of a well-accepted statistical analysis method (PCA) that has been widely used in science, engineering and economics, and is one of the most fundamental backbone tools in big data science and machine learning.

I note that the Authors’ protocols do not appear to include any aspects of validation, which would be useful to detect over-interpretation of the data.

We thank the reviewer for bringing in this point and have provided the following information that addresses three main aspects central to the reviewer’s concerns:

1. The quasi-isoenergetic ITC data (**Fig. 2b in this manuscript**) to illustrate the compaction and associated conformational changes among the three “classes” is thermodynamically distinct from the transition between energetic “states.”
2. Classification/grouping based on the PCA and EOM (Ensemble Optimization Method) analyses (**Extended Data Figs. 4b-d**) in the revised manuscript), each of which uses different classification criteria, and all of which conclude with three conformational classes from the same pool. These analyses at least are consistent with the SAXS data analysis and complement the SAXS data, which has its inherent limitations, such as low resolution and degeneracy, as the reviewer pointed out.
3. PCA of the sub-populations that included models outside of ± 0.5 and ± 1.0 Å of average R_g were removed to test if the original pool contains significant numbers of distorted conformers due to immobilization (**Extended Data Figs. 5d-e**) in the revised manuscript).

As a separate point, I was not able to find in the Methods section the RNA concentration(s) that SAXS data were recorded at and what constitutes the fitted data, and thus, how confident are the Authors that the SAXS-extracted gyration radius and the lowest-q data in general, are free from the concentration effects. This is an important point as the determination of the less-compact states would require high confidence in the lowest-q data.

We thank the reviewer for this comment. In this revision, we provided the specific information in **Methods**. The concentration information was provided along with the deposited SAXS data. Regarding the concentration dependence at the lowest q, please see **Fig. 8 in this response**.

Fig. 8. Left: The SAXS data recorded with two concentrations and differential; right: the low-q regions, residuals, and R_g .

To further analyze the concentration dependence and detect any sample concentration dependence in SAXS data analysis, we performed the CorMap analysis of two data sets recorded at 0.38 and 0.79 mg/ml concentrations, which cover the sample concentrations (0.54 - 0.55 mg/ml) used for the data collected for this study. The scaled SAXS profile with respect to the distinct concentration shows a reduced χ^2 test of 0.99 and CorMap²⁶ test of $C=8$ and P value of 0.84 (**Fig. 10 in this response**). The Guinier region presents the same variation and R_g values with differences at the decimal range, with no distinct difference under the experimental uncertainties.

CorMap is expected to show a random lattice pattern for non-correlated data. However, for systematic differences, the 2D plot will show continuous areas of positive (+1) or negative (-1) correlation. As shown in **Fig. 9 in this response**, the comparison between the two SAXS profiles at different sample concentrations has no clear distinguished pattern using all best practices to assess 1D scattering profile similarity (R_g , χ^2 and CorMap), the differences are not statistically significant.

Figure 9. SAXS data analysis for two different sample concentrations. **a**, SAXS profiles at 0.38 (black) and 0.76 mg/ml (red) scaled by nominal sample concentration. The inset plot shows the respective Guinier plot with determined R_g values. The reduced χ^2 test for the profiles is 0.99. **b**, CorMap correlation matrix as a function of q (\AA^{-1}) with a probability of similarity (P-value) of 0.84. Note that the RNase P RNA concentration for the AFM experiments was ~ 0.33 mg/ml.

All SASX data are deposited in the SAS data Bank:

<https://www.sasbdb.org/data/SASDTA7/pvvn3t155t>
<https://www.sasbdb.org/data/SASDTB7/s177uunamr>
<https://www.sasbdb.org/data/SASDTC7/ljt511ik1v>
<https://www.sasbdb.org/data/SASDTD7/jdc3df6ixm>
<https://www.sasbdb.org/data/SASDTE7/jssqnj377a>
<https://www.sasbdb.org/data/SASDTF7/gzjfc42q9>
<https://www.sasbdb.org/data/SASDTG7/lo4f05cuy6>
<https://www.sasbdb.org/data/SASDTH7/mvbyc36363>
<https://www.sasbdb.org/data/SASDTJ7/5zkojd9cw5>

The project summary can be found by link <https://www.sasbdb.org/project/2201/b9y8c4b6qf>

References:

- 1 Yao, X. Q. & Grant, B. J. Domain-opening and dynamic coupling in the alpha-subunit of heterotrimeric G proteins. *Biophys J* **105**, L08-10, doi:10.1016/j.bpj.2013.06.006 (2013).
- 2 Grant, B. J., McCammon, J. A., Caves, L. S. & Cross, R. A. Multivariate analysis of conserved sequence-structure relationships in kinesins: coupling of the active site and a tubulin-binding sub-domain. *J Mol Biol* **368**, 1231-1248, doi:10.1016/j.jmb.2007.02.049 (2007).
- 3 Grant, B. J., Rodrigues, A. P., ElSawy, K. M., McCammon, J. A. & Caves, L. S. Bio3d: an R package for the comparative analysis of protein structures. *Bioinformatics* **22**, 2695-2696, doi:10.1093/bioinformatics/btl461 (2006).
- 4 Tria, G., Mertens, H. D., Kachala, M. & Svergun, D. I. Advanced ensemble modelling of flexible macromolecules using X-ray solution scattering. *IUCrJ* **2**, 207-217, doi:10.1107/S205225251500202X (2015).
- 5 Bothe, J. R., Nikolova, E. N., Eichhorn, C. D., Chugh, J., Hansen, A. L. & Al-Hashimi, H. M. Characterizing RNA dynamics at atomic resolution using solution-state NMR spectroscopy. *Nat Methods* **8**, 919-931, doi:10.1038/nmeth.1735 (2011).
- 6 Cruz, J. A. & Westhof, E. The dynamic landscapes of RNA architecture. *Cell* **136**, 604-609, doi:10.1016/j.cell.2009.02.003 (2009).
- 7 Frauenfelder, H., Sligar, S. G. & Wolynes, P. G. The energy landscapes and motions of proteins. *Science* **254**, 1598-1603, doi:10.1126/science.1749933 (1991).
- 8 Spitale, R. C. & Incarnato, D. Probing the dynamic RNA structurome and its functions. *Nat Rev Genet* **24**, 178-196, doi:10.1038/s41576-022-00546-w (2023).
- 9 Tomezsko, P. J., Corbin, V. D. A., Gupta, P., Swaminathan, H., Glasgow, M., Persad, S., Edwards, M. D., McIntosh, L., Papenfuss, A. T., Emery, A., Swanstrom, R., Zang, T., Lan, T. C. T., Bieniasz, P., Kuritzkes, D. R., Tsibris, A. & Rouskin, S. Determination of RNA structural diversity and its role in HIV-1 RNA splicing. *Nature* **582**, 438-442, doi:10.1038/s41586-020-2253-5 (2020).
- 10 Yang, M., Zhu, P., Cheema, J., Bloomer, R., Mikulski, P., Liu, Q., Zhang, Y., Dean, C. & Ding, Y. In vivo single-molecule analysis reveals COOLAIR RNA structural diversity. *Nature* **609**, 394-399, doi:10.1038/s41586-022-05135-9 (2022).
- 11 Zemora, G. & Waldsich, C. RNA folding in living cells. *RNA Biol* **7**, 634-641, doi:10.4161/rna.7.6.13554 (2010).
- 12 Ken, M. L., Roy, R., Geng, A., Ganser, L. R., Manghrani, A., Cullen, B. R., Schulze-Gahmen, U., Herschlag, D. & Al-Hashimi, H. M. RNA conformational propensities determine cellular activity. *Nature* **617**, 835-841, doi:10.1038/s41586-023-06080-x (2023).
- 13 Kazantsev, A. V., Krivenko, A. A., Harrington, D. J., Holbrook, S. R., Adams, P. D. & Pace, N. R. Crystal structure of a bacterial ribonuclease P RNA. *Proc Natl Acad Sci U S A* **102**, 13392-13397, doi:10.1073/pnas.0506662102 (2005).
- 14 Pan, T., Artsimovitch, I., Fang, X. W., Landick, R. & Sosnick, T. R. Folding of a large ribozyme during transcription and the effect of the elongation factor NusA. *Proc Natl Acad Sci U S A* **96**, 9545-9550, doi:10.1073/pnas.96.17.9545 (1999).

- 15 Wong, T., Sosnick, T. R. & Pan, T. Mechanistic insights on the folding of a large ribozyme during transcription. *Biochemistry* **44**, 7535-7542, doi:10.1021/bi047560l (2005).
- 16 Kurz, J. C. & Fierke, C. A. The affinity of magnesium binding sites in the *Bacillus subtilis* RNase P x pre-tRNA complex is enhanced by the protein subunit. *Biochemistry* **41**, 9545-9558, doi:10.1021/bi025553w (2002).
- 17 Kurz, J. C., Niranjanakumari, S. & Fierke, C. A. Protein component of *Bacillus subtilis* RNase P specifically enhances the affinity for precursor-tRNA^{Asp}. *Biochemistry* **37**, 2393-2400, doi:10.1021/bi972530m (1998).
- 18 Pan, T. & Sosnick, T. R. Intermediates and kinetic traps in the folding of a large ribozyme revealed by circular dichroism and UV absorbance spectroscopies and catalytic activity. *Nat Struct Biol* **4**, 931-938, doi:10.1038/nsb1197-931 (1997).
- 19 Hsieh, J., Andrews, A. J. & Fierke, C. A. Roles of protein subunits in RNA-protein complexes: lessons from ribonuclease P. *Biopolymers* **73**, 79-89, doi:10.1002/bip.10521 (2004).
- 20 Crary, S. M., Niranjanakumari, S. & Fierke, C. A. The protein component of *Bacillus subtilis* ribonuclease P increases catalytic efficiency by enhancing interactions with the 5' leader sequence of pre-tRNA^{Asp}. *Biochemistry* **37**, 9409-9416, doi:10.1021/bi980613c (1998).
- 21 Sharkady, S. M. & Nolan, J. M. Bacterial ribonuclease P holoenzyme crosslinking analysis reveals protein interaction sites on the RNA subunit. *Nucleic Acids Res* **29**, 3848-3856, doi:10.1093/nar/29.18.3848 (2001).
- 22 Tsai, H. Y., Masquida, B., Biswas, R., Westhof, E. & Gopalan, V. Molecular modeling of the three-dimensional structure of the bacterial RNase P holoenzyme. *J Mol Biol* **325**, 661-675, doi:10.1016/s0022-2836(02)01267-6 (2003).
- 23 Niranjanakumari, S., Stams, T., Crary, S. M., Christianson, D. W. & Fierke, C. A. Protein component of the ribozyme ribonuclease P alters substrate recognition by directly contacting precursor tRNA. *Proc Natl Acad Sci U S A* **95**, 15212-15217, doi:10.1073/pnas.95.26.15212 (1998).
- 24 Ganser, L. R., Kelly, M. L., Herschlag, D. & Al-Hashimi, H. M. The roles of structural dynamics in the cellular functions of RNAs. *Nat Rev Mol Cell Biol* **20**, 474-489, doi:10.1038/s41580-019-0136-0 (2019).
- 25 Faloutsos, C. Getting to Know Your Data. *Mor Kauf D*, 39-82, doi:Book_Doi 10.1057/9780230289895 (2012).
- 26 Franke, D., Jeffries, C. M. & Svergun, D. I. Correlation Map, a goodness-of-fit test for one-dimensional X-ray scattering spectra. *Nat Methods* **12**, 419-422, doi:10.1038/nmeth.3358 (2015).

Reviewer #2

1. Let us know that the methods paper is now accepted and that they hold publication for back-to-back appearance with this paper. I have no problem with this.

We appreciate the reviewer's comment.

2. The authors let me know that this was addressed in the former review cycle, meaning that they dismissed the comment. The authors say "A pool of discrete snapshot structures has been used to derive molecular motions in proteins^{1,2}, and we aren't the first to do so.". Do the authors not see that this statement is wrong? Snapshots of structure do not allow to derive motions. It doesn't really matter if the authors are not the first to do so or not. Readers with background in physics will disagree with it.

Let us try another non-academic way to convey the idea: please think how a motion picture is made, or how a movie is made. It is made using many quick snapshots with time intervals less than human eyes can detect so that the continuous plays of these snapshots of a motion event in sequence give rise to a movie depicting the movements of people or objects. Taking snapshots of conformers and placing them in a "sequential" order using a method called PCA works metaphorically in a similar way.

3. The authors let me know "We have addressed this question in the previous response. In short, we selected isolated particles based on image quality free of damages caused by the AFM probe, or image artifacts caused by instrument random instability." But they do not let me know how they discern between these experimental effects and conformational variations. Also, they let me know "It took about 24 million CPU hours (10,000 CPU for 100 days nonstop) to compute the structures of 158 conformers, not to mention the time spent optimizing the computational protocols.", which was not what I wanted to know, but hoped to learn if each conformer was an aggregate of many particles.

We have detailed explanations in the previous rebuttals. In the meantime, the reviewer has been provided with our method paper as supplementary material for this review. In the method paper, we have addressed the question and provided the authoritative literature.

Overall, I do not find that I get into a fruitful scientific exchange with the authors in the review experience.

The authors have no further response.

I congratulate the authors for the beautiful data and the ground-breaking achievement of extracting statistical ensemble distributions from AFM data, and regret their stubbornness to engage meaningfully into the review process and change/amend their work in the process

beyond their original thoughts.

We thank the reviewer for appreciating our work and more importantly for taking his/her time to review our work.

Referee #4:

My main critique focused on the possibility of mica surface immobilization impacting the AFM-sampled ensemble of conformers. The authors cite references 25-30 as prior evidence for the lack of such effects. I was not able to locate those references in either the edited main text, where refs. 25-30 cover NMR and FRET, or the rebuttal letter, where references end at 26 and refs. 25-26 also have nothing to do with AFM. I thank the authors for supplementing these references with the link to the plain-language document for the Bruker-sponsored white paper. The linked brochure includes evidence of preservation of the major and minor grooves in DNA upon surface immobilization. The authors highlight that point in their response stating that the same interactions are responsible for formation of the fully base paired double helices as those driving the complete process of RNA folding. I agree that the hydrogen binding or base stacking interactions are universal. However, I would argue for differences between the energetics of a cooperatively formed Watson-Crick paired helices and structural elements containing just a handful of hydrogen bonds such as those that may be present in the inter-helical loops. Therefore, the fact that completely base-paired helices roughly retain their groove parameters upon immobilization, within the 10-15Å resolution of the AFM images, does not prove that the same preservation applies to weaker interactions that may be determining the overall architecture of flexible RNAs such as the one studied here.

We understand the reviewer's concern and apologize for not listing the relevant reference in the rebuttal. The authors had assumed that the reviewer has the procession of all previous reviews and rebuttals for this manuscript and another related manuscript provided as supplementary material for this review. In this feedback, we provide the list of relevant papers addressing the question hoping to alleviate the concern.

- Atomic Force Microscopy Imaging and Probing of DNA, Proteins, and Protein-DNA Complexes: Silatrane Surface Chemistry. In: Leblanc, B., Moss, T. (eds) DNA-Protein Interactions. Methods in Molecular Biology™, vol 543. Humana Press. 2009.
- DNA structure and dynamics: an atomic force microscopy study. Cell Biochem Biophys 2004;41(1):75-98. doi: 10.1385/CBB:41:1:075
- Shlyakhtenko, L.S. et al. Silatrane-based surface chemistry for immobilization of DNA, protein-DNA complexes and other biological materials. Ultramicroscopy 97, 279-87 (2003).
- Lyubchenko, Y.L., Shlyakhtenko, L.S. & Gall, A.A. Atomic force microscopy imaging and probing of DNA, proteins, and protein DNA complexes: silatrane surface chemistry. Methods Mol Biol 543, 337-51 (2009).

- Stumme-Diers, M.P., Stormberg, T., Sun, Z. & Lyubchenko, Y.L. Probing The Structure And Dynamics Of Nucleosomes Using Atomic Force Microscopy Imaging. J Vis Exp (2019).
- Shlyakhtenko and Lyubchenko. Mica Functionalization for Imaging of DNA and Protein-DNA Complexes with Atomic Force Microscopy. Methods Mol Biol. 2013; 931: 10.1007/978-1-62703-056-4_14.

The authors have modified the text stating the AFM ensemble members “represent the sampling of the RNA’s conformational space at the time the molecules were immobilized on the mica surface” (p 4), covering “the nearly full if not complete sampling of the conformational space” (the top of p 7) and representing “a conformational sampling of the solution ensemble at 1 mM Mg²⁺” (the beginning of the Discussion section on p 14). While interaction of the RNA with the mica surface is likely non—instantaneous and may include some conformation reorganization, the authors have provided new evidence to support their claims based on the PCA analysis, the comparison to cryo-EM data at 10mM Mg²⁺, and the SAXS data. The results of both the PCA analysis and the cryo-EM data comparison appear compelling. The details on the SAXS data analysis included in the rebuttal letter also corroborate the authors’ claims as the appropriately weighted AFM ensemble provides an excellent fit to the SAXS data. The caveat here is the relatively low resolution of the scattering data and the relatively high noise at high q. However, these issues are nearly universal in the field and these results are as good as can be expected.

We appreciate the reviewer’s comments.

In summary, the authors have provided a sizeable amount of new details and analysis to more thoroughly back up the main claims of the paper, which are indeed quite remarkable. I feel that the manuscript can be accepted for publication in its edited form.

We thank the reviewer for the recommendation.